# Random Circuits in the Black Hole Interior

Javier M. Magán[†], Martin Sasieta[§], Brian Swingle[‡]

[†] *Instituto Balseiro, Centro Atómico Bariloche*
*8400-S.C. de Bariloche, Río Negro, Argentina*

[§] [‡] *Martin Fisher School of Physics, Brandeis University*
*Waltham, Massachusetts 02453, USA*

### ABSTRACT

In this paper, we present a quantitative holographic relation between a microscopic measure of randomness and the geometric length of the wormhole in the black hole interior. To this end, we perturb an AdS black hole with Brownian semiclassical sources, implementing the continuous version of a random quantum circuit for the black hole. We use the random circuit to prepare ensembles of states of the black hole whose semiclassical duals contain *Einstein-Rosen (ER) caterpillars*: long cylindrical wormholes with large numbers of matter inhomogeneities, of linearly growing length with the circuit time. In this setup, we show semiclassically that the ensemble of ER caterpillars of average length $k\ell_\Delta$ and matter correlation scale $\ell_\Delta$ forms an approximate quantum state $k$-design of the black hole. At exponentially long circuit times, the ensemble of ER caterpillars becomes polynomial-copy indistinguishable from a collection of random states of the black hole. We comment on the implications of these results for holographic circuit complexity and for the holographic description of the black hole interior.

[†] javier.magan@cab.cnea.gov.ar
[§] martinsasieta@brandeis.edu
[‡] bswingle@brandeis.edu

# 1 Introduction

The exterior description of black holes has been fairly well established over the last decades, as a result of the development of microscopic models of black holes in AdS/CFT [1–3] and models like SYK [4, 5] describing their low-energy fluctuations. Black holes behave as strongly interacting systems, as some of their thermal and hydrodynamic features universally suggest (see, e.g., [6–11]). A more refined characteristic of black holes is the particularly strong quantum chaos that they exhibit. This manifests at early times in their rapid equilibration at the thermal scale, or fast scrambling, as quantified by their distinctively large chaos exponent [12–16].

On the contrary, the description of the black hole interior remains far more elusive. The general expectation stemming from holography is that the black hole interior is described microscopically by the same system that describes the exterior. However, identifying explicit microscopic quantities of the holographic system that do this remains largely an open problem, even in controlled scenarios such as AdS/CFT.

Quantum complexity was proposed in this context as a central aspect in the microscopic description of the black hole interior [17,18]. The main conjecture is that the quantum complexity of the time-evolution operator of the black hole, understood in the circuit model of quantum computation, serves as a measure of the total amount of emergent space in the interior of black holes. The motivation behind the conjecture essentially lies in the observation that space in the black hole interior grows linearly in time after thermalization, with different holographic measures which capture this growth [17–25]. This growth resembles the universal linear growth of quantum complexity of time evolution expected for generic quantum many-body systems. The linear growth should persist for exponentially long times in the entropy after thermalization [26].

Despite the potential significance of such a relation, there is currently no *ab initio* connection between quantum circuit complexity and the black hole interior. Among the different holographic complexity proposals, the Complexity = Volume (CV) conjecture [19] is arguably the most suggestive one in this sense. The maximal volume slice that CV refers to qualitatively resembles a quantum circuit, one which grows in time under the action of the microscopic time-evolution operator of the black hole acting at the ends of the circuit. The idea that the black hole interior is a quantum circuit is consistent with tensor network models of the emergent bulk in the black hole exterior, parametrizing the entanglement structure of holographic states with thermal features [27]. The circuit picture is further supported by the behavior of HRT surfaces in extended interiors [28], and by the fact that the volume of the maximal slices captures the phenomenology of operator complexity growth [29–36], among other features [37–45].[1] Nevertheless, at a more quantitative level, the ambiguities in the definition of circuit complexity, as well as the lack of realistic dynamical tensor network models of gravity, make this connection somewhat vague.

In this paper, we will be motivated by the quantum circuit picture of the black hole interior, with the aim of finding a precise sense in which this picture is realized for the black hole. In order to do this, we begin with the observation that the dynamical evolution of the black hole is

---

[1] Holographic complexity conjectures have also inspired rigorous results in simpler models, e.g. [46,47], proposals in CFTs [48–52] and other notions like spread/Krylov complexity with geometric avatars [53–58].

only able to generate a very particular "quantum circuit", namely, the time-evolution operator $\exp(-\mathrm{i}tH_0)$ with the black hole Hamiltonian $H_0$. Such a "quantum circuit" cannot be generic, given that $H_0$ is conserved along the evolution. In order to construct more general circuits, we will exploit the fact that we can perturb $H_0$ with time-dependent sources and instead drive the black hole with time-dependent Hamiltonians of the form

$$H(t) = H_0 + \sum_{\alpha=1}^{K} g_\alpha(t)\, \mathcal{O}_\alpha \,. \tag{1.1}$$

We will choose the perturbations $\{\mathcal{O}_\alpha\}$ from a collection of $K$ Hermitian operators that possess a semiclassical description in terms of low-energy matter fields. In full-fledged holographic CFTs, these operators could correspond to smeared insertions of single-trace conformal primaries of low conformal dimension. Here we label these operators abstractly with the index $\alpha$, keeping in mind that $\alpha$ can include space and time coordinates defining the operator.

In this way, the time-evolution operator of the black hole will instead correspond to

$$U(t) = \mathsf{T} \exp\left(-\mathrm{i}\int_0^t \mathrm{d}s\, H(s)\right). \tag{1.2}$$

Throughout this paper, we will select the couplings $g_\alpha(t)$ for the perturbations from an ensemble of independent white-noise correlated gaussian random couplings, with

$$\mathbb{E}[g_\alpha(t)] = 0\,, \qquad \mathbb{E}[g_\alpha(t)g_{\alpha'}(t')] = J\delta(t-t')\delta_{\alpha\alpha'}\,. \tag{1.3}$$

We will normalize the perturbations $\mathcal{O}_\alpha$ to be dimensionless, so that the couplings $g_\alpha(t)$ and the Brownian step $J$ both have dimensions of energy.

The disordered couplings define an ensemble of time-evolution operators of the black hole at any fixed time $t$, which we will denote by $\mathcal{E}_U^t$. For this choice of Brownian couplings, and sufficiently large number of perturbations $K \gg 1$, a typical draw $U(t)$ from $\mathcal{E}_U^t$ will resemble a continuous version of a random quantum circuit, given that Trotterizing $U(t)$ infinitesimally, $U(t) = \prod_i U(t_i, t_i + \delta t)$, the couplings of the large number of operators forming the unitaries at each time step $U(t_i, t_i + \delta t) = \mathbf{1} - \mathrm{i}\delta t H(t_i)$ are chosen independently and at random.

## 1.1 Outline and summary of results

In section 2 we use the ensemble $\mathcal{E}_U^t$ of random circuits $U(t)$ to create an ensemble of states $\mathcal{E}_\Psi^t$ of a two-sided black hole at fixed circuit time $t$. We argue that applying the random circuit $U(t)$ directly from the black hole exterior produces substantial gravitational backreaction on the black hole at circuit times of order $Jt \sim S/K$, where $S$ is the Bekenstein-Hawking entropy of the black hole. To be able to run the circuit for much longer times in a controlled manner, we will gradually cool the black hole down while injecting the time-dependent perturbations.

In section 2.1 we implement a steady perturbing-and-cooling process using a suitable complexified Schwinger-Keldysh time contour in the CFT path integral preparation of the states. This avoids the previous issue since the perturbations are introduced directly in the black hole

interior, and the exterior regions remain equilibrated. In this way, we will define an ensemble of states $\mathcal{E}_{\Psi}^{t}$ at fixed circuit time $t$, where each realization $|\Psi(t)\rangle \in \mathcal{E}_{\Psi}^{t}$ corresponds to a particular realization of the gradually cooled random circuit used to prepare the state.

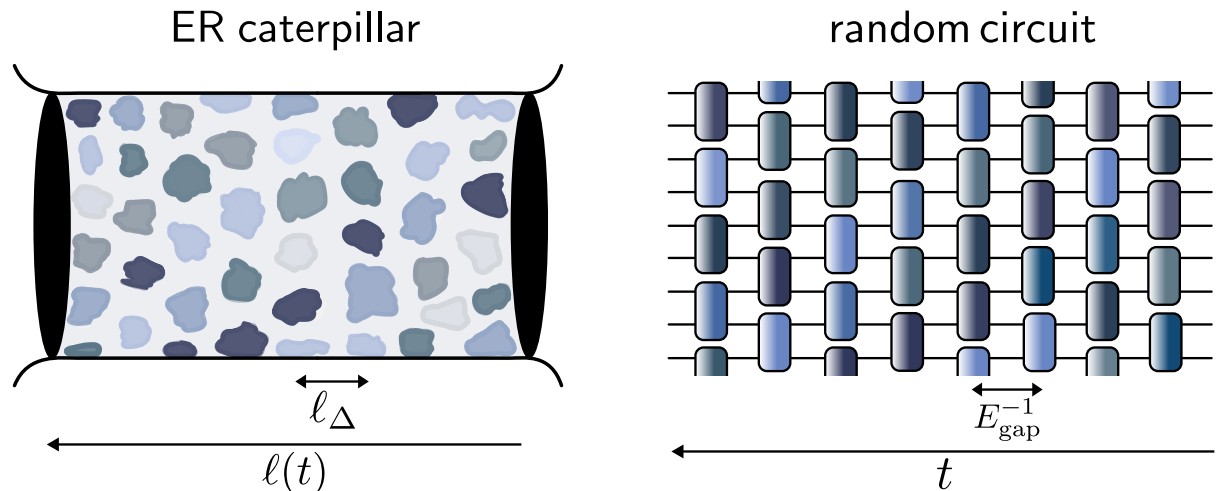

**Figure 1:** The semiclassical dual to a state $|\Psi(t)\rangle \in \mathcal{E}_{\Psi}^{t}$ contains an ER caterpillar (bumpy wormhole with matter inhomogeneities) of linearly growing length $\ell(t)$ with the circuit time $t$. The largest separation scale between matter inhomogeneities is set by $\ell_{\Delta}$, where $\Delta$ refers to the conformal dimension of the lightest bulk matter field on the wormhole. On the right, a discrete toy model of the boundary random quantum circuit used to prepare the microscopic state. The circuit time is set by $E_{\text{gap}}^{-1}$. The individual gates of the random circuit are not unitary since the preparation involves gradual cooling.

In section 2.2 we argue, on general grounds, that the semiclassical duals to the states in $\mathcal{E}_{\Psi}^{t}$ contain *Einstein-Rosen caterpillars*: long cylindrical wormholes with a large number of matter inhomogeneities spread throughout the wormhole.[2] An illustration of an ER caterpillar is presented in Fig. 1. The ensemble of states $\mathcal{E}_{\Psi}^{t}$ defines an ensemble of ER caterpillars, where different draws of $\mathcal{E}_{\Psi}^{t}$ contain wormholes with different semiclassical details. The collection of ER caterpillars will nevertheless share some coarse-grained geometric features. Most importantly for our purposes, the ensemble of ER caterpillars has an average cylindrical wormhole geometry, with a wormhole length $\ell(t)$ scaling linearly with the circuit time used to prepare the states

$$\frac{\ell(t)}{\ell_{\Delta}} = E_{\text{gap}}(t - t_{\star}).\tag{1.4}$$

Here $\ell_{\Delta}$ is the *matter correlation scale* of the ER caterpillar. It sets the largest longitudinal characteristic scale of the matter inhomogeneities throughout the wormhole. Such scale is associated to the bulk matter field with the smallest conformal dimension $\Delta$. On the other hand, the scale $E_{\text{gap}}^{-1}$ is what plays the role of the circuit time in discrete toy models of the boundary random circuit. The energy scale $E_{\text{gap}}$ depends on $J$ and on other scales of the holographic system. The timescale $t_{\star}$ sets the onset time of the linear growth.

In section 2.3 we show that (1.4) is satisfied for a collection of ER caterpillars of near-extremal

---

[2] This term was first proposed in [59] in the context of the saturation of complexity at exp times [60].

black holes, where we compute $E_{\text{gap}}$, $\ell_\Delta$ and we estimate $t_\star$ for these systems. In this case, the states in $\mathcal{E}_\Psi^t$ are low-energy states in two copies of the SYK model. Our explicit construction centrally relies on the connection between the preparation of the states in $\mathcal{E}_\Psi^t$ and the preparation of the eternal traversable wormhole of Maldacena and Qi [61].

In section 2.4 we present additional properties of the ensemble $\mathcal{E}_\Psi^t$. We argue that the reduced states $\rho_{\text{L}}$ and $\rho_{\text{R}}$ for (1.5) correspond to an equilibrium state $\rho_{\text{eq}}$. The particular form of $\rho_{\text{eq}}$ depends on specific details of the ensemble of ER caterpillars, e.g., on the nature of the perturbing operators $\{\mathcal{O}_\alpha\}$, the number of perturbations $K$, the coupling $J$, etc. However, under reasonable and generic assumptions, it follows that in the thermodynamic limit, $\rho_{\text{eq}}$ has non-vanishing overlap with the canonical Gibbs state $\rho_\beta$ at some coupling-dependent temperature $\beta(J)$. Here $\rho_\beta$ is the canonical Gibbs state $\rho_\beta = \exp(-\beta H_0)/Z(\beta)$, where $\beta = T_H^{-1}$ is the inverse Hawking temperature of the black hole and $Z(\beta)$ is its canonical partition function. The difference between $\rho_{\text{eq}}$ and $\rho_\beta$ turns out to be just a difference in energy variance.

In section 3 we quantify the amount of microscopic randomness defining the ensemble $\mathcal{E}_\Psi^t$ of ER caterpillars, as a function of the circuit time $t$. We do this by comparing $\mathcal{E}_\Psi^t$ to an ensemble of random states of the two-sided black hole, $\mathcal{E}_\Psi^{\text{rand}}$. The random state ensemble consists of *random equilibrium states* of the form

$$|\Psi_{\text{rand}}\rangle \propto \sum_{i,j} \left(\sqrt{\rho_{\text{eq}}}\, U \sqrt{\rho_{\text{eq}}}\right)_{ij} |E_i^*\rangle_{\text{L}} \otimes |E_j\rangle_{\text{R}} \ , \tag{1.5}$$

where $U$ is a Haar random unitary matrix and $\rho_{\text{eq}}$ is the equilibrium density matrix mentioned above. Notice that, due to the presence of $\sqrt{\rho_{\text{eq}}}$ factors, the states (1.5) are not totally random states in the Hilbert space of the two holographic CFTs, as the latter would correspond to states of infinite energy. The entries of $\sqrt{\rho_{\text{eq}}}$ control the envelope of the wavefunctions, which decay at high energies. The reduced density matrices for (1.5) are approximately $\rho_{\text{eq}}$. Thus, the state $|\Psi_{\text{rand}}\rangle$ corresponds to a random purification of $\rho_{\text{eq}}$.

At infinite circuit time, the ensemble $\mathcal{E}_\Psi^\infty$ becomes effectively indistinguishable from $\mathcal{E}_\Psi^{\text{rand}}$. On the other hand, at finite circuit time $t$, the ensemble $\mathcal{E}_\Psi^t$ is at most quasi-random. With enough draws and precision, $\mathcal{E}_\Psi^t$ is distinguishable from $\mathcal{E}_\Psi^{\text{rand}}$. We will quantify the approach to random using the quantum information theoretic notion of a *quantum state $k$-design*: an ensemble of states which reproduces the first $k$ moments of the random state ensemble. Quantum state $k$-designs are $k$-copy indistinguishable from the random state ensemble. As opposed to random states, approximate quantum state $k$-designs form efficiently – at sub-exponential timescales in the entropy of the system – in physical systems, such as in discrete-time random circuits, or dynamically in chaotic many-body quantum systems. For this reason, the formation of approximate $k$-designs is of central relevance in quantum information theory (see, e.g., [62–67]) and in many-body quantum chaos [68–76]. Unitary designs have been used in models of the dynamics of evaporating black holes [77].

The central result of section 3 is that the ensemble of ER caterpillars $\mathcal{E}_\Psi^t$ becomes an approximate quantum state $k$-design of the black hole in a circuit time which, up to logarithmic factors, scales linearly with $k$. Moreover, the slope of this linear growth will be controlled by the

same energy scale $E_{\text{gap}}$ that appears in (2.40), thereby implying a randomness-length relation.

In more detail, we will study the proximity of $\mathcal{E}_\Psi^t$ to a $k$-design in terms of a convenient notion of quantum information theoretic distance[3]

$$\text{distance to } k\text{-design} = F_k(t) - F_k(\infty),\tag{1.6}$$

defined in terms of the *k-th frame potential* of the ensemble of states $\mathcal{E}_\Psi^t$,

$$F_k(t) \equiv \mathbb{E}_{\mathcal{E}_\Psi^t} |\langle \Phi(t)|\Psi(t)\rangle|^{2k}.\tag{1.7}$$

Given the way in which the states in $\mathcal{E}_\Psi^t$ are prepared with the random circuit, $F_k(t)$ corresponds to a disordered $2k$-replica partition function of the holographic CFT. Effectively, the disorder over couplings (1.3) produces local-in-time interactions between replicas in the form of a time-independent Hamiltonian $H_{\text{eff},k}$. The frame potential essentially corresponds to the thermal partition function of $2k$ interacting copies of the holographic CFT at inverse temperature $2t$,

$$F_k(t) \propto \text{Tr}\exp(-2tH_{\text{eff},k}).\tag{1.8}$$

In section 3.1 we study the behavior of $F_k(t)$ microscopically in the microcanonical window of the two-sided black hole. In section 3.2 we use the path integral of gravity to evaluate the frame potentials $F_k(t)$ in a saddle point approximation. The disorder is introduced in the bulk via the Brownian boundary conditions for the bulk fields dual to the perturbations. The disorder average creates effective bi-local interactions between replicas. These interactions support connected Euclidean wormhole contributions to the frame potentials, as shown in Fig. 2 for $F_1(t)$.

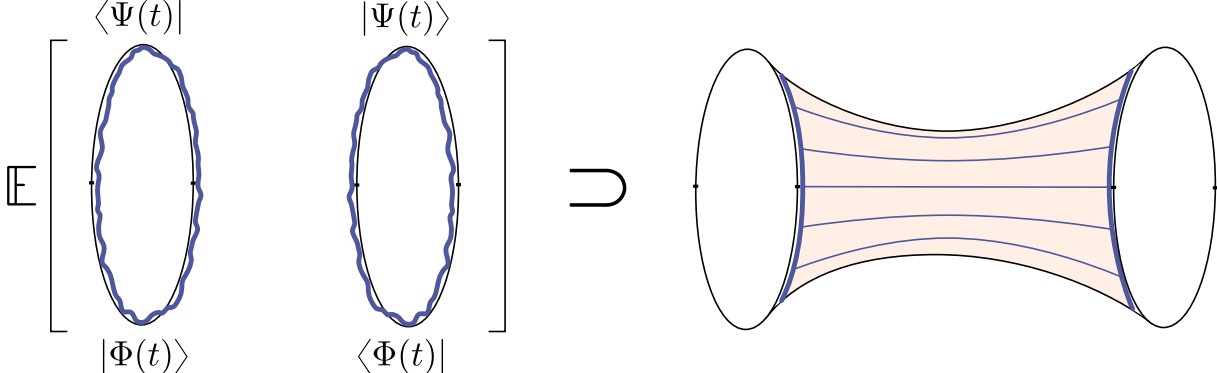

**Figure 2:** On the left, the first frame potential $F_1(t)$ of the ensemble of ER caterpillars. The classical correlations of the Brownian couplings between the two replicas generate time-independent interactions and the frame potential $F_1(t)$ corresponds to an effective thermal partition function of two interacting holographic systems at inverse temperature $2t$. For a large number of perturbations, $F_1(t)$ receives a semiclassical contribution from a connected Euclidean wormhole stabilized by the effective interactions.

At infinite time, or zero effective temperature in (1.8), we will argue that the two-replica wormhole saddle-point configuration presented in Fig. 2 suffices to account for $F_k(\infty)$, which

---

[3] For the sake of clarity, we will omit finite temperature subtleties in the introduction.

only depends on the dimensionality of the ground space of the effective Hamiltonian $H_{\text{eff},k}$. The ground states of $H_{\text{eff},k}$ break the replica symmetry of the Hamiltonian and are captured by $k$ copies of the two-boundary wormhole. In section 3.3 we construct this wormhole for near-extremal black holes and relate it to the stabilized double trumpet wormhole constructed by Maldacena and Qi [61].

The distance to design $F_k(t) - F_k(\infty)$ is then governed by the gapped fluctuations around this wormhole. More precisely, this wormhole produces a decay of the frame potential of the form

$$F_k(t) - F_k(\infty) \approx N_* \, k! \, k \, e^{-2tE_{\text{gap}}} \, , \tag{1.9}$$

where $N_*$ is the number of first excited states on the wormhole, and $E_{\text{gap}}$ is their energy. The factor of $k!$ corresponds to the number of two-replica wormhole configurations, or independent ground states of $H_{\text{eff},k}$. The factor of $k$ arises from the number of single-replica excitations on top of each ground state. This behavior of the frame potential at late times is expected more generally in Brownian quantum many-body systems at infinite temperature [75, 76].[4]

The behavior (1.9) produces a growth in design which, up to logarithmic factors, is linear in the circuit time. The ensemble $\mathcal{E}_\Psi^t$ will become an approximate quantum state $k$-design of the black hole in a time

$$t_k \approx (2E_{\text{gap}}^{-1})k + t_\varepsilon \, , \tag{1.10}$$

where $t_\varepsilon = (2E_{\text{gap}}^{-1}) \log \varepsilon^{-1}$ is a timescale which depends on the resolution $\varepsilon$ in the approximate notion of a $k$-design.

Eqs. (1.4) and (1.10) imply a direct relationship between the amount of randomness defining the ensemble of ER caterpillars and the average geometric length of the wormhole. More precisely, the implication is that the ensemble of ER caterpillars of average length $\ell$ and characteristic matter correlation scale $\ell_\Delta$ defines an approximate quantum state $k$-design of the black hole for

$$k \approx 2 \, \frac{\ell - \ell_\varepsilon}{\ell_\Delta} \, , \tag{1.11}$$

where $\ell_\varepsilon = \ell_\Delta \log \varepsilon^{-1}$. The main result of this paper is Eq. (1.11), represents a quantitative relation between a microscopic notion of "quantum complexity" associated with the generation of randomness by the circuit and the geometry of the wormhole. It formalizes the idea that the geometry of the ER caterpillars behaves as a bulk random quantum circuit, with total elapsed time $\ell$ and a circuit time $\ell_\Delta$ set by the separation of matter inhomogeneities on the ER caterpillar.

In section 4 we consider the microscopic linear growth of randomness (1.10) for circuit times scaling exponentially with the entropy of the black hole. In this regime, the general expectation is that the ensemble of ER caterpillars $\mathcal{E}_\Psi^t$ is polynomial-copy indistinguishable from the ensemble $\mathcal{E}_\Psi^{\text{rand}}$ (1.5) of random states of the black hole. We provide microscopic arguments in favor of an ever-lasting linear growth of randomness, and contrast this with the saturation of randomness

---

[4] Our approach has precursors in the study of convergence of moment superoperators, including [78–82]. There has also been rapid progress on the rigorous construction of approximate unitary $k$-designs, including [83–91].

for any observer with finite resolution in Hilbert space.

In section 5 we discuss potential implications of (1.11). These include consequences for holographic circuit complexity, the holographic description of the black hole interior, and firewalls.

We end with appendices that describe supplementary material:

In appendix A we discuss alternative possibilities to apply the random circuit to the black hole while avoiding substantial gravitational backreaction for long times.

In appendix B we discuss a simple toy model for the ensemble of ER caterpillars. The toy model allows us to analyze relevant geometric properties of the ensemble explicitly in any dimension.

In appendix C we discuss the phase structure of the $2k$-replica interacting Hamiltonian $H_{\text{eff},k}$. We mainly focus on its ground state and first excited states. We comment on the generic replica symmetry breaking of the ground space at sufficiently high values of the coupling $J$, which is what we observe in gravity for the case of the black hole.

In appendix D we consider aspects of $H_{\text{eff},k}$ associated with the ensemble of ER caterpillars in the SYK model. We write down the Schwinger-Dyson equations of the model and solve them explicitly at infinite temperature, making contact with previous work [61, 75, 92].

In appendix E we discuss measures of randomness growth which are intrinsic to the ensemble of random circuits $\mathcal{E}_U^t$. These measures refer to the notion of a unitary $k$-design.

In appendix F we provide an extended comparison between the first frame potential of the ensemble of random circuits $\mathcal{E}_U^t$ and the spectral form factor of the Hamiltonian of the black hole $H_0$. We discuss the gravitational origin of the connected correlations of the frame potential.

In appendix G we present asymptotic formulas for the moments of the Haar distribution. These moments have a nice combinatorial interpretation, which has been discussed in detail in the mathematical literature over the past two decades.

## 2 Einstein-Rosen caterpillars

As illustrated in Fig. 3 a way to try to prepare random states of the black hole is to directly apply the random circuit $U(t)$ on the right system to a high-temperature TFD state on $\mathcal{H}_L \otimes \mathcal{H}_R$. This generates states of the form

$$|\Psi(t)\rangle = (\mathbf{1}_L \otimes U(t)_R)\,|\text{TFD}\rangle\,, \tag{2.1}$$

where

$$|\text{TFD}\rangle = \frac{1}{\sqrt{Z(\beta)}} \sum_i e^{-\beta E_i/2}\,|E_i^*\rangle_L \otimes |E_i\rangle_R\,. \tag{2.2}$$

Here, the state $|E_i\rangle$ is an eigenstate of $H_0$ with energy $E_i$, and $|E_i^*\rangle \equiv \Theta\,|E_i\rangle$ for an anti-unitary operator $\Theta$ such as CPT. From (2.1) the state on the right system is the time-evolved thermal state $\rho_R(t) = U(t)\rho_\beta U(t)^\dagger$. Discrete versions of these states have appeared in different contexts in the literature, see e.g. [13, 14, 19, 20, 60, 93]. We could also apply the random circuit on both sides of the black hole, with independent couplings, or to a one-sided black hole formed by the

collapse of matter.

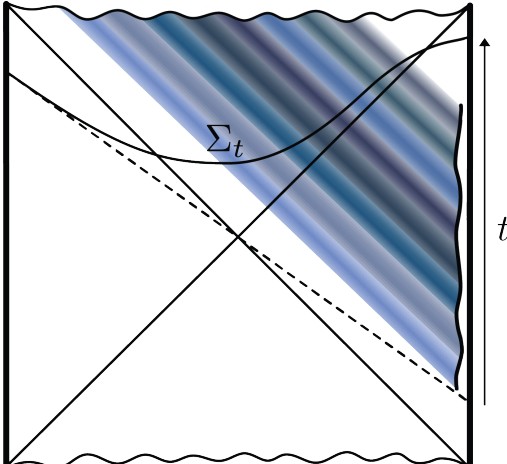

**Figure 3:** Driving the TFD with the random circuit generates an ensemble of random states of the black hole. The perturbations take a time $t_{\text{scr}} \sim \beta \log S/K$ to scramble. The dashed line represents the $t = 0$ slice. The wormhole $\Sigma_t$ is a ER caterpillar.

This way of applying the random circuit has a clear limitation which is difficult to avoid. Namely, the matter perturbations will gravitationally backreact on the black hole and increase its energy to $M(t) = M + Jt\,\delta E$, where $\delta E$ is the characteristic energy of the perturbations. Ideally, we would like for each perturbation to have the natural energy per quanta of a thermal system, $\delta E/K \sim T_H$. According to this, the backreaction of matter precludes us from running the circuit for much longer than

$$Jt_{\text{max}} \sim \frac{M}{\delta E} \sim \frac{S}{K}\,, \tag{2.3}$$

before the original black hole changes substantially.[5] This, in turn, is consistent with the fact that the black hole has a thermodynamic entropy given by $S$, and that we are introducing at least $K$ "qubits" from the exterior per time $J^{-1}$. In this paper, we will be interested in monitoring a fixed black hole for much longer circuit times, where in general $Jt$ scales polynomially with $S$. During this process we want the black hole to effectively remain in equilibrium. Therefore, we must find an alternative way to prepare states with the random circuit and avoid backreaction.

One possibility, discussed in appendix A, is to apply the random circuit from the black hole exterior using very low frequency modes of the Hawking atmosphere. In this paper, we will nevertheless follow a different route, which we describe in what follows.

## 2.1 State preparation

In the remainder of this section, we construct a family of states with the random circuit, avoiding the issue of large backreaction for very long circuit times. The states contain perturbations which are directly injected in the black hole interior. The black hole exterior regions of the states will

---

[5] If the circuit runs at a thermal rate $J \sim T_H$ and $K \ll S$ the timescale $t_{\text{max}}$ is of the order of the Page time.

be effectively equilibrated. The perturbations backreact in the black hole interior and support long wormholes. In appendix B we present a simple toy model for these states.

Physically, we want to cool the black hole down gradually while injecting the perturbations. A simple way to do this is to consider a suitable "Schwinger-Keldysh" contour $\Gamma_t$ in complexified time. As shown in Fig. 4 the contour $\Gamma_t$ interchangeably contains regions of Lorentzian evolution, where the circuit is implemented, and Euclidean evolution with $H_0$, where the state is cooled down. We will consider the evolution via

$$U(\Gamma_t) = \mathsf{P} \, \exp\left(-\mathrm{i} \int_{\Gamma_t} \mathrm{d}s \, H(s)\right),\tag{2.4}$$

where $\mathsf{P}$ represents the path-ordering induced by the contour $\Gamma_t$. On the Euclidean parts of the contour, the evolution is performed with the bare Hamiltonian $H_0$. For convenience, we will choose the Lorentzian parts of the contour evolved with $\exp(\mathrm{i}\delta t H_0)U(\delta t) = U_I(\delta t)$ so that the contour has time-folds and $U_I(\delta t)$ is the time-evolution in the interaction picture. Effectively, the evolution in a time-fold via $U_I(\delta t)$ is performed with the interaction Hamiltonian

$$H_I(s) = \sum_{\alpha=1}^{K} g_\alpha(s)\mathcal{O}_\alpha(s), \qquad 0 \leqslant s \leqslant \delta t,\tag{2.5}$$

where $\mathcal{O}_\alpha(s) = \exp(-\mathrm{i}sH_0)\mathcal{O}_\alpha \exp(\mathrm{i}sH_0)$ is the operator evolved in the interaction picture.

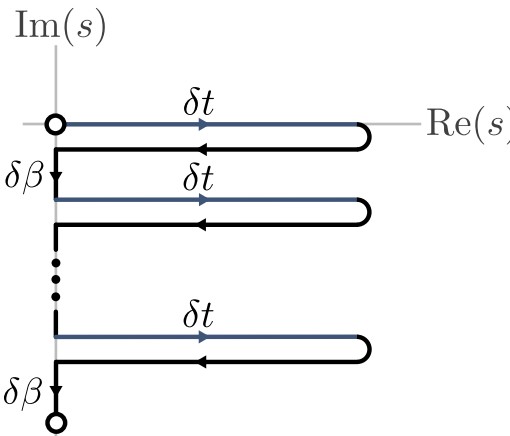

**Figure 4:** The multi Schwinger-Keldysh contour $\Gamma_t$ in the complexified $s$ plane. The Lorentzian parts of the contour include forward evolution with the random circuit $U(\delta t)$ (dark blue) and backward evolution with the bare Hamiltonian $\exp(\mathrm{i}\delta t H_0)$. Additionally, the Euclidean parts of the contour correspond to a cooling process with the bare Hamiltonian $\exp(-\delta\beta H_0)$. We take an infinitesimal version of the evolving-and-cooling process with $\delta t = \delta\beta \to 0$. The total circuit and cooling times are both given by $t$.

The disorder is introduced via the gaussian Brownian couplings of the random circuit

$$\mathbb{E}\left[g_\alpha(s)\right] = 0, \qquad \mathbb{E}\left[g_\alpha(s)g_{\alpha'}(s')\right] = J\delta(s - s')\delta_{\alpha\alpha'}.\tag{2.6}$$

The couplings are drawn independently for every $U_I(\delta t)$ on the contour.

We will take $\delta t = \delta\beta \to 0$, so that we can divide the operator $U(\Gamma_t)$ infinitesimally as[6]

$$U(\Gamma_t) = e^{-\delta\beta H_0} U_I(n, \delta t) e^{-\delta\beta H_0} \cdots e^{-\delta\beta H_0} U_I(2, \delta t) e^{-\delta\beta H_0} U_I(1, \delta t), \tag{2.7}$$

where we are explicitly labeling the Lorentzian time-evolution operators $U_I(i, \delta t)$ for $i = 1, ..., n$ to remark that the couplings are drawn independently in each of them. As we make the process infinitesimal, the number of steps goes to infinity $n \to \infty$ keeping $n\delta t = t$ fixed.

**Ensemble of states.** The ensemble of states $\mathcal{E}_\Psi^t$ of the two-sided black hole that we will study is

$$|\widetilde{\Psi}(t)\rangle = \frac{1}{\sqrt{Z(\beta)}} \sum_{i,j} e^{-\frac{\beta}{4}(E_i+E_j)} U(\Gamma_t)_{ij} |E_i^*\rangle_{\mathsf{L}} \otimes |E_j\rangle_{\mathsf{R}}, \tag{2.8}$$

where the ensemble is generated by the time-dependent disorder defining $U(\Gamma_t)$. For the time being, we can interpret $\beta$ in (2.8) as a free parameter that we can tune. We will later see that $\beta$ corresponds to the inverse temperature of the two-sided black hole that we want to study. We include the factor of $Z(\beta)$ in (2.8) for convenience.

The physical state is $|\Psi(t)\rangle = Z_\Psi(t)^{-1/2}|\widetilde{\Psi}(t)\rangle$, where $Z_\Psi(t) = \langle\widetilde{\Psi}(t)|\widetilde{\Psi}(t)\rangle$. The norm $Z_\Psi(t)$ corresponds to the CFT path integral along a closed time-contour, i.e., to the trace

$$Z_\Psi(t) = Z(\beta)^{-1} \mathrm{Tr}\left(e^{-\frac{\beta}{2}H_0} U(\Gamma_t) e^{-\frac{\beta}{2}H_0} U(\Gamma_t)^\dagger\right). \tag{2.9}$$

It will be convenient for us to write the norm as the survival amplitude

$$Z_\Psi(t) = \langle \mathrm{TFD}|\, U(\Gamma_t)_1 \otimes U(\Gamma_t)_{\bar{1}}^* \,|\mathrm{TFD}\rangle, \tag{2.10}$$

Here $|\mathrm{TFD}\rangle$ is the TFD state at inverse temperature $\beta$ between forward (1) and backward ($\bar{1}$) contours. The complex time contour $\mathcal{C}_t$ corresponding to the overlap (2.10) is shown in Fig. 5. The time-orientation of the backward contour is reversed, and $U(\Gamma_t)_{\bar{1}}^* = \Theta U(\Gamma_t)_{\bar{1}} \Theta^{-1}$, for $\Theta$ anti-unitary, is applied on this contour.

## 2.2   Average geometry

To elucidate the dual geometry of the states in $\mathcal{E}_\Psi^t$ (2.8) using the standard AdS/CFT dictionary, we start from the norm $Z_\Psi(t)$ (2.9). For individual states, $Z_\Psi(t)$ is computed in the bulk via the gravitational path integral with sources for the bulk fields given by the time-dependent couplings of the perturbations.[7] The latter is determined in a saddle point approximation by some geometry $M_\Psi$ which computes the norm of the bulk semiclassical state dual to (2.8)

$$Z_\Psi(t) \sim Z_{\mathrm{grav}}[M_\Psi], \qquad \partial M_\Psi = \mathcal{C}_t \times \mathbf{S}^{d-1}, \tag{2.11}$$

---

[6] The relative magnitude between $\delta t$ and $\delta\beta$ is absorbed into the definition of the Brownian coupling $J$.

[7] For us, the perturbations $\mathcal{O}_\alpha$ need not be local operators on $\mathbf{S}^{d-1}$. They may correspond to HKLL operators that create massive excitations deep in the bulk. Examples of this kind are broadly discussed in appendix A.

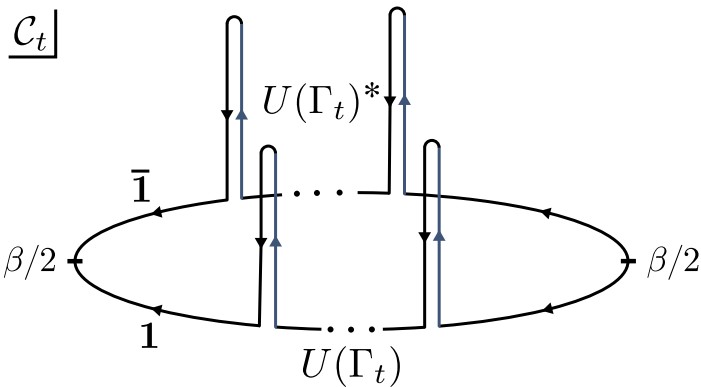

**Figure 5:** The time-contour $\mathcal{C}_t$ of the CFT path integral computing the norm $Z_\Psi(t)$, interpreted as a survival amplitude. The Euclidean evolutions on the left and right prepare unnormalized TFD states between contours 1 (bottom) and $\bar{1}$ (top). The gradually cooled random circuits $U(\Gamma_t)$ and $U(\Gamma_t)^*$ have the same time-orientation, and the Brownian couplings will be equal at equal times on both contours.

where $Z_{\text{grav}}[M_\Psi] = e^{-I_{\text{grav}}[M_\Psi]} Z_{\text{1-loop}}[M_\Psi]$ is the on-shell classical gravitational action $I_{\text{grav}}[M_\Psi]$, and $Z_{\text{1-loop}}[M_\Psi]$ is the 1-loop contribution coming from the bulk fields on $M_\Psi$. Given the complex boundary conditions, in general, the metric on $M_\Psi$ is complex, with the restriction that it must satisfy the Kontsevich-Segal-Witten criterion [94].[8]

For a specific realization of the Brownian couplings, the geometry of $M_\Psi$ is expected to be highly non-trivial, due to the time-dependent boundary conditions implemented for the bulk fields. A trick that we will use to elucidate coarse-grained properties of the ensemble is to average $Z_\Psi(t)$ over different realizations of the Brownian couplings

$$Z(t) \equiv \mathbb{E}_{\mathcal{E}_\Psi^t} \left[ Z_\Psi(t) \right] . \tag{2.12}$$

Given that the perturbations that we have chosen are semiclassical for single realizations of the couplings, $Z(t)$ will give us information about the average geometry over the ensemble of states.[9] This relies on the assumption that the norm of the states is self-averaging over the ensemble, an assumption which we corroborate semiclassically in section 2.4.

We can take the average over Brownian couplings exactly, using the identity

$$\mathbb{E} \left[ U(\Gamma_t)_1 \otimes U(\Gamma_t)_{\bar{1}}^* \right] = \exp(-t H_{\text{eff},1}) , \tag{2.13}$$

for the effective time-independent Hamiltonian

$$H_{\text{eff},1} = H_0^1 + H_0^{\bar{1}} + \frac{J}{2} \sum_{\alpha=1}^{K} \left( \mathcal{O}_\alpha^1 - \mathcal{O}_\alpha^{\bar{1} *} \right)^2 . \tag{2.14}$$

Namely, the disorder average over the Brownian couplings effectively produces local-in-time interactions between both contours in the form of a time-independent Hamiltonian $H_{\text{eff},1}$. The

---

[8] See [95, 96] for examples of bulk complex manifolds contributing to real-time thermal processes of the CFT.

[9] An alternative interpretation is that $Z(t)$ is the normalization of the average state $\tilde{\rho}_1(\mathcal{E}_\Psi^t) = \mathbb{E}[|\tilde{\Psi}(t)\rangle\langle\tilde{\Psi}(t)|]$.

quadratic form for the interaction in (2.14) arises from the choice of gaussian random couplings.

Inserting the identity (2.13) in (2.12) we find that the average norm corresponds to the matrix element

$$Z(t) = \langle \text{TFD} | \exp(-tH_{\text{eff},1}) | \text{TFD} \rangle \, . \tag{2.15}$$

As shown in Fig. 6, the CFT path integral associated with the matrix element (2.15) is now composed of two Euclidean-time contours of length $t$ with bi-local interactions among them. The average over the Brownian couplings of the perturbations has effectively gotten rid of the Lorentzian part of the contour, as a consequence of the time reflection symmetry of the ensemble of Brownian couplings. The Euclidean contours are connected together by the Euclidean preparation of the TFD bra and ket states.

The bulk manifold $M$ computing the norm of the average state,

$$Z(t) \sim \mathbb{E}_{\mathcal{E}^t_\Psi}[Z_{\text{grav}}[M_\Psi]] = Z_{\text{grav}}[M] \, , \tag{2.16}$$

is purely Euclidean, i.e., it admits a Riemannian metric. The relevant slice that determines the average geometry of the ensemble of ER caterpillars is the reflection-symmetric slice on $M$, represented in red in Fig. 6.

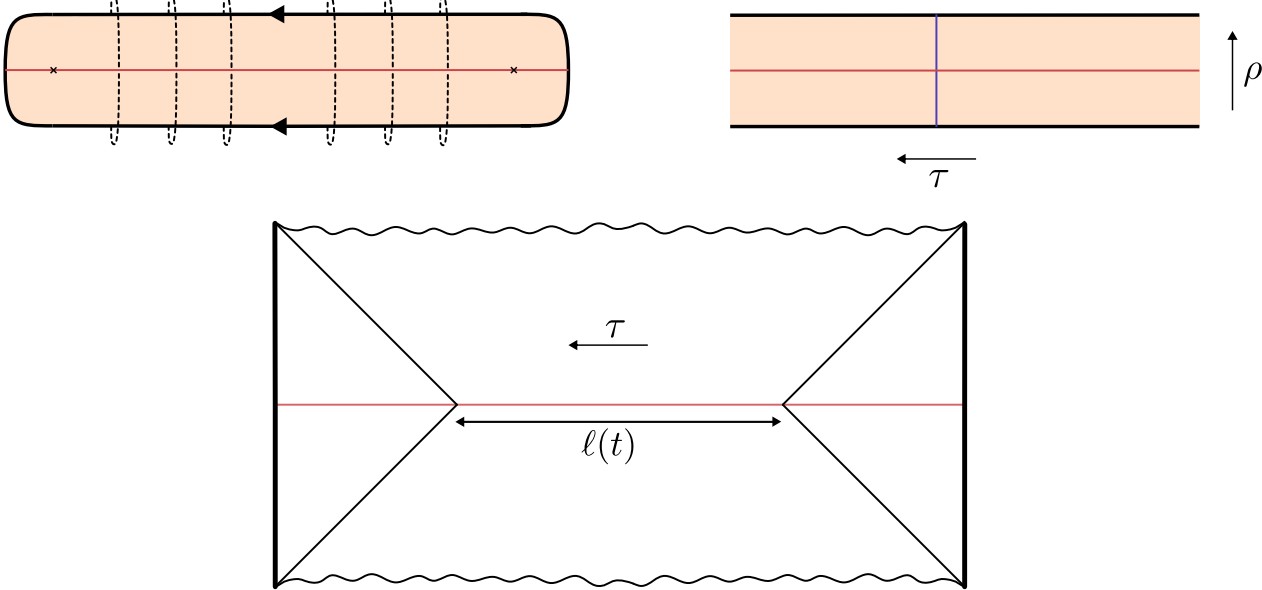

**Figure 6:** On the top left, the Euclidean time contour of the CFT path integral computing $Z(t)$. Both arrows point in the same direction, which corresponds to the Euclidean time flow of the effective Hamiltonian $H_{\text{eff},1}$ coupling both replicas. On the top right, the operator $\exp(-\tau H_{\text{eff},1}) | \text{TFD} \rangle$ prepares the semiclassical ground state of $H_{\text{eff},1}$ on the blue cut for large enough values of the time coordinate $\tau$. The geometry of $M$ is that of an Euclidean wormhole with an approximate $\tau$-translation symmetry. The average geometry of the ensemble of ER caterpillars is reflected on the red slice: a cylindrical wormhole whose length $\ell(t)$ grows linearly with the preparation time $t$. On the bottom, the Lorentzian continuation of $M$ along the red slice is an eternal black hole with a long cylindrical wormhole at the moment of Lorentzian time symmetry.

**Ground state of the effective Hamiltonian.** In order to keep the discussion general in this section, we will make the following assumption:

> *Assumption:* The effective Hamiltonian $H_{\text{eff},1}$ contains a unique ground state $|\text{GS}\rangle$, which admits a semiclassical description as a connected spatial wormhole.

We now explain why such an assumption can be physically realized in our setup.

The effective Hamiltonian $H_{\text{eff},1}$ in (2.14) contains bi-local interactions which provide a small energy when the correlations between replicas are large. Therefore, when the effect of the perturbations is large enough, the ground state $|\text{GS}\rangle$ will maximize inter-replica correlations while minimizing the energy as measured by $H_0$.

When $J \to \infty$, in the absence of symmetries, $H_{\text{eff},1}$ has a unique ground state, corresponding to the infinite temperature TFD state $|I\rangle$. At finite $J$, the ground state will not exactly be a TFD state. Some general properties however follow if $H_0$ is a strongly mixing Hamiltonian that satisfies the Eigenstate Thermalization Hypothesis (ETH) [97,98], and if $H_{\text{eff},1}$ includes a large number $K$ of simple perturbations. The number of perturbations should scale with the central charge $K = O(N^2)$ for a large-$N$ holographic CFT. Under these general conditions, it is possible to argue that the effective Hamiltonian $H_{\text{eff},1}$ is gapped and that it contains a unique ground state $|\text{GS}\rangle$ [99]. The ground state $|\text{GS}\rangle$ is diagonal in the energy basis of $H_0$ and it has a substantial $O(N^0)$ overlap with the TFD at some coupling-dependent temperature $\beta(J)$ in the thermodynamic limit [99]. The gap of $H_{\text{eff},1}$ is $O(N^0)$ in these situations. Although this class of Hamiltonian has been studied extensively in e.g. [61,99–101], we provide some numerical evidence for such a structure in appendix C.

In the holographic context, the interaction $-J\mathcal{O}_\alpha^1 \mathcal{O}_\alpha^{\bar{1}*}$ in (2.14) can be recognized as an eternal Gao-Jafferis-Wall interaction between the two replicas [102,103].[10] Such interactions are known to support wormholes, rendering them perturbatively traversable. This picture provides the bulk mechanism behind protocols of quantum teleportation in holographic systems. For a large number of perturbations, the Hamiltonian $H_{\text{eff},1}$ has the form of the Maldacena-Qi Hamiltonian [61]. In the specific construction of [61] the ground state $|\text{GS}\rangle$ corresponds to a semiclassical wormhole with a gap. Moreover, $|\text{GS}\rangle$ has substantial overlap with the TFD at some $\beta(J)$.[11]

**Geometry of $M$.** Under the assumption above, we are now ready to provide a general symmetry-based argument to show that $M$ contains a linearly growing cylindrical wormhole.

For large values of $t$, the geometry of $M$ develops an approximate Euclidean time-translation symmetry owing to the fact that $\exp(-sH_{\text{eff},1})|\text{TFD}\rangle$ will approximately prepare the unnormalized ground state of $H_{\text{eff},1}$ for any large enough value of $t_\star \lesssim s \lesssim t$. The onset time $t_\star$ of this symmetry is controlled by the gap $E_{\text{gap}}$ of $H_{\text{eff},1}$ and the number of first excited states $N_*$, and

---

[10] As mentioned in footnote 7, in our case, the operators $\mathcal{O}_\alpha$ need not be local single-trace relevant perturbations of the CFT; in particular, they could be non-local operators providing the HKLL representation of excitations deep in the bulk.

[11] In fact, the overlap is approximately 1 for sufficiently small coupling $J/\mathcal{J} \ll 1$, where $\mathcal{J}$ is the SYK coupling defining $H_0$, and it is exactly 1 for any coupling in the large-$q$ limit of the SYK model [61]. We will make a direct connection with [61] in the next subsection.

it is independent of $t$. Each constant Euclidean time slice of this geometry must prepare the semiclassical dual to the ground state (see top right in Fig. 6). For perturbations which are approximately uniformly distributed along the sphere on average, the ground state will approximately preserve spherical symmetry. This allows us to write the ansatz for the approximate bulk metric on $M$ as a Euclidean spherically-symmetric eternal traversable wormhole

$$\mathrm{d}s^2 = g(\rho)\mathrm{d}\tau^2 + \mathrm{d}\rho^2 + r(\rho)^2\mathrm{d}\Omega_{d-1}^2\,, \qquad \frac{t_\star}{2} \leqslant \tau \leqslant t - \frac{t_\star}{2}\,, \tag{2.17}$$

where $\rho \in \mathbf{R}$. The two boundaries are located at $\rho = \pm\infty$. The functions $g(\rho)$ and $r(\rho)$ are even around the center of the wormhole at $\rho = 0$ and will depend on the particular matter perturbations. Since the geometry is static, $g(\rho)$ must be non-vanishing along the wormhole.

It is more convenient to use $r$ as a coordinate and write the geometry locally as

$$\mathrm{d}s^2 = g(r)\mathrm{d}\tau^2 + \frac{\mathrm{d}r^2}{f(r)} + r^2\mathrm{d}\Omega_{d-1}^2\,, \tag{2.18}$$

where $f(r) = \frac{\mathrm{d}\rho}{\mathrm{d}r}$ has a simple zero at the center of the wormhole, at $r(\rho = 0) = r_0$, and $r \in [r_0, \infty)$ covers half of the wormhole. The energy-momentum tensor supporting (2.18) contains mass density $T_{\tau\tau}(r)$, radial pressure $T_{rr}(r)$, and angular pressure $T_{ij} = p(r)h_{ij}$, where $h_{ij}$ is the round metric on $\mathbf{S}^{d-1}$. For a perfect fluid, the radial and angular pressure coincide $T_{rr}(r) = p(r)$. The radial component of the metric reads

$$f(r) = 1 + \frac{r^2}{\ell_{\mathrm{AdS}}^2} - \frac{16\pi G m(r)}{(d-1)V_\Omega r^{d-2}}\,, \tag{2.19}$$

where $V_\Omega = \mathrm{Vol}(\mathbf{S}^{d-1})$ is the volume of the unit sphere. The mass function $m(r)$ is determined by the mass density of matter,

$$m(r) = M_0 + V_\Omega \int_{r_0}^r \mathrm{d}R\, R^{d-1}\, T_{\tau\tau}(R)\,, \tag{2.20}$$

for some constant $M_0$. The remaining component of the metric, $g(r)$, is determined additionally by $T_{rr}(r)$ and $p(r)$, with an extra equation demanding the hydrostatic equilibrium of matter on the wormhole.

The Lorentzian traversable wormhole, obtained by the $\tau \to \mathrm{i}t_e$ analytic continuation of (2.17) or (2.18), must be supported by exotic matter which violates the null energy condition (NEC), given that light rays anti-focus at the center of the wormhole. In terms of the energy-momentum tensor, this means that $T_{rr} < -T_{\tau\tau}$ at least in the vicinity of $r = r_0$. The NEC-violating matter arises semiclassically from the bi-local coupling of a large number of quanta between both boundaries in $H_{\mathrm{eff},1}$.[12] Notice, however, that the exact isometry that leads to the Lorentzian eternal traversable wormhole picture is "fake": it arises as an artifact of the average

---

[12] It is beyond the scope of this section to provide a more detailed description $T_{\mu\nu}$ starting from the bi-local interaction of a large number of operators in $H_{\mathrm{eff},1}$. In section 3.2 we explicitly describe $T_{\mu\nu}$ and the geometry of the wormhole for a specific choice of perturbations of near extremal black holes.

over the different realizations of the Brownian couplings and it is not there for a single realization. Moreover, this Lorentzian section of the geometry is not relevant for our purposes, given that the states of the ensemble do not live there.

The relevant Lorentzian section is the continuation $\rho \to i t_L$ at the moment of reflection symmetry of the Euclidean wormhole (2.17). The resulting spacetime, the black hole interior, corresponds to an anisotropic big-bang/big-crunch cosmology with approximately cylindrical slices

$$\mathrm{d}s^2 = -\mathrm{d}t_L^2 + g(t_L)\mathrm{d}\tau^2 + r(t_L)^2 \mathrm{d}\Omega_{d-1}^2\,, \qquad \frac{t_\star}{2} \leqslant \tau \leqslant t - \frac{t_\star}{2}\,. \tag{2.21}$$

In this geometry, $\tau$ is still a spatial coordinate (see Fig. 6), and the classical matter satisfies the NEC. The spheres at $t_L = 0$ and constant-$\tau$ are trapped surfaces and the cosmology crunches towards the future, generating the black hole singularity. Under the general assumption of weak cosmic censorship, this part of the spacetime must lie inside of the black hole – we will show that this is the case explicitly in a particular example in the next subsection.[13]

The spatial wormhole characterizing the average geometry of the ensemble of ER caterpillars sits at $t_L = 0$ in (2.21) (or $\rho = 0$ in (2.17)) and has geometric length $\ell(t)$ and volume $\mathrm{Vol}(t)$ given by

$$\ell(t) = g_0\left(t - t_\star\right)\,, \tag{2.22}$$

$$\mathrm{Vol}(t) = V_\Omega\, g_0\, r_0^{d-1}\left(t - t_\star\right)\,, \tag{2.23}$$

The constant $g_0$ can be fixed relative to the gap of the effective Hamiltonian $H_{\mathrm{eff},1}$. The lightest excitations on the wormhole are matter fields $\phi_\Delta$ of smallest possible conformal dimension $\Delta$. These excitations are gapped. In the bulk, the correlation lengthscale can be found from the two-point function on the wormhole,

$$\langle \phi_\Delta(x)\phi_\Delta(0)\rangle_M \sim e^{-x/\ell_\Delta}\,. \tag{2.24}$$

Here $x$ is a proper length coordinate on the $\rho = 0$ slice of the Euclidean wormhole, measuring the longitudinal separation of the operator insertions. The *matter correlation scale* is set by $\ell_\Delta$.[14] On the other hand, from the gap of $H_{\mathrm{eff},1}$, the same two-point function computed on the boundary yields

$$\langle \phi_\Delta(x(s))\phi_\Delta(0)\rangle_{CFT} \sim e^{-sE_{\mathrm{gap}}}\,, \tag{2.25}$$

where $s$ is the boundary Euclidean time associated to the bulk coordinate $x$. The relation between the proper length coordinate $x$ on the $\rho = 0$ slice of the wormhole and boundary time is essentially (2.22), i.e., $x = g_0 s$. Equating (2.24) and (2.25) yields $g_0 = \ell_\Delta E_{\mathrm{gap}}$. Using this in

---

[13] Notice that in our case this cosmology is not closed; it is connected to the exterior regions of the black hole. The metric (2.21) represents a patch of spacetime. However, the way the cosmology arises from the analytic continuation of a traversable wormhole is completely analogous to e.g. [104–106].

[14] We recall that $\ell_\Delta$ is not related to the scaling dimension of the bulk field via the standard AdS/CFT dictionary. The Hamiltonian here is $H_{\mathrm{eff},1}$ and to get $\ell_\Delta$ or $E_{\mathrm{gap}}$ one needs to quantize the field in the wormhole geometry.

(2.22) gives

$$\frac{\ell(t)}{\ell_\Delta} = E_{\text{gap}}(t - t_\star).$$ (2.26)

This rather general argument reproduces the linear growth (1.4) and shows that the average geometry of the ensemble of ER caterpillars is that of a cylindrical wormhole of length proportional to the circuit time $t$ used to prepare the states.

## 2.3  Near-extremal caterpillars

In the case of near-extremal black holes we can describe the average geometry of the ER caterpillars more explicitly, relating the preparation of the ER caterpillars to the preparation of the eternal traversable wormhole of Maldacena and Qi [61].

In this case we think microscopically of the ensemble of states $\mathcal{E}_\Psi^t$ of the form (2.8) as low-energy states in the Hilbert space of two SYK models,

$$|\Psi(t)\rangle \in \mathcal{H}_{\text{L}}^{\text{SYK}} \otimes \mathcal{H}_{\text{R}}^{\text{SYK}}.$$ (2.27)

Each SYK model is a system of $N$ Majorana fermions $\{\psi_i : i = 1, ..., N\}$, satisfying $\{\psi_i, \psi_j\} = 2\delta_{ij}$, with a disordered Hamiltonian of the form [4, 5]

$$H_{\text{SYK}} = \mathrm{i}^{\frac{q}{2}} \sum_{i_1 < ... < i_q} J_{i_1...i_q} \psi_{i_1}...\psi_{i_q}, \qquad \mathbb{E}[J_{i_1...i_q}^2] = \frac{\mathcal{J}^2 N}{2q^2 \binom{N}{q}}.$$ (2.28)

The drive operators $\{\mathcal{O}_\alpha\}$ defining the random circuit $U(\Gamma_t)$ in (2.5) are chosen to be Majorana strings of fixed even length $p$ in the large-$N$ limit of the model, with Brownian couplings

$$\mathcal{O}_\alpha = \mathrm{i}^{\frac{p}{2}} \psi_{i_1}...\psi_{i_p}, \qquad \mathbb{E}[g_\alpha(s)g_{\alpha'}(s')] = \frac{JN}{\binom{N}{p}}\delta(s - s')\delta_{\alpha\alpha'}.$$ (2.29)

where in this case $\alpha = (i_1, ..., i_p)$ with $i_1 < ... < i_p$. Notice that the number of perturbations is $K = \binom{N}{p}$ and in this case the normalization of the couplings in (2.29) is different from (1.3) and it ensures that the perturbation is extensive in $N$. The effective Hamiltonian (2.14) in this case is the Maldacena-Qi Hamiltonian [61]

$$H_{\text{eff},1} = H_{\text{SYK}}^1 + H_{\text{SYK}}^{\bar{1}} + JN - \frac{JN}{K}\sum_{\alpha=1}^K \mathcal{O}_\alpha^1 \mathcal{O}_\alpha^{\bar{1}\,*},$$ (2.30)

where we have used that the Majorana strings square to the identity. We will expand on aspects of $H_{\text{eff},1}$ beyond the low-temperature limit in appendix D.

**Bulk effective action.** We now briefly review the construction of the bulk effective action associated to (2.30) derived in [61]. To do that, we start from the appropriate limit where we focus on the near-horizon region of the black hole, of nearly AdS$_2 \times Y$ geometry with $Y$ compact, whose low-energy dynamics is effectively described by Jackiw-Teitelboim (JT) gravity

with matter (we set $\ell_{\text{AdS}} = 1$)

$$I_{\text{JT}}[M] = -4\pi\Phi_0\chi(M) - \int_M \mathrm{d}^2x\sqrt{g}\,\Phi(R+2) - 2\int_{\partial M}\mathrm{d}x\sqrt{\gamma}\,\Phi(K-1)\,, \qquad (2.31)$$

$$I[M] = I_{\text{JT}}[M] + I_{\text{matter}}[M]\,. \qquad (2.32)$$

Here $\chi(M)$ is the Euler characteristic of $M$, $4\pi\Phi_0$ plays the role of the extremal entropy, $g_{\mu\nu}$ is the metric on $M$ and $\gamma_{\mu\nu}$ is the induced metric on $\partial M$. In $I_{\text{matter}}[M]$ the matter is minimally coupled to the two-dimensional metric $g_{\mu\nu}$. The nearly-AdS$_2$ boundary conditions that define the theory are

$$\mathrm{d}s^2\big|_{\partial M} = \frac{\mathrm{d}u^2}{\epsilon^2}\,, \qquad \Phi\big|_{\partial M} = \frac{\phi_b}{\epsilon}\,, \qquad \epsilon \to 0\,. \qquad (2.33)$$

In this section we will work at the level of the disk topology. The metric of $\mathbf{H}^2$ in global and black hole coordinates is

$$\text{global:} \quad \mathrm{d}s^2 = \frac{\mathrm{d}\tau^2 + \mathrm{d}\sigma^2}{\sin^2\sigma}\,, \qquad \text{black hole:} \quad \mathrm{d}s^2 = (r^2 - r_h^2)\,\mathrm{d}\tau_S^2 + \frac{\mathrm{d}r^2}{r^2 - r_h^2}\,, \qquad (2.34)$$

where $\tau_S \sim \tau_S + \frac{2\pi}{r_h}$. In global coordinates, the conformal boundary is located at $\sigma = 0, \pi$ (and $\tau = \pm\infty$). These coordinates can be related to the Poincaré upper half-plane model complex coordinate $z$ (with $\text{Im}(z) \geqslant 0$), for which the metric reads $\mathrm{d}s^2 = \frac{\mathrm{d}z\mathrm{d}\bar{z}}{\text{Im}(z)^2}$, via

$$\text{global:} \quad z = \tanh\left(\frac{\tau + \mathrm{i}\sigma}{2}\right)\,, \qquad \text{black hole:} \quad z = \tan\left(\frac{r_h\tau_S}{2} + \frac{\mathrm{i}}{2}\text{arccoth}\left(\frac{r}{r_h}\right)\right)\,. \qquad (2.35)$$

It is convenient to use global coordinates to describe the forward and backward contours coupled by the effective Hamiltonian (2.30). The bulk system that we have is that of two boundary Schwarzian modes, with Euclidean action [107–110]

$$I = -\phi_b\int\mathrm{d}u\left(\left\{\tanh\left(\frac{\tau_1(u)}{2}\right), u\right\} + \left\{\tanh\left(\frac{\tau_{\bar{1}}(u)}{2}\right), u\right\}\right) + I_{\text{int}}\,, \qquad (2.36)$$

subject to the effective bi-local interaction in (2.30) of an operator of conformal dimension $\Delta$ [61]

$$I_{\text{int}} = -N\int\mathrm{d}u\,J(u)\frac{c_\Delta}{(2\mathcal{J}\ell_{\text{AdS}})^{2\Delta}}\left(\frac{\tau_1'(u)\tau_{\bar{1}}'(u)}{\cosh^2\left(\frac{\tau_1(u)-\tau_{\bar{1}}(u)}{2}\right)}\right)^\Delta + N\int\mathrm{d}u\,J(u)\,, \qquad (2.37)$$

where $\tau_i'(u) = \frac{\mathrm{d}\tau_i}{\mathrm{d}u}$ and the constant $c_\Delta$ can be found in [61]. The first term in (2.37) is the two-point correlator of $\mathcal{O}_\alpha$ of conformal dimension $\Delta$, $\langle\mathcal{O}_\alpha^1(\tau_1)\mathcal{O}_\alpha^{\bar{1}}(\tau_{\bar{1}})\rangle$, coupled to the gravitating boundaries via a reparametrization of the global time. We have restored the factors of $\ell_{\text{AdS}}$ to make it explicit that $J$ has dimensions of energy. We write the time-dependence of the coupling $J(u)$ explicitly to make it manifest that the interaction in the quantities that we will compute is only turned on in parts of the contour. The second term in (2.37) is the constant energy term in (2.30). The dimensionful coupling of the Schwarzian is related to the SYK fermion couplings

via

$$\phi_b = \frac{N\alpha_S}{\mathcal{J}},\qquad\qquad(2.38)$$

where $\alpha_S$ is a $q$-dependent coefficient which can be found in [5]. Moreover, it is convenient to define the dimensionless time

$$\tilde{u} = N\frac{u}{\phi_b} = \frac{u\mathcal{J}}{\alpha_S},\qquad\qquad(2.39)$$

corresponding to the SYK time in units of the fermion couplings $\mathcal{J}$. We will denote $\dot{\tau}_i(\tilde{u}) = \frac{\mathrm{d}\tau_i}{\mathrm{d}\tilde{u}}$.

In the form (2.36) the action still requires the $\mathsf{PSL}(2,\mathbf{R})$ gauge constraints, i.e. the isometries of $\mathbf{H}^2$, to be imposed on physical configurations [31, 61]. Partially, this is done by demanding that the global time of the boundaries be the same $\tau_1(u) = \tau_{\bar{1}}(u) = \tau(u)$. There is an additional $\mathsf{PSL}(2,\mathbf{R})$ gauge charge which still must be set to zero. After doing this, the resulting theory, expressed in terms of the dimensionless "renormalized length" variable[15]

$$\ell \equiv -2\log\left(\frac{\dot{\tau}(\tilde{u})}{\ell_{\mathrm{AdS}}}\right),\qquad\qquad(2.40)$$

is the non-relativistic particle mechanics [61, 110, 111]

$$I = \frac{N}{2}\int \mathrm{d}\tilde{u}\left(\frac{1}{2}\dot{\ell}^2 + 2e^{-\ell} - \eta(\tilde{u})e^{-\Delta\ell}\right) + \frac{N}{2\tilde{c}_\Delta}\int \mathrm{d}\tilde{u}\,\eta(\tilde{u}) + I_{\mathrm{reg}},\qquad\qquad(2.41)$$

where we have introduced the dimensionless coupling[16]

$$\eta = \frac{J}{\mathcal{J}}\frac{c_\Delta}{(2\alpha_S)^{2\Delta-1}}.\qquad\qquad(2.42)$$

In (2.41) $I_{\mathrm{reg}}$ is a regularization of the Euclidean action that makes $I$ finite for on-shell solutions that reach $\ell \to -\infty$. In (2.41) we have introduced the constant $\tilde{c}_\Delta = 2c_\Delta/(2\alpha_S)^{2\Delta}$.

We will study the system in the parameter regime

$$\frac{1}{N} \ll \eta \ll 1,\qquad N \to \infty.\qquad\qquad(2.43)$$

The upper bound of (2.43) is required for the consistency of the approximation leading to (2.41) from (2.30), given that the bi-local interaction has been replaced by its expectation value. The lower bound in (2.43) will guarantee that the interaction produces substantial gravitational backreaction at the classical level. This leads to a ground state of the effective Hamiltonian (2.30) with a semiclassical dual described by a connected wormhole geometry.

**Euclidean bounce solution.** We will now evaluate the average norm $Z(t)$ of the ensemble of ER caterpillars (2.15) in the JT theory. The corresponding matrix element can be expressed

---

[15] The dynamical boundaries in global coordinates sit at $\sigma(u) = \epsilon\tau'(u)$ and $\sigma(u) = \pi - \epsilon\tau'(u)$ and the geodesic length between them is $-2\ell_{\mathrm{AdS}}\log\epsilon\tau'(u)$ as $\epsilon \to 0$. For a particular renormalization the variable $\ell$ in (2.40) is the finite part of the length of the geodesic in units of $\ell_{\mathrm{AdS}}$.

[16] Our definitions of $\ell$ in (2.40) and of $\eta$ in (2.42) are related to the ones in [61] by a factor of 2 (and a minus sign for $\ell$).

as

$$Z(t) = Z(\beta)^{-1} \langle I | e^{-\frac{\beta}{4}(H_0^1 + H_0^{\bar{1}})} e^{-t H_{\text{eff},1}} e^{-\frac{\beta}{4}(H_0^1 + H_0^{\bar{1}})} | I \rangle \,, \tag{2.44}$$

where $|I\rangle$ is the infinite temperature TFD of the two contours. Semiclassically, (2.44) is computed in a WKB approximation by the on-shell action of an Euclidean bounce solution of the $\ell$-particle,

$$\langle I | e^{-\frac{\beta}{4}(H_0^1 + H_0^{\bar{1}})} e^{-t H_{\text{eff},1}} e^{-\frac{\beta}{4}(H_0^1 + H_0^{\bar{1}})} | I \rangle \sim e^{-I} \,. \tag{2.45}$$

The bra and ket states in (2.45) enforce $\ell = -\infty$ boundary conditions in the Euclidean past and future. The bounce solution is represented in Fig. 10.

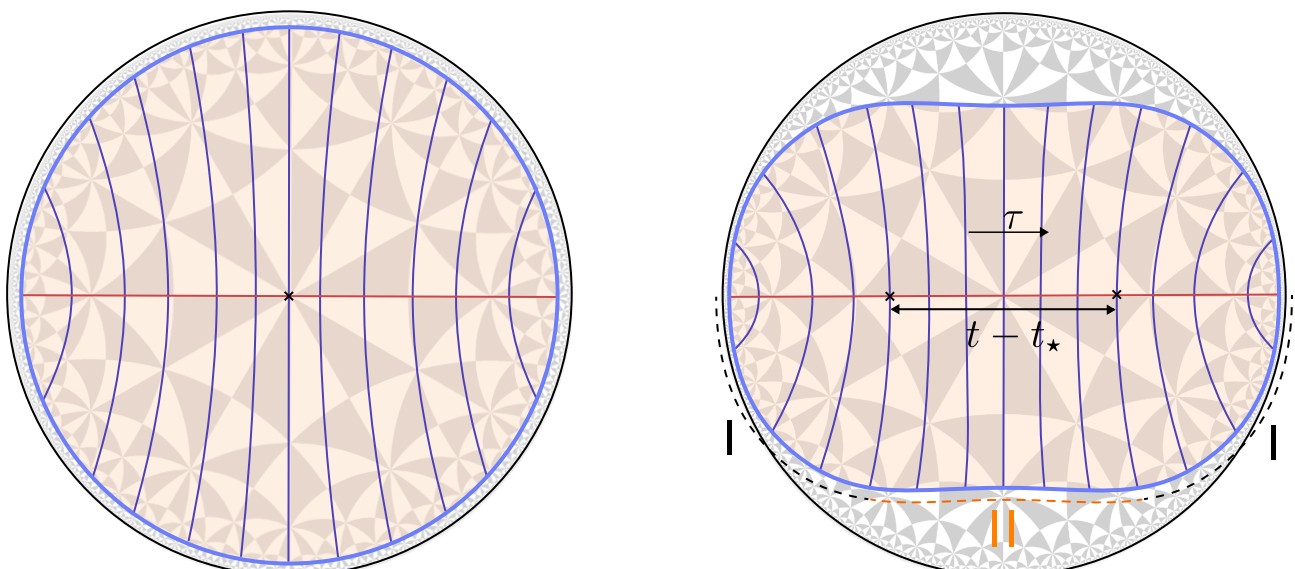

**Figure 7:** Euclidean bounce solutions of the dynamical boundary (blue curve) in $\mathbf{H}^2$. The $\ell$ variable is the renormalized length of the geodesics in dark blue. On the left, the Euclidean black hole solution for $\eta = 0$. In this case the Euclidean bounce computes the norm of the Hartle-Hawking state in the WKB approximation. On the right, the double-bounce contribution to $Z(t)$. The effective double-trace interaction $\eta \neq 0$ makes the boundary particles bend inwards and the length of the ER bridge (red slice) eventually grows linearly with the preparation time of the circuit. The crosses represent minima of the dilaton, where the apparent horizons are located.

In order to work with dimensionless quantities, it will be useful to define

$$\tilde{\beta} = \frac{\beta N}{\phi_b} \,, \qquad \tilde{t} = \frac{t N}{\phi_b} \,. \tag{2.46}$$

We shall pick a reflection-symmetric time coordinate $\tilde{u} \in [-\frac{\tilde{\beta}}{4} - \frac{\tilde{t}}{2}, \frac{\tilde{\beta}}{4} + \frac{\tilde{t}}{2}]$ for the $\ell$-particle. The time-dependent interaction is

$$\eta(\tilde{u}) = \begin{cases} 0 & \text{for } \dfrac{\tilde{t}}{2} < |\tilde{u}| < \dfrac{\tilde{t}}{2} + \dfrac{\tilde{\beta}}{4} \quad (\text{regime I}) \,, \\[2ex] \eta & \text{for } |\tilde{u}| < \dfrac{\tilde{t}}{2} \quad (\text{regime II}) \,. \end{cases} \tag{2.47}$$

On each of these two regimes, the total energy of the $\ell$-particle is respectively

$$-E = \frac{1}{2}\dot{\ell}^2 + V(\ell)\,, \qquad V(\ell) = -2e^{-\ell} + \eta e^{-\Delta\ell}\,, \tag{2.48}$$

where $\eta$ is set to zero in regime I. The minus sign in front of $E$ in (2.48) is chosen for consistency with the Lorentzian convention of $E$ to be bounded from below. As illustrated in Fig. 8, when the interaction is turned on, the Euclidean potential $V(\ell)$ has a maximum at $\ell_{\max} = \frac{1}{\Delta-1}\log(\eta\Delta/2)$, provided that $0 < \Delta < 1$. This will always be possible if we choose $p/q < 1$ for the perturbations in (2.29). The value of the maximum is $V(\ell_{\max}) = 2\frac{(1-\Delta)}{\Delta}e^{-\ell_{\max}}$. For concreteness, below we will consider $\Delta = \frac{1}{2}$, i.e. a free fermion $\text{CFT}_2$, which leads to $\ell_{\max} = -2\log(\eta/4)$ and $V(\ell_{\max}) = \eta^2/8$. The minimum energy the particle can have is in this case $E_{\min} = -\eta^2/8$.

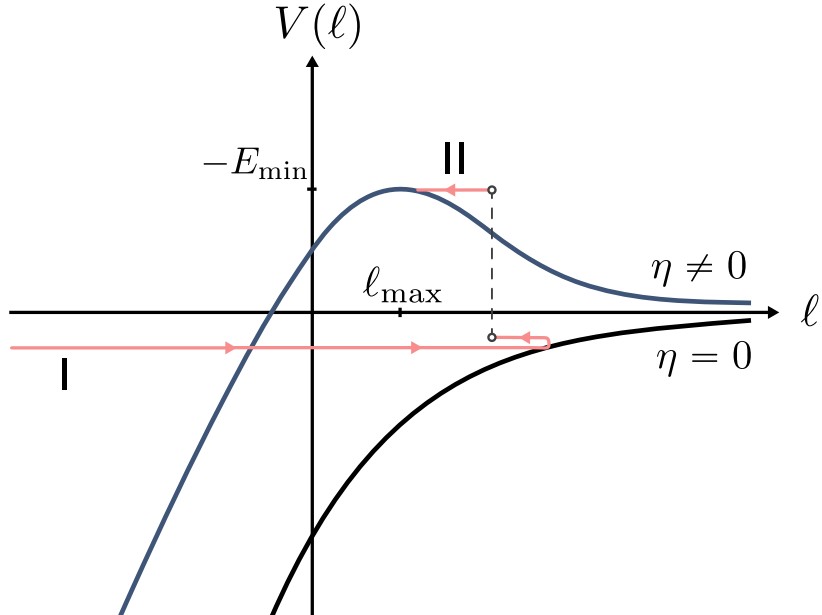

**Figure 8:** Euclidean potential of the $\ell$-particle. The Euclidean bounce solution is shown in red. The particle starts at $\ell = -\infty$ and bounces back in region I. The particle enters the interacting region II ($\eta \neq 0$) and its conserved energy changes. The Euclidean time that it takes to bounce close to the maximum of the interacting potential in region II becomes arbitrarily large since the maximum is an equilibrium point of the system. After bouncing the particle travels in the opposite direction along the trajectory until it reaches $\ell = -\infty$ again.

In regime I, for $\tilde{u} > 0$, the trajectory of the particle is[17]

$$\ell_I(\tilde{u}) = 2\log\cos\left((\tilde{u} - \tilde{u}_0)\sqrt{\frac{E}{2}}\right) - \log\frac{E}{2}\,. \tag{2.49}$$

---

[17] This trajectory describes the Euclidean black hole solution. For the two-sided black hole, the total energy above extremality $2M$ is fixed demanding that the Euclidean time to reach $\ell = -\infty$ is $\beta/4$. This gives $2M = N^2 E/(2\phi_b) = 4\pi^2\phi_b/\beta^2$, where $N^2 E/(2\phi_b)$ is the extensive energy measured with respect to the dimensionful time $u$. The regularized free energy from (2.41) is $F = I/\beta = -2\pi^2\phi_b/\beta^2$ and the entropy is $S = \beta(M-F) = 4\pi^2\phi_b/\beta$. In this case the length of the throat is $\ell = 2\log N - 2\log(2\pi\phi_b/\beta)$. For large enough $\beta\mathcal{J} \sim N$ the classical solution cannot be trusted. In this regime the one-loop fluctuations become important.

Here $\tilde{u}_0$ is an integration constant that will be fixed below.

When the interaction is turned on, $\eta \neq 0$, in regime II, the trajectory becomes

$$\ell_{\text{II}}^{\rightharpoonup}(\tilde{u}) = 2 \log \left( \frac{\sqrt{\eta^2 + 8E'}}{2E'} \cos \left( \tilde{u} \sqrt{\frac{E'}{2}} \right) - \frac{\eta}{2E'} \right) . \tag{2.50}$$

In this case, the integration constant has been fixed so that $\ell_{\text{II}}^{\rightharpoonup}(0) = -2 \log \left( \eta + \sqrt{\eta + 8E'} \right) + 4 \log 2 \leqslant \ell_{\text{max}}$ is the turning point. Notice that the energy $E'$ in this part of the trajectory is in general different from $E$.

For $E' < 0$ there exists another relevant solution

$$\ell_{\text{II}}(\tilde{u}) = 2 \log \left( -\frac{\sqrt{\eta^2 + 8E'}}{2E'} \cos \left( \tilde{u} \sqrt{\frac{E'}{2}} \right) - \frac{\eta}{2E'} \right) . \tag{2.51}$$

In this case, the particle bounces on the right side of the potential, at the length $\ell_{\text{II}}(0) = -2 \log \left( \eta - \sqrt{\eta^2 + 8E'} \right) + 4 \log 2 \geqslant \ell_{\text{max}}$, and starts and ends at $\ell = +\infty$ (the time to reach $\ell = \infty$ is infinite).

In order to match both trajectories, the length and its derivative must be continuous across $\tilde{u} = \frac{\tilde{t}}{2}$,

$$\ell_{\text{I}} \left( \frac{\tilde{t}}{2} \right) = \ell_{\text{II}} \left( \frac{\tilde{t}}{2} \right) , \qquad \dot{\ell}_{\text{I}} \left( \frac{\tilde{t}}{2} \right) = \dot{\ell}_{\text{II}} \left( \frac{\tilde{t}}{2} \right) . \tag{2.52}$$

These two conditions will relate $E, E'$ and $\tilde{u}_0$ and leave a single free parameter. The value of $\tilde{u}_0$ will be determined from the boundary condition imposed by the infinite TFD state,

$$\ell_{\text{I}} \left( \frac{\tilde{t}}{2} + \frac{\tilde{\beta}}{4} \right) = -\infty \implies u_0 = \frac{\tilde{t}}{2} + \frac{\tilde{\beta}}{4} - \frac{\pi}{2} \sqrt{\frac{2}{E}} . \tag{2.53}$$

The jump in the energy can be determined from the conditions (2.52) and it yields

$$E' = E - \frac{\eta \sqrt{\frac{E}{2}}}{\sin \left( \frac{\tilde{\beta}}{4} \sqrt{\frac{E}{2}} \right)} . \tag{2.54}$$

Together with a constraint from (2.52) this fixes the energy via

$$\cos \left( \frac{\tilde{\beta}}{4} \sqrt{\frac{E}{2}} \right) = \pm \frac{\sqrt{\eta^2 + 8E'}}{\sqrt{8E'}} \sin \left( \frac{\tilde{t}}{2} \sqrt{\frac{E'}{2}} \right) , \tag{2.55}$$

where $+$ corresponds to (2.50) and $-$ to (2.51).

The solution with a $+$ in (2.55) corresponds to a single bounce solution, where the particle starts at $\ell = -\infty$ and only bounces on the potential of region II. We represent this solution in Fig. 9.

When $\tilde{\beta} > \tilde{\beta}_c$, the single bounce solution will not exist. The reason is that $\dot{\ell}_{\text{I}}(\tilde{t}/2) \leqslant 0$ for

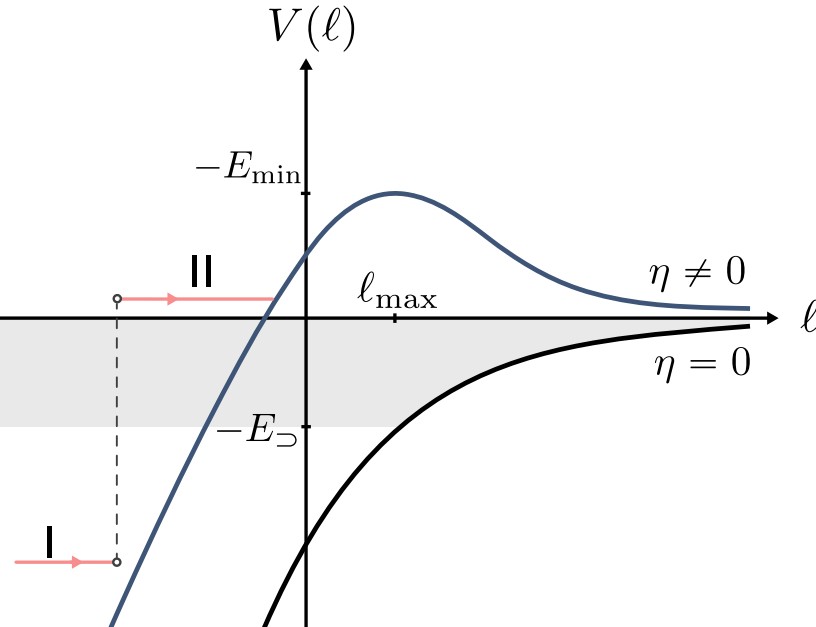

**Figure 9:** The single bounce solution in red. The particle starts at $\ell = -\infty$ and only bounces back in region II. The solution does not exist when $\tilde{\beta} \geqslant \tilde{\beta}_c$ because the energy in region I would need to be on the gray region, which produces a jump to region II exceeding the maximum of the pontential.

this solution, and this imposes $0 < E \leqslant \frac{8\pi^2}{\tilde{\beta}^2}$. Using (2.54), the increase in energy can at most be

$$E - E' \leqslant \frac{2\pi\eta}{\tilde{\beta}} \,. \tag{2.56}$$

However, the energy jump $E - E'$ is monotonically decreasing with $\ell$ for $\ell < \ell_{\max}$. The minimum value is attained for $\ell_{\max}$, for which the rest energy in region I is $E_\supset \equiv 2\exp(-\ell_{\max}) = \eta^2/8$ (see Fig. 9) and the energy jump is

$$E - E' \geqslant E_\supset - E_{\min} = \frac{\eta^2}{4} \,. \tag{2.57}$$

Therefore, the single bounce solution will not exist for $\tilde{\beta} > \tilde{\beta}_c$ where the critical inverse temperature is

$$\tilde{\beta}_c \equiv \frac{8\pi}{\eta} \,. \tag{2.58}$$

We will tune our Euclidean preparation parameter $\tilde{\beta} = \tilde{\beta}_c + 0^+$ slightly above $\tilde{\beta}_c$ so that the solution that we are interested in is the double bounce solution of Fig. 8, with a very small but long-lived bounce in region II. From (2.43) $1 \ll \tilde{\beta}_c \ll N$, so this regime is under control.

**Linear wormhole growth.** In regime I, for $\tilde{u} > 0$, using the definition of the renormalized geodesic length (2.40), the solution (2.49) fixes

$$\tanh\left(\frac{\tau - \tau_0}{2\ell_{\mathrm{AdS}}}\right) = \tan\left(\frac{\tilde{u} - \tilde{u}_0}{2}\sqrt{\frac{E}{2}}\right) \,, \tag{2.59}$$

which, up to rescaling of $\tilde{u}$, it is the relation between the global and black hole times.

In regime II, adding the interaction and using (2.51) gives

$$\tanh\left(\frac{\tau}{2\ell_{\text{AdS}}}\right) = \tan\left(\frac{\tilde{u}}{2}\sqrt{\frac{E'}{2}}\right)\tan\left(\frac{\varphi_{E'}}{2}\right), \quad \cos\varphi_{E'} = \frac{\eta}{\sqrt{\eta^2 + 8E'}}. \tag{2.60}$$

The proper length of the ER caterpillars corresponds to the value of the $\tau$ coordinate, given that the relevant bulk slice where $Z(t)$ is interpreted as a norm corresponds to the reflection-symmetric slice of the disk, $\sigma = \frac{\pi}{2}$ in global coordinates (red line in Fig. 10). We will use the subscript $\Sigma$ to avoid confusion. Using (2.59) and (2.60) the length of this slice is[18]

$$\ell_{\Sigma}(t) = 4\ell_{\text{AdS}}\left(\operatorname{arctanh}\left(\tan\left(\frac{\tilde{t}}{4}\sqrt{\frac{E'}{2}}\right)\tan\left(\frac{\varphi_{E'}}{2}\right)\right) + \operatorname{arctanh}\left(\tan\left(\frac{\pi}{4} - \frac{\epsilon}{8}\sqrt{\frac{E}{2}}\right)\right)\right), \tag{2.61}$$

where, as in the case of the black hole solution, the second term produces logarithmic divergences as $\epsilon \to 0$, corresponding to the asymptotic regions of the slice, which must be renormalized.

For times $\tilde{t} > \tilde{t}_\star$, Eqs. (2.54) and (2.55) will fix $E' = E_{\min}$ and $E = E_\supset$. We can define the length of the ER caterpillars as $\ell(t)_{\text{ER}} \equiv \ell(t) - \ell_{\text{out}}$, where $\ell_{\text{out}} = -\log 4\pi^2/\tilde{\beta}_c^2$ is the length of the two-sided throat for a near-extremal black hole at inverse temperature $\beta_c = \phi_b\tilde{\beta}_c/N$. Using (2.61) gives

$$\ell_{\text{ER}}(t) = \frac{\eta\ell_{\text{AdS}}}{4}\tilde{t} = g_0 t, \qquad g_0 \equiv \frac{J\ell_{\text{AdS}}Kc_{1/2}}{4N\alpha_S}, \tag{2.62}$$

This reproduces the linear growth (2.22). Obtaining the onset time $t_\star$ from this classical analysis requires solving for $E(\tilde{t})$ and $E'(\tilde{t})$ from (2.54) and (2.55) for arbitrary values of $\tilde{t}$, and plugging them in (2.61) and we shall not do that here.

For a general perturbation $0 < \Delta < 1$, following similar considerations, the linear growth can be computed from the constant rate $-2\log(\dot{\tau}_0/\ell_{\text{AdS}}) \equiv \ell_{\max}$ that the solution attains at the maximum of the potential. In this case one must tune $\tilde{\beta} = \tilde{\beta}_c = 2\pi(\eta\Delta/2)^{\frac{1}{2(\Delta-1)}}$ instead. This leads to a solution which for $t \gg t_\star$ satisfies

$$\ell_{\text{ER}}(t) = \tau_0't, \qquad \tau_0' \equiv \frac{\dot{\tau}_0\mathcal{J}}{\alpha_S} = \frac{\ell_{\text{AdS}}\mathcal{J}}{\alpha_S}\left(\frac{\eta\Delta}{2}\right)^{\frac{1}{2(1-\Delta)}}. \tag{2.63}$$

The gap of the effective Hamiltonian is given by the first excited state of the matter field of smallest conformal dimension $\Delta_0$ on the wormhole,

$$E_{\text{gap}} = \Delta_0\tau_0'\ell_{\text{AdS}}^{-1} = \frac{\Delta_0\mathcal{J}}{\alpha_S}\left(\frac{\eta\Delta}{2}\right)^{\frac{1}{2(1-\Delta)}}. \tag{2.64}$$

Note that $N^{-2(1-\Delta)} \ll E_{\text{gap}}\ell_{\text{AdS}} \ll 1$ from (2.43). The gap of the gravitational degree of freedom is larger than this [61]. The onset time can be estimated as $t_\star = E_{\text{gap}}^{-1}\log N$ from the number $N$

---

[18] We remark that in (2.61) $\ell_{\Sigma}(t)$ corresponds to the length of the Cauchy slice $\Sigma$ associated to the states in $\mathcal{E}_\Psi^t$, and not to the renormalized geodesic length between both Euclidean boundaries.

of first excited states on the wormhole. Therefore, inserting (2.64) in (2.63) implies

$$\frac{\ell_{\text{ER}}(t)}{\ell_{\Delta_0}} = E_{\text{gap}}(t - t_\star)\,, \tag{2.65}$$

where the matter correlation scale is in this case

$$\ell_{\Delta_0} = \frac{\ell_{\text{AdS}}}{\Delta_0}\,. \tag{2.66}$$

This yields the linear growth (1.4) and (2.26) for the family of near-extremal ER caterpillars. Note that here we have used $\Delta_0$ to refer to the conformal dimension of the lightest matter field in the bulk, which in general is different from $\Delta$, the conformal dimension of the perturbations in the random circuit. In the case of the SYK caterpillars, $\Delta_0 = 1/q$ associated to single fermion excitations in the bulk, and there are $N$ such excitations.

**Profile of the dilaton.** We are also interested in the profile of the dilaton, to see that the long wormhole sits inside of the black hole. From (2.31), the dilaton satisfies the equation of motion

$$\nabla_\mu \nabla_\nu \Phi - g_{\mu\nu} \nabla^2 \Phi + \Phi g_{\mu\nu} = -8\pi G \langle T_{\mu\nu} \rangle\,. \tag{2.67}$$

It is convenient to use coordinates $x_\pm = \tau \pm \sigma$, in terms of which the equations reduce to

$$-\partial_\pm \left( \sin^2 \sigma\, \partial_\pm \Phi \right) = 8\pi G \sin^2 \sigma \langle T_{\pm\pm} \rangle\,, \tag{2.68}$$

$$2 \sin^2 \sigma \partial_+ \partial_- \Phi - \Phi = 16\pi G \sin^2 \sigma \langle T_{+-} \rangle\,. \tag{2.69}$$

The negative energy $\langle T_{\pm\pm} \rangle$ required to support the wormhole comes from the bi-local effective interactions. This interaction can be interpreted as a non-standard boundary condition between both boundaries. Note that, from $\mathsf{PSL}(2,\mathbf{R})$ invariance, the null energy of any $\text{CFT}_2$ in global $\text{AdS}_2$ with standard CPT-symmetric boundary conditions between both boundaries vanishes. The Casimir null energy on the flat strip cancels the conformal anomaly in $\text{AdS}_2$ [61].

We will work in an adiabatic approximation, where the effect of the moving boundaries can be neglected at leading order. In order to be explicit, as above, we will assume that we have $N$ free fermions in the bulk, with $\Delta = 1/2$. The stress-energy of the free fermion in global $\text{AdS}_2$ with non-trivial boundary conditions associated to the bi-local coupling can be found in Appendix C of [61]

$$\langle T_{++} \rangle = \langle T_{--} \rangle = -\frac{\eta N}{L}(1 - 4\eta)\,, \qquad \langle T_{+-} \rangle = 0\,, \tag{2.70}$$

where we are leaving explicit that $L = \pi$ for the case of the static boundaries in global $\text{AdS}_2$.

In the adiabatic approximation, the boundaries can be considered to be moving very slowly $L = L(\tau)$, and neglect the effects of the velocity. Solving for the dilaton in (2.68) and (2.69) gives,

$$\Phi(\sigma, \tau) \approx \frac{8\pi G \eta N (1 - 4\eta)}{L(\tau)} \left[ \frac{\frac{\pi}{2} - \sigma}{\tan \sigma} + 1 \right] + \Phi_0\,, \tag{2.71}$$

where we have absorbed the additive constant of the solution into $\Phi_0$.

The areas of the transverse spheres relevant for the average geometry of the ER caterpillars are determined by $\Phi(\frac{\pi}{2}, \tau)$. From the solution for the boundary particle described in this section, the length $L(\tau)$ varies very slowly in region II, while it attains its minimum at the center of the wormhole. Thus, the sphere at the center of the wormhole is a maximal surface, as the dilaton $\Phi(\frac{\pi}{2}, \tau)$ decreases to second order in both directions of the $\tau$ coordinate. The minima of $\Phi(\frac{\pi}{2}, \tau)$ are located in the regions of the $\tau$ coordinate where (2.71) stops being valid, and where one must connect the solution to the eternal black hole solution, as shown in Fig. 10. This shows that the linearly growing ER caterpillars lie inside of the two-sided black hole, as expected from the general considerations of section 2.2.

## 2.4 Other properties

We will now discuss other properties of the ensemble $\mathcal{E}_\Psi^t$ of ER caterpillars (2.8). Consider the ground state of the effective Hamiltonian $H_{\text{eff},1}$ (2.14), which in the energy basis has wavefunction[19]

$$|\text{GS}\rangle = \sum_i c_i \, |E_i^*\rangle \otimes |E_i\rangle \, , \qquad \rho(\text{GS}) = \sum_i c_i \, |E_i\rangle\langle E_i| \, . \tag{2.72}$$

As explained in appendix C, if $H_0$ is a strongly mixing Hamiltonian that satisfies ETH, for a large number $K$ of simple perturbations, the ground state will be diagonal in the energy basis of $H_0$ [99]. Moreover, the wavefunction $c_i$ has real Gaussian form

$$c_i^2 = \frac{1}{\rho(E_i)} \exp\left(-\frac{(E_i - \overline{E})^2}{2\sigma_E^2}\right) \, , \tag{2.73}$$

peaked at $E_i = \overline{E}$ and with an extensive energy variance $\sigma_E^2$ in the thermodynamic limit. In (2.73) the function $\rho(E_i)$ is the thermodynamic density of states of the system at energy $E_i$, suitably normalized for $\sum_i c_i^2 = 1$.

**Value of the parameter $\beta$.** The parameter $\beta$ in the wavefunction (2.8) is chosen to maximize the overlap between the TFD at inverse temperature $\beta$ and the ground state $|\text{GS}\rangle$ of the effective Hamiltonian

$$\beta \equiv \max_{\hat{\beta}}\{\langle\text{TFD}(\hat{\beta})|\text{GS}\rangle\} \, . \tag{2.74}$$

Under the assumptions outlined in section 2.1, and as explained in appendix C, we expect that the ground state generally has a substantial overlap with the TFD in the large-$N$ limit of the holographic CFT[20]

$$\langle\text{TFD}|\text{GS}\rangle \approx \sqrt{\frac{2\sigma_E\sigma_{\text{TFD}}}{\sigma_E^2 + \sigma_{\text{TFD}}^2}} = O(N^0) \qquad N \to \infty \, , \tag{2.75}$$

---

[19] The state $\rho(\text{GS})$ is the operator that corresponds to $|\text{GS}\rangle$ by the state/operator correspondence. It should not be confused with the reduced state in the ground state.

[20] This applies to any thermodynamic limit.

where $\sigma^2_{\text{TFD}}$ is the energy variance in the TFD state, which controls the canonical heat capacity. For (2.75) to be true, the ground state wavefunction (2.73) must be peaked at $\overline{E} = E_\beta$, at the average energy $E_\beta$ of the canonical Gibbs state $\rho_\beta$.

In the case of the near-extremal ER caterpillars studied in section 2.3, the semiclassical analysis fixes $\beta \mathcal{J} = 2\pi\alpha_s(\eta\Delta/2)^{\frac{1}{2(\Delta-1)}}$. The overlap of the ground state with the TFD is close to 1 for $\Delta < 1/2$, corresponding to relevant perturbations, and sufficiently small $\eta$, and it is exactly 1 for any $\eta$ in the large-$q$ limit of the model [61].

**Effective equilibrium states.** For $t \gg t_\star$, the ensemble $\mathcal{E}^t_\Psi$ corresponds to effective equilibrium states of the two-sided black hole. To see this, we consider the reduced density matrix

$$\rho_{\mathsf{R}} = \text{Tr}_{\mathsf{L}}(|\Psi(t)\rangle\langle\Psi(t)|) = \frac{1}{Z(\beta)Z_\Psi(t)} \sum_{i,j,k} e^{-\frac{\beta}{4}(E_i+E_j+2E_k)} U(\Gamma_t)_{ki} U(\Gamma_t)^*_{kj} |E_i\rangle\langle E_j| \,. \quad (2.76)$$

Using the identity (2.13) we can take the average of $\rho_{\mathsf{R}}$ over the ensemble of states,[21] $\mathbb{E}_{\mathcal{E}^t_\Psi}[\rho_{\mathsf{R}}] = \rho_{\text{eq}}$, where the equilibrium state corresponds to

$$\rho_{\text{eq}} = \frac{1}{\mathcal{N}} e^{-\frac{\beta}{4}H_0} \rho(\text{GS}) e^{-\frac{\beta}{4}H_0} \,. \quad (2.77)$$

The equilibrium state is normalized by

$$\mathcal{N} \equiv Z(\beta)^{1/2} \langle\text{TFD}|\text{GS}\rangle \,. \quad (2.78)$$

More explicitly, the equilibrium state has entries

$$\rho_{\text{eq}} = \sum_i p_i |E_i\rangle\langle E_i| \,, \qquad p_i = \frac{e^{-\frac{\beta E_i}{2}} c_i}{\mathcal{N}} \,. \quad (2.79)$$

The same considerations apply to the L subsystem.

From the large overlap (2.75), the state $\rho_{\text{eq}}$ is close to a thermal state in the thermodynamic limit. If $\langle\text{TFD}|\text{GS}\rangle = 1$ then $c_i = \exp(-\beta E_i/2)$ and the equilibrium state $\rho_{\text{eq}}$ (2.79) is a canonical Gibbs state at inverse temperature $\beta$. This is always approximately achieved for large enough values of the coupling $J$, for which $\beta(J)$ becomes small. More generally, for the wavefunction (2.73) the equilibrium ensembles will be equivalent in the large-$N$ limit

$$\text{Tr}_{\mathsf{L}}(\rho_{\text{eq}}A_{\mathsf{L}}) \sim \text{Tr}_{\mathsf{L}}(\rho_\beta A_{\mathsf{L}}) \,, \qquad N \to \infty \,, \quad (2.80)$$

for any simple operator $A_{\mathsf{L}}$ of $O(N^0)$ energy. In the bulk, this implies that the exterior geometry of the ER caterpillars is that of an equilibrium black hole at inverse temperature $\beta(J)$.

**Onset time.** The onset time $t_\star$ of the linear wormhole growth can be estimated from the lowest excitations of the wormhole. The energy of such excitations corresponds to the gap $E_{\text{gap}}$

---

[21] To get (2.77), we are taking an "annealed" average on $\mathcal{E}^t_\Psi$, where we average over the ensemble of unnormalized states and then divide by the average norm $Z(t)$. As we comment below, this is reasonable because the norm is self-averaging over the ensemble of ER caterpillars. Moreover, we replace $\exp(-tH_{\text{eff},1})$ by the projector into the ground state of $H_{\text{eff},1}$ for $t \gg t_\star$.

of the effective Hamiltonian $H_{\text{eff},1}$, while the number of first excited states is $N_*$. The onset time is essentially the timescale for these states to decay,

$$t_\star \sim E_{\text{gap}}^{-1} \log N_* \,, \tag{2.81}$$

since, assuming a quasi-particle spectrum, all the rest of the states will have decayed at this timescale. From the bulk description of the ground state, the first excited states correspond to states of light bulk fields on the wormhole. The number of first excited states $N_*$ then corresponds to the number of bulk species of light fields. For large-$N$ holographic CFTs, this number is $N_* = O(N^0)$. Therefore we find that $E_{\text{gap}} t_\star = O(N^0)$.[22]

**Variances from replica wormholes.** A natural question from the analysis of section 2.2 is whether the length of the wormhole changes substantially from instance to instance of the ensemble of states $\mathcal{E}_\Psi^t$.

The norm $Z_\Psi(t)$ for an individual state (2.8) is computed in the bulk (2.11) by a complex saddle-point geometry $M_\Psi$. The value of $Z_\Psi(t)$ depends smoothly on the geometric properties of $M_\Psi$. We shall consider the variance of the norm over different realizations of the state

$$Z_2(t) \equiv \text{Var}_{\mathcal{E}_\Psi^t} \left[ Z_\Psi(t) \right] = \mathbb{E}_{\mathcal{E}_\Psi^t} \left[ Z_\Psi(t)^2 \right] - Z(t)^2 \,. \tag{2.82}$$

The variance $Z_2(t)$ can be computed using the identity for the random circuit

$$\mathbb{E} \left[ U(\Gamma_t)^{\otimes 2} \otimes U(\Gamma_t)^{* \, \otimes 2} \right] = \exp(-t H_{\text{eff},2}) \,, \tag{2.83}$$

for the four-replica time-independent effective Hamiltonian

$$H_{\text{eff},2} = \sum_{r=1}^{2} \left( H_0^r + H_0^{\bar{r}} \right) + \frac{J}{2} \sum_{\alpha=1}^{K} \left( \sum_{r=1}^{2} \mathcal{O}_\alpha^r - \mathcal{O}_\alpha^{\bar{r} \, *} \right)^2 \,. \tag{2.84}$$

The effective Hamiltonian $H_{\text{eff},2}$ couples the 4 replicas via bi-local interactions. These interactions arise effectively from the gaussian correlations of the Brownian couplings defining the random circuit. The index $r$ ($\bar{r}$) labels the forward (backward) replica where the operator $\mathcal{O}_\alpha^r$ ($\mathcal{O}_\alpha^{\bar{r}}$) acts.

The variance arises from the survival amplitude in the four replica Hilbert space,

$$\mathbb{E}_{\mathcal{E}_\Psi^t} \left[ Z_\Psi(t)^2 \right] = \left( \langle \text{TFD}|_{1\bar{1}} \otimes \langle \text{TFD}|_{2\bar{2}} \right) \exp(-t H_{\text{eff},2}) \left( |\text{TFD}\rangle_{1\bar{1}} \otimes |\text{TFD}\rangle_{2\bar{2}} \right) \,. \tag{2.85}$$

As we explain in detail in section 3.2 and in appendix C, for sufficiently large $J$ the ground space of $H_{\text{eff},2}$ breaks replica symmetry and decomposes into the product of the ground spaces of two $H_{\text{eff},1}$ factors. In gravity, this amounts to the dominance of two-replica wormhole configurations reproducing the contributions from the ground space to various quantities. In our case of interest

---

[22] Since we expect $\beta E_{\text{gap}} = O(N^0)$, the onset time $t_\star$ is parametrically smaller in $N$ than the scrambling time $t_{\text{scr}} \sim \beta \log N$ for a single perturbation. This is to be contrasted with other systems, like the SYK model, where the onset timescale is $E_{\text{gap}} t_\star \sim \log N$, where $N$ here is the number of Majorana fermions. The reason is that for SYK the bulk contains $N$ lightest fermion excitations of the same mass. It is interesting to note that the timescale $t_\star$ behaves parametrically like the Thouless time [92, 112–114].

here, this means that for $t \gg t_\star$,

$$\exp(-tH_{\text{eff},2}) \approx \sum_{\sigma \in \text{Sym}(2)} \left( |\text{GS}\rangle_{1\sigma(\bar{1})} \otimes |\text{GS}\rangle_{2\sigma(\bar{2})} \right) \left( \langle\text{GS}|_{1\sigma(\bar{1})} \otimes \langle\text{GS}|_{2\sigma(\bar{2})} \right) . \qquad (2.86)$$

Moreover, the ground state energy of $H_{\text{eff},2}$ is twice the ground state energy of $H_{\text{eff},1}$.

Plugging (2.86) in (2.85), and using the definition of the equilibrium state (2.77) yields

$$\frac{Z_2(t)}{Z_1^2(t)} = e^{-2S_2(\rho_{\text{eq}})} , \qquad (2.87)$$

where $S_2(\rho_{\text{eq}}) = -\log \text{Tr}\rho_{\text{eq}}^2$ is the second Rényi entropy of $\rho_{\text{eq}}$. We assume that $S_2(\rho_{\text{eq}}) \to \infty$ in the thermodynamic limit. This calculation can also be done semiclassically assuming that the two-replica ground state $|\text{GS}\rangle$ is semiclassical. The leading contribution to the variance comes from a two-boundary replica wormhole schematically presented in Fig. 10. See sections 3.2 and 3.3 for the explicit construction of similar replica wormholes.

From this discussion, we see that the norm is self-averaging over the ensemble. Since the norm of individual states depends smoothly on the semiclassical features of individual states, this strongly suggests that the coarse-grained geometric properties of the ER caterpillars which can affect the gravitational action of $M_\Psi$ are self-averaging. One such quantity is the length of the wormhole. In appendix B we present a simplified toy model of the ensemble of ER caterpillars, where we show that the length of the wormhole is self-averaging over the ensemble.

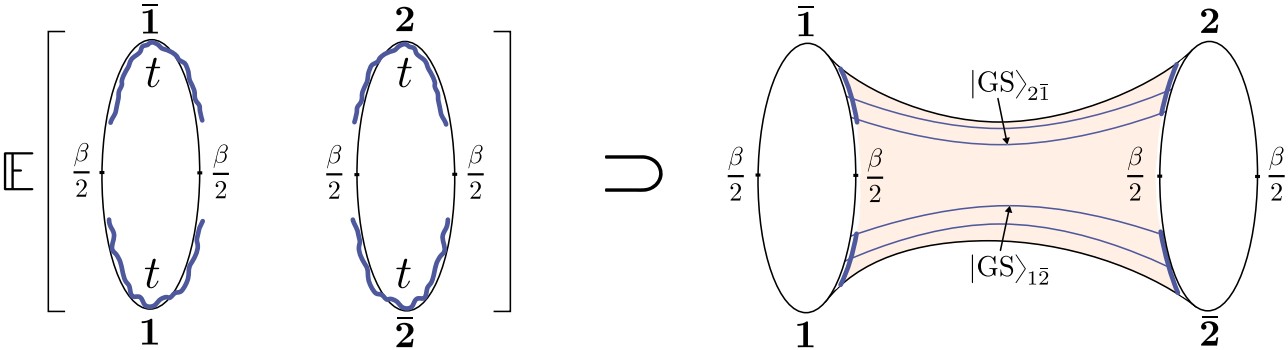

**Figure 10:** Replica wormhole contribution to $Z_2(t)$. The linearly growing regions provide a contribution $\exp(-2tE_0)$ from the ground state energy, whereas the thermal regions provide a contribution $\text{Tr}(e^{-\frac{\beta}{2}H_0}\rho(\text{GS})e^{-\frac{\beta}{2}H_0}\rho(\text{GS})) \propto \text{Tr}\rho_{\text{eq}}^2$ each. In sections 3.2 and 3.3 we provide a more detailed description of the replica wormhole and of its contribution.

A related question is whether the state $\rho_\text{L}$ (or $\rho_\text{R}$) differs from $\rho_{\text{eq}}$ by a substantial amount for single realizations in the ensemble $\mathcal{E}_\Psi^t$. The 2-norm distance between the two states is exponentially suppressed on average,

$$\mathbb{E}_{\mathcal{E}_\Psi^t} \left[ \left\| \rho_\text{L} - \rho_{\text{eq}} \right\|_2^2 \right] = e^{-S_2(\rho_{\text{eq}})} . \qquad (2.88)$$

However, this does not imply the states are close in trace distance.[23] Hence, a single realization of $\rho_L$ may still be distinguishable from $\rho_{eq}$.

## 3  The growth of randomness in gravity

In this section, we quantify the amount of microscopic randomness in the ensemble of states $\mathcal{E}_\Psi^t$ constructed in section 2, as a function of the circuit time $t$ used to prepare the states. We use standard notions in quantum information theory that we now briefly introduce.

The $k$-th moment of a general ensemble of states $\mathcal{E}_\Psi$ is defined as

$$\rho_k(\mathcal{E}_\Psi) = \mathbb{E}_{\mathcal{E}_\Psi}[(|\Psi\rangle\langle\Psi|)^{\otimes k}]. \tag{3.1}$$

The state $\rho_k(\mathcal{E}_\Psi)$ is defined on $2k$ replicas of the original Hilbert space. It is generally a mixed state, given the correlations between replicas that arise from the average over $\mathcal{E}_\Psi$. The moments (3.1) provide a very precise characterization of $\mathcal{E}_\Psi$.[24]

A *quantum state $k$-design* is an ensemble of states which has the same $k$-th moment as the random state ensemble, $\rho_k(\mathcal{E}_\Psi) = \rho_k(\mathcal{E}_\Psi^{rand})$. In order to introduce an approximate notion of a $k$-design we define the *$p$-norm distance to $k$-design* $\Delta_{k,p}(\mathcal{E}_\Psi)$ as

$$\Delta_{k,p}(\mathcal{E}_\Psi) \equiv \frac{\left\|\rho_k(\mathcal{E}_\Psi) - \rho_k(\mathcal{E}_\Psi^{rand})\right\|_p}{\left\|\rho_1(\mathcal{E}_\Psi^{rand})\right\|_p^k}. \tag{3.2}$$

for the Schatten $p$-norm $\|\cdot\|_p$, where $\|\cdot\|_1$ is the trace norm and $\|\cdot\|_2$ is the Frobenius norm. Here $\|\rho_1(\mathcal{E}_\Psi^{rand})\|_p$ sets the distance scale in the space of density matrices. Accordingly, the factor of $\|\rho_1(\mathcal{E}_\Psi^{rand})^{\otimes k}\|_p$ in the denominator of (3.2) sets the distance scale in the replicated space.

The ensemble $\mathcal{E}_\Psi$ is an *$\varepsilon$-approximate ($p$-norm) quantum state $k$-design* if $\Delta_{k,p}(\mathcal{E}_\Psi) < \varepsilon$. For $p = 1$, an $\varepsilon$-approximate quantum state $k$-design is $k$-copy indistinguishable, with a small precision $\varepsilon$, from the random state ensemble. As opposed to ensembles of random states, approximate $k$-designs form efficiently in physical systems.

The ensemble of ER caterpillars $\mathcal{E}_\Psi^t$ eventually becomes an approximate quantum state $k$-design of the black hole. The goal of this section is to compute the time to approximate $k$-design

$$t_{k,2} = \min_t \left\{\Delta_{k,2}(\mathcal{E}_\Psi^t) < \varepsilon\right\}. \tag{3.3}$$

The reason to consider the 2-norm definition of an approximate quantum state $k$-design is that, on the one hand, we will compute $t_{k,2}$ explicitly using a microscopic analysis, as well as

---

[23] For example, given a random bipartite state on $\mathcal{H} \otimes \mathcal{H}$ for a Hilbert space $\mathcal{H}$ of dimension $L$, the average reduced density matrix of one side is maximally mixed. However, while the average 2-norm distance is $1/L$, the average trace distance (1-norm) is order unity.

[24] The expectation value of any observable $\langle A \rangle = \langle\Psi|A|\Psi\rangle$ is on average fixed by the first moment $\mathbb{E}_{\mathcal{E}_\Psi}[\langle A\rangle] = \mathrm{Tr}(\rho_1(\mathcal{E}_\Psi)A)$, while its variance is additionally determined by the second moment, $\mathrm{Var}_{\mathcal{E}_\Psi}[\langle A\rangle] = \mathrm{Tr}(\rho_2(\mathcal{E}_\Psi)A \otimes A) - \mathrm{Tr}(\rho_1(\mathcal{E}_\Psi)A)^2$. Moments with $k \geqslant 2$ also determine the average value of non-linear functions of the states over $\mathcal{E}_\Psi$, such as the entanglement spectrum or other finer properties.

semiclassically in gravity. The computation of $t_{k,1}$ is more involved and it is not illuminating for the purposes of the present paper. More importantly, due to the normalization in (3.2), in the situations of interest, the 2-norm definition implies the 1-norm definition. We will show this toward the end of section 3.1.

Notice that in this general presentation we have not defined what we mean by $\mathcal{E}_\Psi^{\text{rand}}$. We will be more specific in the next subsections.

## 3.1 Microcanonical linear growth

To build intuition on the microscopic growth of randomness, we consider the microcanonical window of the black hole, of energies $E \in [M - \frac{\Delta M}{2}, M + \frac{\Delta M}{2}]$ with $\Delta M \sim T_H \ll M$. The dimension of this Hilbert space is $L = e^S$. For microcanonically regularized perturbations $\mathcal{O}_\alpha^M$ that only act within the microcanonical window, the random circuit $U(t)$ is a microcanonical unitary transformation in $\mathsf{U}(L)$ driven by the Hamiltonian $H(t) = \sum_\alpha g_\alpha(t) \mathcal{O}_\alpha^M$. This defines an ensemble of time-evolution operators at fixed time $t$ in such microcanonical window, $\mathcal{E}_{U_M}^t$. Such an ensemble can be used to prepare an ensemble of states of the form

$$|\Psi_M(t)\rangle = \frac{1}{\sqrt{L}} \sum_{i,j=1}^{L} U(t)_{ij} |E_i^*\rangle_{\mathsf{L}} \otimes |E_j\rangle_{\mathsf{R}} \; . \tag{3.4}$$

This ensemble is a microcanonical version of the ensemble of the ensemble of ER caterpillars at fixed circuit time constructed in section 2.1.

We want to compare $\mathcal{E}_{\Psi_M}^t$ to the ensemble of random states $\mathcal{E}_{\Psi_M}^{\text{rand}}$ of the form

$$|\Psi_M^{\text{rand}}\rangle = \frac{1}{\sqrt{L}} \sum_{i,j=1}^{L} U_{ij} |E_i^*\rangle_{\mathsf{L}} \otimes |E_j\rangle_{\mathsf{R}} \; , \tag{3.5}$$

where $U$ is drawn from the Haar ensemble $\mathcal{E}_{U_M}^{\text{Haar}}$ in $\mathsf{U}(L)$.[25]

**Frame potentials.** Given this setup, we will be interested in comparing the purity of the moment states for both ensembles. This is called the *k-th frame potential* of the ensemble, respectively,

$$F_k(t) \equiv \text{Tr}\left(\rho_k(\mathcal{E}_{\Psi_M}^t)^2\right) \; , \qquad F_k^{\text{rand}} \equiv \text{Tr}\left(\rho_k(\mathcal{E}_{\Psi_M}^{\text{rand}})^2\right) \; . \tag{3.6}$$

Explicitly, using the definition of the moment states (3.1) and of the ensembles (3.4) and (3.5) the frame potentials are given by

$$F_k(t) = \mathbb{E}_{\mathcal{E}_{\Psi_M}^t} |\langle \Phi_M(t)|\Psi_M(t)\rangle|^{2k} \; , \tag{3.7}$$

and

$$F_k^{\text{rand}} = \mathbb{E}_{\mathcal{E}_{\Psi_M}^{\text{rand}}} \left|\langle \Phi_M^{\text{rand}}|\Psi_M^{\text{rand}}\rangle\right|^{2k} \; . \tag{3.8}$$

---

[25] Here $\mathcal{E}_{\Psi_M}^{\text{rand}}$ is different from what is usually called the random state ensemble in the doubled microcanonical Hilbert space, i.e., states of the form $U|\Psi_0\rangle$ with $U \in \mathsf{U}(L^2)$ distributed according to the Haar measure in $\mathsf{U}(L^2)$. In particular, all of the draws of $\mathcal{E}_{\Psi_M}^{\text{rand}}$ are maximally entangled.

The expectation value is taken independently for both states in $\mathcal{E}^t_{\Psi_M}$ (3.7) or in $\mathcal{E}^{\text{rand}}_{\Psi_M}$ (3.8). Thus, the frame potential corresponds to the $k$-th moment of the magnitude of the overlap $|\langle \Phi | \Psi \rangle|^2$, over the two independent draws $|\Psi\rangle, |\Phi\rangle$ in the ensemble of quantum states.

The frame potentials are relevant quantum information theoretic measures of distinguishability because they control the 2-norm distance between moments

$$\left\| \rho_k(\mathcal{E}^t_{\Psi_M}) - \rho_k(\mathcal{E}^{\text{rand}}_{\Psi_M}) \right\|_2^2 = F_k(t) - F_k^{\text{rand}}. \tag{3.9}$$

The identity (3.9) follows from the left/right invariance of the Haar measure, which implies the property $\text{Tr}(\rho_k(\mathcal{E}^t_{\Psi_M})\rho(\mathcal{E}^{\text{rand}}_{\Psi_M})) = F_k^{\text{rand}}$.[26]

The 2-norm distance to $k$-design is then

$$\Delta_{k,2}(\mathcal{E}^t_{\Psi_M}) = \left( \frac{F_k(t) - F_k^{\text{rand}}}{(F_1^{\text{rand}})^k} \right)^{1/2}. \tag{3.10}$$

**Frame potentials of the random state ensemble.** For the random state ensemble $\mathcal{E}^{\text{rand}}_{\Psi_M}$ the relevant overlap in the frame potential $F_k^{\text{rand}}$ (3.8) can be written as

$$\langle \Phi_M^{\text{rand}} | \Psi_M^{\text{rand}} \rangle = L^{-1} \langle I | U \otimes V^* | I \rangle, \tag{3.11}$$

where we have introduced the infinite temperature TFD $|I\rangle$ between forward and backward replicas, similar to what we did in section 2.1 for the norm of the states.

Therefore, the $k$-th frame potential of $\mathcal{E}^{\text{rand}}_{\Psi_M}$ is

$$F_k^{\text{rand}} = L^{-2k} \mathbb{E}_{\mathcal{E}^{\text{Haar}}_{U_M}} \langle I |^{\otimes 2k} (U \otimes V^*)^{\otimes k} \otimes (U^* \otimes V)^{\otimes k} |I\rangle^{\otimes 2k}, \tag{3.12}$$

where the expectation is taken over $U, V$, two independent draws of the Haar ensemble $\mathcal{E}^{\text{Haar}}_{U_M}$.[27] The overlap (3.12) is defined on the $2k$ replicas of the original Hilbert space. Throughout this paper we will refer to the first $k$ replicas associated to the action of $U$ as forward replicas, and to the other $k$ replicas associated to the action of $U^*$ as backward replicas.

It is useful to define $k$-th moment superoperator of the Haar ensemble,

$$\hat{\Phi}^{(k)}_{\text{Haar}} \equiv \mathbb{E}_{\mathcal{E}^{\text{Haar}}_{U_M}} [\underbrace{U \otimes ... \otimes U}_{k} \otimes \underbrace{U^* \otimes ... \otimes U^*}_{k}]. \tag{3.13}$$

The superoperator $\hat{\Phi}^{(k)}_{\text{Haar}}$ acts on $2k$ replicas of the original Hilbert space. As explained in appendix E, the Haar moment superoperator $\hat{\Phi}^{(k)}_{\text{Haar}}$ is an orthogonal projector into a subspace of the $2k$-replica Hilbert space linearly generated by the *invariant states* $|W_\sigma\rangle$ of the form

$$|W_\sigma\rangle = \bigotimes_{r=1}^{k} |I\rangle_{r\sigma(\bar{r})}, \qquad \sigma \in \text{Sym}(k). \tag{3.14}$$

---

[26] See [115] for a recent review of the Haar measure in quantum information theory.

[27] In fact, for (3.12) the unitary $V$ can be absorbed into $U$ using the invariance of the Haar measure under left/right multiplication, and (3.12) is effectively an average over only $U$.

Each state $|W_\sigma\rangle$ corresponds to a pairing of $k$ forward and $k$ backward replicas forming infinite temperature TFD states $|I\rangle_{r\bar{s}}$. The permutation $\sigma \in \mathrm{Sym}(k)$ labels the pairing. The states $|W_\sigma\rangle$ are invariant under the action $U_0^{\otimes k} \otimes U_0^{* \otimes k}$ for $U_0 \in \mathsf{U}(L)$.[28] See appendix E for more details.

In terms of the Haar moment superoperator, the frame potential (3.12) is

$$F_k^{\mathrm{rand}} = L^{-2k} \, \mathrm{Tr}\, \hat{\Phi}_{\mathrm{Haar}}^{(k)}, \qquad (3.15)$$

where we have used that $(\hat{\Phi}_{\mathrm{Haar}}^{(k)})^2 = \hat{\Phi}_{\mathrm{Haar}}^{(k)}$. Notice that $\mathrm{Tr}\, \hat{\Phi}_{\mathrm{Haar}}^{(k)}$ simply counts the dimension of the invariant subspace.

For $k \leqslant L$ all the invariant states $|W_\sigma\rangle$ are linearly independent, and the invariant subspace has dimension $\mathrm{Tr}\, \hat{\Phi}_{\mathrm{Haar}}^{(k)} = |\mathrm{Sym}(k)| = k!$ [117]. Therefore, in this regime,

$$F_k^{\mathrm{rand}} = \frac{k!}{L^{2k}} \qquad \text{for } k \leqslant L. \qquad (3.16)$$

This agrees with other methods to compute the moments of the Haar distribution [118, 119].

**Linear growth of randomness.** Similarly, we can follow the previous steps for the ensemble of states $\mathcal{E}_{\Psi_M}^t$, for which the frame potentials $F_k(t)$ are given by

$$F_k(t) = L^{-2k} \, \mathrm{Tr}\, \left( \hat{\Phi}_t^{(k)\,2} \right), \qquad (3.17)$$

where we have defined the $k$-th moment superoperator of the ensemble of random circuits,

$$\hat{\Phi}_t^{(k)} \equiv \mathbb{E}_{\mathcal{E}_{U_M}^t} [\underbrace{U(t) \otimes ... \otimes U(t)}_{k} \otimes \underbrace{U(t)^* \otimes ... \otimes U(t)^*}_{k}]. \qquad (3.18)$$

In [75, 76] it was noticed that, for the ensemble of random circuits $\mathcal{E}_{U_M}^t$, the $k$-th moment superoperator $\hat{\Phi}_t^{(k)}$ is the Euclidean time-evolution operator

$$\hat{\Phi}_t^{(k)} = \exp(-t H_{\mathrm{eff},k}). \qquad (3.19)$$

for the time-independent effective Hamiltonian

$$H_{\mathrm{eff},k} = \frac{J}{2} \sum_{\alpha=1}^{K} \left( \sum_{r=1}^{k} \mathcal{O}_\alpha^r - \mathcal{O}_\alpha^{\bar{r}\,*} \right)^2. \qquad (3.20)$$

The effective Hamiltonian couples the $2k$ replicas via bi-local interactions. These interactions arise effectively from the gaussian correlations of the Brownian couplings defining the random circuit. The index $r$ ($\bar{r}$) labels the forward (backward) replica where the operator $\mathcal{O}_\alpha^r$ ($\mathcal{O}_\alpha^{\bar{r}\,*}$) acts. The effective Hamiltonian $H_{\mathrm{eff},k}$ has replica symmetry $\mathrm{Sym}(k) \times \mathrm{Sym}(k)$ for independent

---

[28] This symmetry is the representation of the unitary group in the replicated Hilbert space. By the Schur-Weyl duality, the product representation decomposes into irreducible representations which themselves form irreducible representations of the symmetric group $\mathrm{Sym}(k)$ of permutations between replicas. This is in accord with the fact that the collection of invariant states constitutes a trivial representation of $\mathsf{U}(L)$ in the tensor product, as well as a natural representation of $\mathrm{Sym}(k)$ (see e.g. [65, 68, 115, 116]).

permutations of the forward and backward replicas.

Therefore, the $k$-th frame potential (3.17) corresponds to an effective thermal partition function of an interacting Hamiltonian in $2k$ replicas, at inverse temperature $2t$,

$$F_k(t) = \frac{1}{L^{2k}} \operatorname{Tr} \exp\left(-2t H_{\text{eff},k}\right), \tag{3.21}$$

as represented in Fig. 11.

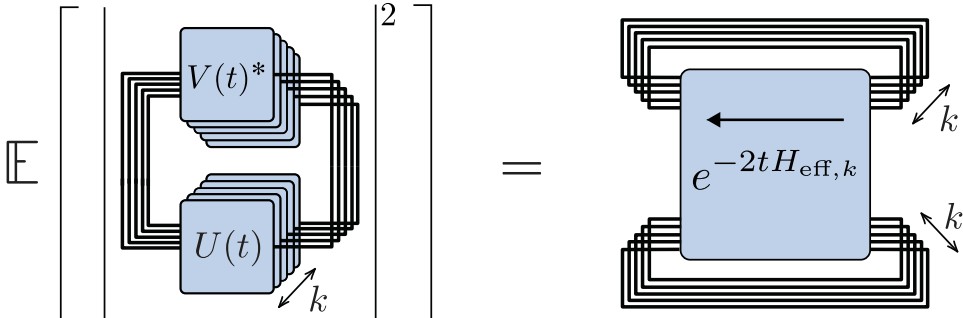

**Figure 11:** The frame potential $F_k(t)$. The average over the Brownian couplings produces local-in-time effective interactions between replicas in terms of an effective time-independent Hamiltonian $H_{\text{eff},k}$ implementing Euclidean time evolution on the $2k$ replicas. The frame potential is the thermal partition function of $H_{\text{eff},k}$ at inverse temperature $2t$.

The invariant states $|W_\sigma\rangle$ defined in (3.14) are ground states of the effective Hamiltonian $H_{\text{eff},k}$,

$$H_{\text{eff},k} |W_\sigma\rangle = 0. \tag{3.22}$$

The infinite temperature TFD has the property that $A^r |I\rangle_{r\bar{s}} = (A^{\bar{s}\,*})^\dagger |I\rangle_{r\bar{s}}$ for any operator $A$, and thus $|W_\sigma\rangle$ annihilates $H_{\text{eff},k}$. Notice that the invariant states (3.14) break the replica symmetry of the effective Hamiltonian $H_{\text{eff},k}$.

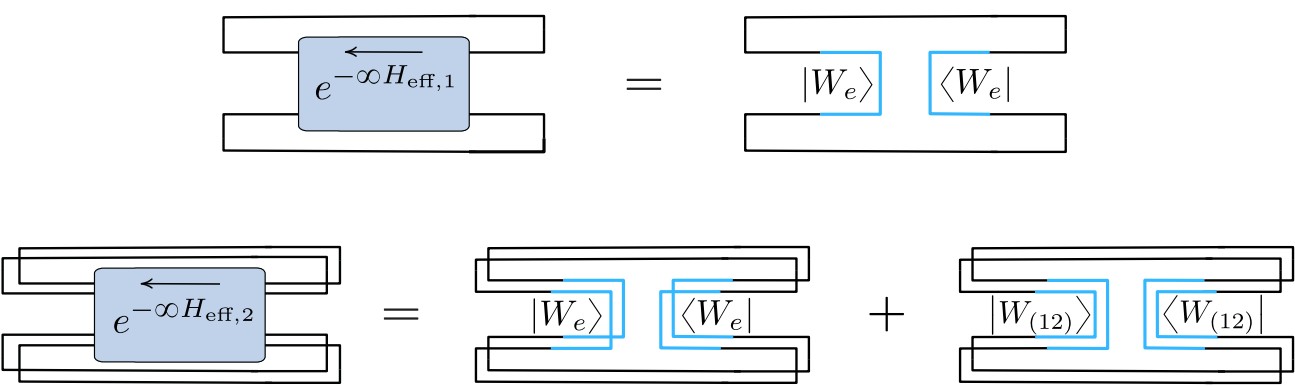

**Figure 12:** At infinite times, the frame potential $F_k(\infty)$ is determined by the number of independent ground states of the effective Hamiltonian $H_{\text{eff},k}$. Under genericity assumptions, the invariant states $|W_\sigma\rangle$ generate the ground space of $H_{\text{eff},k}$ and the ground space dimension is $k!$ for $k \leqslant L$.

Assuming that the set of drive operators $\{\mathcal{O}_\alpha\}$ is generic enough, then there will be no other

ground state of $H_{\text{eff},k}$. By generic, we mean that the set of operators lacks of global symmetries preserved by all of the operators, and that there are no clusters of degrees of freedom which are not coupled by the operators [75, 76]. The late-time limit of the spectral partition function then corresponds to the number of ground states of the effective Hamiltonian (see Fig. 12). Under this general assumption, we conclude that

$$F_k(\infty) = \frac{k!}{L^{2k}} = F_k^{\text{rand}} \qquad \text{for } k \leqslant L \,. \tag{3.23}$$

Therefore, at infinite time, the ensemble $\mathcal{E}_\Psi^\infty$ is a quantum state $k$-design.

At finite time $t$, the approach to a $k$-design is controlled by the spectral gap $E_{\text{gap}}^{(k)}$ and the number of first excited states $N_*^{(k)}$ of the effective Hamiltonian $H_{\text{eff},k}$. For large enough $t$, or low effective temperatures, the contribution to the thermal partition function $F_k(t)$ of the excited states will be approximately dominated by the first excited states of the effective Hamiltonian, and

$$\frac{F_k(t) - F_k^{\text{rand}}}{(F_1^{\text{rand}})^k} \approx N_*^{(k)} \, e^{-2tE_{\text{gap}}^{(k)}} \,. \tag{3.24}$$

Given this structure, generally, there are two properties which guarantee a linear time to design. The properties are:

1. *The gap of the effective Hamiltonian is independent of $k$,*

$$E_{\text{gap}}^{(k)} = E_{\text{gap}} \,. \tag{3.25}$$

2. *The number of first excited states scales exponentially with $k$, and in particular,*

$$N_*^{(k)} = N_* \, k! \, k \,. \tag{3.26}$$

The reason to expect (3.25) and (3.26) for the black hole is explained in appendix C. For large-$N$ systems with semiclassical description, the ground states of $H_{\text{eff},k}$ admit collective semiclassical description in terms of two-replica connected configurations representing infinite temperature TFDs [75]. The low-energy sector of $H_{\text{eff},k}$ breaks the replica symmetry of the Hamiltonian and it is controlled by two-replica physics. The first excited states correspond to approximate single-replica quasi-particle excitations on top of each ground state. Therefore, the number of first excited states is proportional to the number of ground states, $k!$, to the number of single particle excitations per ground state, $N_*$, and to the number of replicas one can excite per ground state, $k$.[29] Moreover, the energy of the single particle excitations is independent of the number of ground states.

Using (3.10) and (3.24), under the two conditions above, the 2-norm distance to design behaves as

$$\Delta_{k,2}(\mathcal{E}_{\Psi_M}^t)^2 \approx N_* \, k! \, k e^{-2tE_{\text{gap}}} \,, \tag{3.27}$$

---

[29] This number is $k$ instead of $2k$ because acting on the forward $r$ replica of $|I\rangle_{r\bar{s}}$ generates the same action as acting on the backward $\bar{s}$ replica with the CPT conjugate of the operator.

For $k \gg 1$, this leads to a time to approximate $k$-design

$$t_{k,2} \approx \frac{1}{2} E_{\text{gap}}^{-1} \left( k \log k - k + \log k + \log N_* + 2 \log \varepsilon^{-1} \right) . \tag{3.28}$$

Up to logarithmic factors, this is a linear growth with $k$, with slope $(2E_{\text{gap}})^{-1}$. From $k = 1$, we can read the onset time of this linear growth is $t_\star \sim (2E_{\text{gap}})^{-1} \log N_*$.

**Spectral randomness.** As an additional remark, to understand where the growth of randomness comes from, it is instructive to write the states (3.4) in diagonal form as

$$|\Psi_M(t)\rangle = \frac{1}{\sqrt{L}} \sum_{i=1}^{L} e^{-i\theta_i(t)} |\psi_i(t)^*\rangle_\mathsf{L} \otimes |\psi_i(t)\rangle_\mathsf{R} , \tag{3.29}$$

where $|\psi_i(t)\rangle$ is an eigenvector of $U(t)$ with eigenphase $\exp(-i\theta_i(t))$. In this form, the eigenphases parametrize different purifications of the maximally mixed state in the doubled Hilbert space.

For the ensemble of states $\mathcal{E}_\Psi^t$, the frame potentials $F_k(t)$ only depend on the distribution of eigenphases $\exp(-i\theta_i(t))$ over the ensemble.[30] Formally, the ensemble of eigenphases is determined by a spectral probability distribution $p_t(\theta_1, ..., \theta_L)$ induced by the probability distribution in $\mathsf{U}(L)$ defining the ensemble of random circuits $\mathcal{E}_{U_M}^t$ at fixed circuit time $t$. The frame potential is given by the $k$-th spectral form factor (SFF) of the ensemble of random circuits

$$F_k(t) = \mathbb{E}_{\mathcal{E}_{U_M}^t} \left[ \text{SFF}(2t)^k \right] , \qquad \text{SFF}(t) \equiv \frac{|\text{Tr}\, U(t)|^2}{L^2} . \tag{3.30}$$

In this form it is manifest that the frame potential only depends on the distribution of eigenphases of $U(t)$. More explicitly, let $\rho_t(\theta) = \sum_i \delta(\theta - \theta_i(t))$ be the eigenphase density of $U(t)$. Then, $\mathbb{E}_{\mathcal{E}_{U_M}^t}[\text{SFF}(t)]$ is completely determined by $\mathbb{E}_{\mathcal{E}_{U_M}^t}[\rho_t(\theta)\rho_t(\theta')]$.[31] Similarly, the value of $F_k(t)$ is determined by the $2k$-th moment of the eigenphase density, $\mathbb{E}_{\mathcal{E}_{U_M}^t}[\rho_t(\theta_1)...\rho_t(\theta_{2k})]$.

Therefore, the distance of $\mathcal{E}_\Psi^t$ to the random ensemble is determined by the proximity, in a weak sense, of the probability distribution $p_t(\theta_1, ..., \theta_L)$ characterizing the distribution of eigenphases of the random circuit $U(t)$, to the probability distribution determining the eigenphases of the Haar distribution. The eigenphases of a Haar random unitary are distributed according to the circular unitary ensemble (CUE) [120]

$$p_{\text{CUE}}(\theta_1, ..., \theta_L) = \frac{1}{L!(2\pi)^L} \prod_{i<j} \left| e^{-i\theta_i} - e^{-i\theta_j} \right|^2 . \tag{3.31}$$

A distinctive property of the CUE is that the nearby eigenphases tend to repel each other. We illustrate this empirically in Fig. 13.

---

[30] This follows from the Markovian nature of the Brownian couplings, since the frame potential (3.7) satisfies $\mathbb{E}_{\mathcal{E}_{U_M}^t} \left| \text{Tr}\left( U(t)V(t)^\dagger \right) \right|^{2k} = \mathbb{E}_{\mathcal{E}_{U_M}^t} |\text{Tr}\, U(2t)|^{2k}$.

[31] For unitaries generated by time-independent Hamiltonians the eigenphases are of the form $\theta_j = E_j t$ where $E_j$ is the energy. The $\text{SFF}(t)$ – viewed as a function at different times – is then proportional to the Fourier transform of the two-point eigenvalue density in the spectrum of the time-independent Hamiltonian. In appendix F we provide a detailed comparison between $F_1(t)$ and the SFF for a time-independent Hamiltonian.

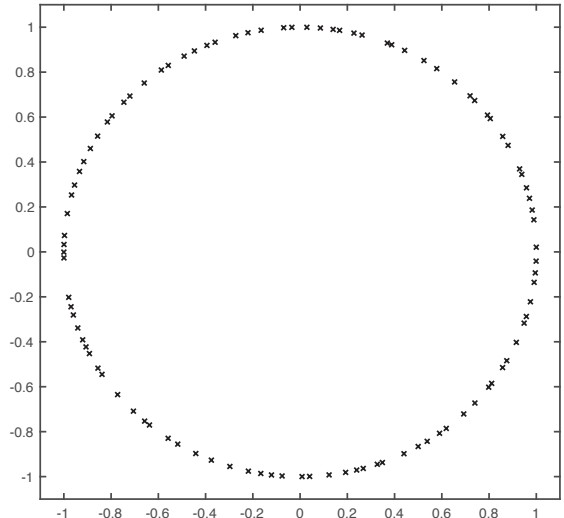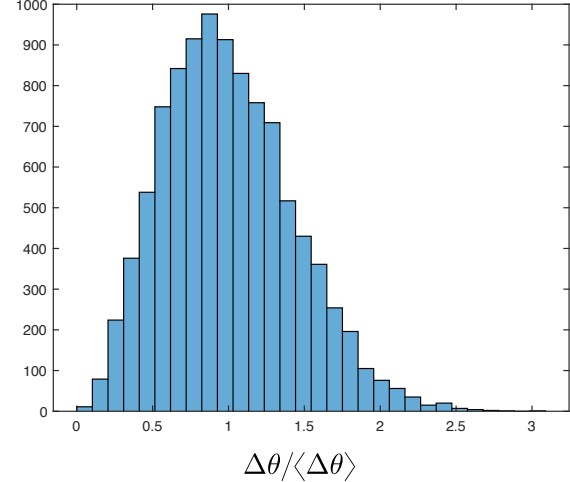

**Figure 13:** On the left, the spectrum of a Haar random unitary on a Hilbert space of dimension $L = 100$. On the right, histogram of the unfolded eigenphase separations for a Haar random unitary with $L = 10^4$. In the CUE, the eigenphases tend to repel each other.

As circuit time $t$ evolves, the linear growth in randomness essentially signals the weak convergence of $p_t(\theta_1, ..., \theta_L)$ to $p_{\text{CUE}}(\theta_1, ..., \theta_L)$.

**Relation to trace distance definition.** From the monotonicity of Schatten $p$-norms, we know that $\|\rho_k(\mathcal{E}_{\Psi_M}) - \rho_k(\mathcal{E}^{\text{rand}}_{\Psi_M})\|_2 \leqslant \|\rho_k(\mathcal{E}_{\Psi_M}) - \rho_k(\mathcal{E}^{\text{rand}}_{\Psi_M})\|_1$. Moreover, from Hölder's inequality, $\|\rho_k(\mathcal{E}_{\Psi_M}) - \rho_k(\mathcal{E}^{\text{rand}}_{\Psi_M})\|_1 \leqslant L^k \|\rho_k(\mathcal{E}_{\Psi_M}) - \rho_k(\mathcal{E}^{\text{rand}}_{\Psi_M})\|_2$. Using that $\|\rho_1(\mathcal{E}^{\text{rand}}_{\Psi_M})\|_2 = (F_1^{\text{rand}})^{k/2} = L^{-k}$ and the definition (3.2) we have that,

$$L^{-k} \Delta_{k,2} \leqslant \Delta_{k,1} \leqslant \Delta_{k,2} \ . \tag{3.32}$$

Therefore, the 2-norm definition of a $k$-design that we use in terms of $\Delta_{k,2}$ is more restrictive. The upper bound implies that $t_{k,1} \leqslant t_{k,2}$.

## 3.2 Semiclassical linear growth

We now study the growth of randomness for the ensemble of ER caterpillars $\mathcal{E}_\Psi^t$ of the black hole constructed in section 2.1. We write the states down here again

$$|\Psi(t)\rangle \ = \ \frac{1}{\sqrt{Z(t)Z(\beta)}} \sum_{i,j} e^{-\frac{\beta}{4}(E_i+E_j)} U(\Gamma_t)_{ij} |E_i^*\rangle_{\mathsf{L}} \otimes |E_j\rangle_{\mathsf{R}} \ , \tag{3.33}$$

where $U(\Gamma_t)$ is the gradually cooled random circuit (2.4). The states (3.33) are approximately normalized given that they are unit-normalized on average, and their norm is self-averaging over the ensemble, according to the discussion in section 2.4.

We want to compare the ensemble of ER caterpillars $\mathcal{E}_\Psi^t$ to the ensemble of random states of the two-sided black hole, $\mathcal{E}_\Psi^{\text{rand}}$. For the time being, we will use the infinite-time ensemble $\mathcal{E}_\Psi^\infty$ as

our reference random state ensemble

$$|\Psi_{\text{rand}}\rangle = \frac{1}{\sqrt{Z(\infty)Z(\beta)}} \sum_{i,j} e^{-\frac{\beta}{4}(E_i+E_j)} U(\Gamma_\infty)_{ij} |E_i^*\rangle_{\mathsf{L}} \otimes |E_j\rangle_{\mathsf{R}} .$$
(3.34)

In section 3.4 we provide a more explicit and intrinsic definition of the random ensemble $\mathcal{E}_\Psi^{\text{rand}}$.

**Moment two-point function.** In this case we shall consider the overlap between states at different circuit times

$$\langle\Phi(t)|\Psi(t')\rangle = \frac{1}{Z(t)} \langle\text{TFD}| U(\Gamma_{t'}) \otimes V(\Gamma_t)^* |\text{TFD}\rangle .$$
(3.35)

Microscopically, this overlap is defined as a closed CFT path integral of two independent random circuits for time $t$ and $t'$, glued together by the Euclidean preparation of the TFD states, as shown in Fig. 14.

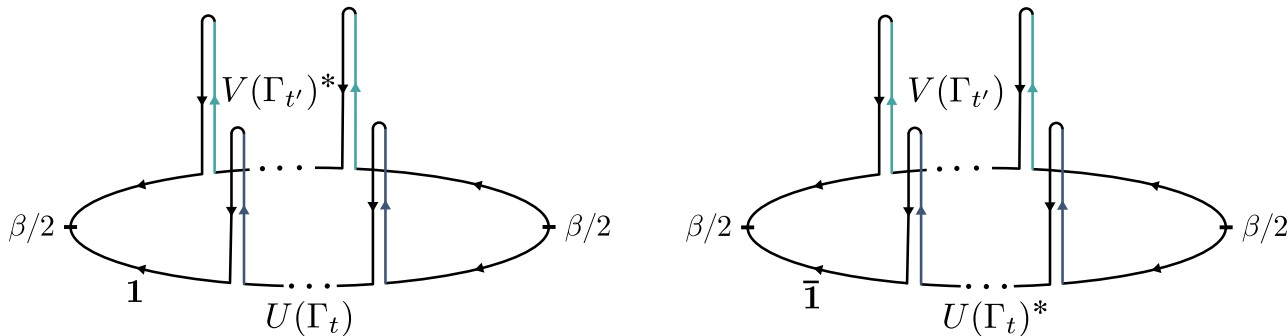

**Figure 14:** The boundary path integral time-contours for $|\langle\Phi(t)|\Psi(t')\rangle|^2$ consist of two disconnected replicas. We denote them by 1 (forward) and $\bar{1}$ (backward) replicas, corresponding to $U(\Gamma_t)$ and $U(\Gamma_t)^*$, respectively. Each replica involves two independent draws of the gradually cooled random circuit, $U(\Gamma_t)$ and $V(\Gamma_{t'})$, at times $t$ and $t'$, respectively. The quantity $G_k(t,t')$ involves $k$ copies of the forward and backward replicas. The disorder over couplings is implemented independently on $U(\Gamma_t)$ and on $V(\Gamma_{t'})$.

We shall define the *k-th moment two-point function* of the ensemble as

$$G_k(t,t') \equiv \text{Tr}\left(\rho_k(\mathcal{E}_\Psi^t)\rho_k(\mathcal{E}_\Psi^{t'})\right) = \mathbb{E}_{\mathcal{E}_\Psi^t, \mathcal{E}_\Psi^{t'}} |\langle\Phi(t)|\Psi(t')\rangle|^{2k} .$$
(3.36)

This quantity corresponds to a disordered CFT path integral on $k$ forward contours computing the overlap (3.35) and other $k$ backward contours computing its complex conjugate. Note that $F_k(t) = G_k(t,t)$ is the $k$-th frame potential of $\mathcal{E}_\Psi^t$.

The moment two-point functions are useful quantum information theoretic measures of distinguishability because they control the 2-norm distance to design

$$\left\|\rho_k(\mathcal{E}_\Psi^t) - \rho_k(\mathcal{E}_\Psi^{\text{rand}})\right\|_2^2 = F_k(t) - 2G_k(t,\infty) + F_k(\infty) .$$
(3.37)

In the infinite temperature case, $G_k(t,\infty) \to F_k(\infty)$, and this recovers (3.9). In the finite temperature case, this will no longer be true and one must use (3.37) instead. The distance to

design (3.2) is then

$$\Delta_{k,2}(\mathcal{E}_\Psi^t) = \left( \frac{F_k(t) - 2G_k(t,\infty) + F_k(\infty)}{F_1(\infty)^k} \right)^{1/2} . \tag{3.38}$$

We will take exact averages over the random couplings using the identity

$$\mathbb{E}\left[ U(\Gamma_t)^{\otimes k} \otimes U(\Gamma_t)^{*\,\otimes k} \right] = \exp(-tH_{\text{eff},k}) , \tag{3.39}$$

for the time-independent effective Hamiltonian

$$H_{\text{eff},k} = \sum_{r=1}^{k} (H_0^r + H_0^{\bar{r}}) + \frac{J}{2} \sum_{\alpha=1}^{K} \left( \sum_{r=1}^{k} \mathcal{O}_\alpha^r - \mathcal{O}_\alpha^{\bar{r}\,*} \right)^2 . \tag{3.40}$$

Moreover, it will be useful to define the bare $2k$ replica Hamiltonian

$$H_{0,k} \equiv \sum_{r=1}^{k} (H_0^r + H_0^{\bar{r}}) . \tag{3.41}$$

The term $H_{0,k}$ distinguishes the effective Hamiltonian $H_{\text{eff},k}$ (3.40) from its infinite temperature version (3.20). Notice that for $k = 1$ the effective Hamiltonian $H_{\text{eff},1}$ has already appeared in (2.14) in the preparation of the ER caterpillars. For $k = 2$ the effective Hamiltonian $H_{\text{eff},2}$ has been used in (2.84) to compute variances over the ensemble of states in section 2.4.

Using (3.35), (3.36) and (3.39), the $k$-th moment two-point function is

$$G_k(t,t') = \frac{1}{Z(t)^k Z(t')^k Z(\beta)^{2k}} \text{Tr} \left( e^{-\frac{\beta}{2} H_{0,k}} e^{-t H_{\text{eff},k}} e^{-\frac{\beta}{2} H_{0,k}} e^{-t' H_{\text{eff},k}} \right) . \tag{3.42}$$

**Infinite time.** At infinite time $\exp(-\infty H_{\text{eff},k})$ acts as an orthogonal projector, up to an overall normalization, into the ground space of the effective Hamiltonian. In appendix C we provide general comments on the structure of the $2k$-replica effective Hamiltonian $H_{\text{eff},k}$ (3.40), in particular when it comes to the expectation for its ground space and lowest excitations. At infinite temperature $J \to \infty$, $H_{\text{eff},k}$ reduces to (3.20), and its ground states break replica symmetry and consist of products of the two-replica ground state (3.14). Here we will assume that for large enough $J$ the same replica-symmetry breaking structure holds for the ground space

$$|\text{GS},\sigma\rangle = \bigotimes_{r=1}^{k} |\text{GS}\rangle_{r\sigma(\bar{r})} , \qquad \sigma \in \text{Sym}(k) , \tag{3.43}$$

where $|\text{GS}\rangle$ is the ground state of the two-replica effective Hamiltonian $H_{\text{eff},1}$. The energy of $|\text{GS},\sigma\rangle$ is $kE_0$ in the thermodynamic limit, where $E_0$ is the ground state energy of $H_{\text{eff},1}$ (see the argument around (C.8)).

It will be useful to define the orthogonal projector to the $2k$-replica ground space

$$\Pi_{\text{GS},k} = \sum_{\sigma,\tau \in \text{Sym}(k)} \tilde{c}_{\sigma,\tau} |\text{GS},\sigma\rangle\langle\text{GS},\tau| . \tag{3.44}$$

The matrix of coefficients $\tilde{c}_{\sigma,\tau}$ is the inverse of the Gram matrix of overlaps $\tilde{c}_{\sigma,\tau}^{-1} = \langle \text{GS}, \sigma | \text{GS}, \tau \rangle.$[32]

Moreover, it will be convenient to define the replica equilibrium states,

$$|\rho_{\text{eq}}, \sigma\rangle \equiv \bigotimes_{r=1}^{k} |\rho_{\text{eq}}\rangle_{r\sigma(\bar{r})} \,, \qquad |\rho_{\text{eq}}\rangle \equiv \sum_{i} (\rho_{\text{eq}})_{ii} |E_i^*\rangle \otimes |E_i\rangle \,, \tag{3.45}$$

where the two-replica equilibrium state $\rho_{\text{eq}}$ is defined in (2.77) or more explicitly in 2.79. The relation between the replica equilibrium states and the replica ground states is

$$|\rho_{\text{eq}}, \sigma\rangle = \mathcal{N}^{-k} e^{-\frac{\beta}{4} H_{0,k}} |\text{GS}, \sigma\rangle \,. \tag{3.46}$$

This relation follows from the definition of the equilibrium state $\rho_{\text{eq}}$ in (2.77). The normalization $\mathcal{N}$ has been defined in (2.78).

The replica equilibrium states are not orthonormal, given that for $\sigma, \tau \in \text{Sym}(k)$,

$$\langle \rho_{\text{eq}}, \sigma | \rho_{\text{eq}}, \tau \rangle = \text{Tr}\, \rho_{\text{eq}}^{2s_1} ... \text{Tr}\, \rho_{\text{eq}}^{2s_\#} \tag{3.47}$$

where $\#$ is the number of cycles of $\sigma\tau^{-1}$ and $\{s_1, ..., s_n\}$ are the lengths of the cycles. For $\sigma = \tau$ we get

$$\langle \rho_{\text{eq}}, \sigma | \rho_{\text{eq}}, \sigma \rangle = \left( \text{Tr}\, \rho_{\text{eq}}^2 \right)^k \,. \tag{3.48}$$

For $\sigma \neq \tau$, the overlaps are suppressed with respect to (3.48) by powers of the second and higher Rényi entropies of the equilibrium state $\rho_{\text{eq}}$.

The operator $\exp(-\infty H_{\text{eff},k})$ is an orthogonal projector to its ground space, up to the normalization $\exp(-\infty k E_0)$ from the ground state energy. This normalization partially cancels with $Z(\infty)^k$ in the denominator of (3.42). The projector then becomes,

$$\frac{1}{Z(\infty)^k Z(\beta)^k} e^{-\frac{\beta}{4} H_{0,k}} e^{-\infty H_{\text{eff},k}} e^{-\frac{\beta}{4} H_{0,k}} = \sum_{\sigma,\tau \in \text{Sym}(k)} \tilde{c}_{\sigma,\tau} |\rho_{\text{eq}}, \sigma\rangle\langle \rho_{\text{eq}}, \tau| \,. \tag{3.49}$$

We can insert the definition (3.49) in (3.42) to compute the infinite time frame potential $G_k(\infty, \infty) = F_k(\infty)$. This leads to

$$F_k(\infty) = \sum_{\sigma,\omega,\tau,\kappa \in \text{Sym}(k)} \tilde{c}_{\sigma,\omega} \tilde{c}_{\kappa,\tau} \langle \rho_{\text{eq}}, \sigma | \rho_{\text{eq}}, \tau \rangle\langle \rho_{\text{eq}}, \kappa | \rho_{\text{eq}}, \omega \rangle \,. \tag{3.50}$$

Up to small corrections, the leading term in (3.50) comes from the diagonal terms in the sum, with $\sigma = \tau = \omega = \kappa$. The infinite time frame potentials thus have the form is of the same form as (3.23),

$$F_k(\infty) \approx \frac{k!}{L_{\text{eff}}^{2k}} \,, \tag{3.51}$$

---

[32] The Gram matrix of overlaps is $\langle \text{GS}, \sigma | \text{GS}, \tau \rangle = \text{Tr}\, \rho(\text{GS})^{2s_1} ... \text{Tr}\, \rho(\text{GS})^{2s_\#}$, where $\#$ is the number of cycles of $\sigma\tau^{-1}$ and $\{s_1, ..., s_n\}$ are the lengths of the cycles. The density matrix $\rho(\text{GS})$ has been defined in (2.72). To leading order in the purity of $\rho(\text{GS})$ the states are orthonormal. The coefficients $\tilde{c}_{\sigma,\tau}$ are the inverse of the Gram matrix, which exists for $k^{-1} \gg \text{Tr}\rho(\text{GS})^2$.

where the role of the Hilbert space dimension is given by the inverse purity of the equilibrium state,

$$L_{\text{eff}} \equiv (\text{Tr}\, \rho_{\text{eq}}^2)^{-1} = e^{S_2(\rho_{\text{eq}})}.\tag{3.52}$$

Here $S_2(\rho_{\text{eq}})$ is the second Rényi entropy of the equilibrium state, assumed to scale extensively in the thermodynamic limit. As long as $k \ll L_{\text{eff}}$, the rest of the terms in (3.50) are suppressed with respect to (3.51) by factors of $L_{\text{eff}}$ or other exponential factors in higher Rényi entropies of the equilibrium state. They can be computed using (3.47) and the form of $\tilde{c}_{\sigma,\omega}$ in footnote 32.

**Linear growth from a two-boundary wormhole.** The ground state $|\text{GS}\rangle$ of the two-replica effective Hamiltonian $H_{\text{eff},1}$ has been argued to have a semiclassical description in section 2.2 in terms of a connected two-boundary spatial wormhole under some general assumptions. Moreover, in section 2.3, we have constructed this wormhole for near-extremal black holes appealing to the construction of the eternal traversable wormhole by Maldacena and Qi [61].

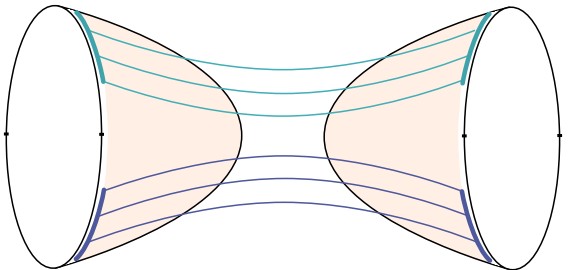 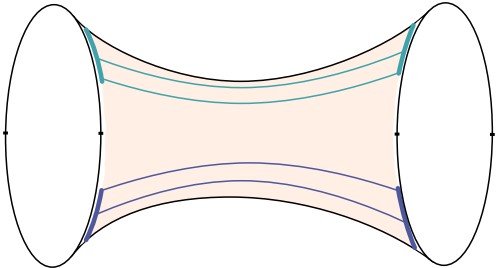

**Figure 15:** Semiclassical disconnected and connected contributions to $G_1(t,t')$. The disconnected configuration has small correlations between replicas. Its effective energy above the ground state is $\sim JK$ and its contribution to $G_1(t,t')$ decays as $\exp(-JK(t+t'))$. On the other hand, the connected wormhole has large correlations between replicas and its action vanishes at late times. The wormhole represents the contribution from the semiclassical ground state $|\text{GS}\rangle$ of the effective Hamiltonian $H_{\text{eff},1}$.

Consider $k = 1$. As in section 2.2, our argument will be mostly symmetry-based in this section, and we will leave the details for section 3.3. The first moment two-point function $G_1(t,t')$ can be evaluated in the semiclassical limit using the gravitational path integral, where we expect two saddle point contributions of the form

$$G_1(t,t') \sim Z_{\text{disc}}(t,t') + Z_{\text{wh}}(t,t').\tag{3.53}$$

We illustrate both saddle points in Fig. 15. The contribution $Z_{\text{disc}}(t,t')$ refers to the disconnected saddle point, composed of two manifolds of topology $\mathbf{D}_2 \times \mathbf{S}^{d-1}$, where $\mathbf{D}_2$ is a two-dimensional disk. The metric on each connected component of this saddle point corresponds to a Euclidean black hole backreacted by the presence of matter fields. In this case, the two replicas contain small correlations, and the energy of the configuration as measured by $H_{\text{eff},1}$ is large, scaling with $JK$. Therefore, such a contribution decays very rapidly,

$$Z_{\text{disc}}(t,t') \propto e^{-JK(t+t')}.\tag{3.54}$$

The second and more relevant contribution for our purposes is the Euclidean wormhole contribution $Z_{\text{wh}}(t,t')$ in (3.53). Under the same assumptions as in section 2.2, the wormhole captures

the ground state contribution to $G_1(t, t')$. When $t, t' \gg t_\star$, the Euclidean evolutions $\exp(-tH_{\text{eff},1})$ and $\exp(-t'H_{\text{eff},1})$ in (3.42) prepare the ground state of $H_{\text{eff},1}$, up to some normalization. Thus, the wormhole geometry must include two very long regions of a spherically-symmetric Euclidean wormhole with approximate $\tau$-translation symmetry,

$$\mathrm{d}s_E^2 = g(\rho)\mathrm{d}\tau^2 + \mathrm{d}\rho^2 + r(\rho)^2\mathrm{d}\Omega_{d-1}^2 , \tag{3.55}$$

where, again, $g(\rho)$ and $r(\rho)$ are specific to the details of the perturbations and characterize the ground state $|\text{GS}\rangle$ of $H_{\text{eff},1}$. The difference with the wormhole used to prepare the states in section 2.2 is that in this case the $\tau$ coordinate is globally completed to a periodic coordinate, and the topology of the saddle point is $\mathbf{R} \times \mathbf{S}^1 \times \mathbf{S}^{d-1}$, as shown in Fig. 16.

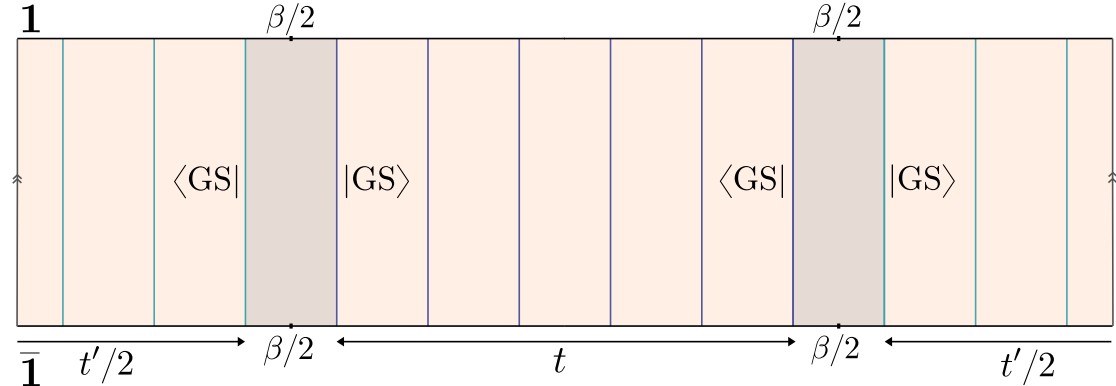

**Figure 16:** Deconstruction of the Euclidean wormhole for $t, t' \gg t_\star$. Each point corresponds to a spatial $\mathbf{S}^{d-1}$. The wormhole consists of two linearly growing regions (with $t$ and $t'$ respectively) where the metric is approximately static (3.55). Additionally the two dark regions are $t, t'$-independent and their contribution is, up to some normalization, $\langle\rho_{\text{eq}}|\rho_{\text{eq}}\rangle = \text{Tr}\,\rho_{\text{eq}}^2$, i.e., the purity of the equilibrium state. The leading corrections to the action of the wormhole at large $t, t'$ come from the lightest excitations on the static regions.

At late times $t, t' \gg t_\star$ the wormhole action $Z_{\text{wh}}(t, t')$ can be divided into four parts, as shown in Fig. 16. Two parts consist of static solutions of evolution by $t$ and $t'$ with (3.55), whose contribution is $\exp(-tE_0)$ and $\exp(-t'E_0)$ from the ground state energy. These factors cancel with the normalization factors $Z(t)$ and $Z(t')$ in (3.42). Additionally, each of the other two parts of the geometry semiclassically computes the overlap

$$\text{Tr}\left(e^{-\frac{\beta}{2}H_0}\rho(\text{GS})e^{-\frac{\beta}{2}H_0}\rho(\text{GS})\right) = \mathcal{N}^2 \langle\rho_{\text{eq}}|\rho_{\text{eq}}\rangle , \tag{3.56}$$

where we used the relation (3.46) of the replica equilibrium state (2.77). Since the normalization is $Z(t)Z(\beta)^{1/2} = \exp(-tE_0)\mathcal{N}$, we have that the total contribution from the wormhole at late times is

$$Z_{\text{wh}}(\infty, \infty) = \lim_{t,t'\to\infty} \frac{e^{-(t+t')E_0}\mathcal{N}^4}{Z(t)Z(t')Z(\beta)^2} \langle\rho_{\text{eq}}|\rho_{\text{eq}}\rangle^2 = \left(\text{Tr}\,\rho_{\text{eq}}^2\right)^2 = \frac{1}{L_{\text{eff}}^2} . \tag{3.57}$$

This coincides with the ground state contribution to $F_1(\infty)$ (3.51).

The approach to (3.57) is controlled by the gapped 1-loop fluctuations on the wormhole,

which at late times provide the contribution

$$Z_{\text{wh}}(t, t') \approx F_1(\infty) \left( 1 + N_* e^{-(t+t')E_{\text{gap}}} \right) , \tag{3.58}$$

where $N_*$ is the number of first excited states, and $E_{\text{gap}}$ is their energy above the ground state energy $E_0$, as measured by $H_{\text{eff},1}$.

Using (3.38), this leads to a 2-norm distance

$$\Delta_{1,2}(\mathcal{E}_\Psi^t)^2 \approx N_* e^{-2tE_{\text{gap}}} , \tag{3.59}$$

and the corresponding time to approximate 1-design,

$$t_{1,2} \approx \frac{1}{2} E_{\text{gap}}^{-1} \left( \log N_* + 2 \log \varepsilon^{-1} \right) . \tag{3.60}$$

Consider now $k > 1$. Our main assumption here is that the structure of the ground space of $H_{\text{eff},k}$ is the one presented in (3.43). Semiclassically, each ground state $|\text{GS}, \sigma\rangle$ corresponds to $k$ two-boundary wormhole configurations between forward and backward replicas, connected in a way given by the permutation $\sigma$, as shown in Fig. 17 for $k = 2$.

In what follows we shall only restrict to disconnected saddle points constructed out of the disk and the two-boundary wormhole. This will be enough for our purposes, given that we will be able to capture the late-time regime of the $k$-th moment two-point function $G_k(t, t')$ coming from the ground space and first excited states. From these two contributions, we expect a semiclassical contribution of the form

$$G_k(t, t') \sim \sum_{n=0}^{k} n! \binom{k}{n}^2 Z_{\text{wh}}(t, t')^n Z_{\text{disc}}(t, t')^{k-n} , \tag{3.61}$$

where $n$ represents the number of two-boundary wormholes of the corresponding saddle point. The combinatorial prefactor is the degeneracy of $2k$-replica saddle points with $n$ two-boundary wormholes. The only non-vanishing contribution at infinite time comes from $n = k$, as expected from the ground space structure. All the rest of the contributions decay extensively with the number of perturbations.

Therefore, using (3.61) for $k = n$ and (3.58), at late times the $k$-th moment two-point function of the ensemble is

$$G_k(t, t') \approx F_k(\infty) \left( 1 + k N_* e^{-E_{\text{gap}}(t+t')} \right) , \tag{3.62}$$

where $F_k(\infty) = k! F_1(\infty)$, and the number of first excited states is $k N_*$, corresponding to excitations on each of the $k$ two-boundary replicas. Using (3.37), this leads to a 2-norm distance to $k$-design

$$\Delta_{k,2}(\mathcal{E}_\Psi^t)^2 \approx N_* k! \, k \, e^{-2tE_{\text{gap}}} , \tag{3.63}$$

and thus, to a time to approximate $k$-design,

$$t_{k,2} \approx \frac{1}{2} E_{\text{gap}}^{-1} \left( k \log k - k + \log k + \log N_* + 2 \log \varepsilon^{-1} \right) . \tag{3.64}$$

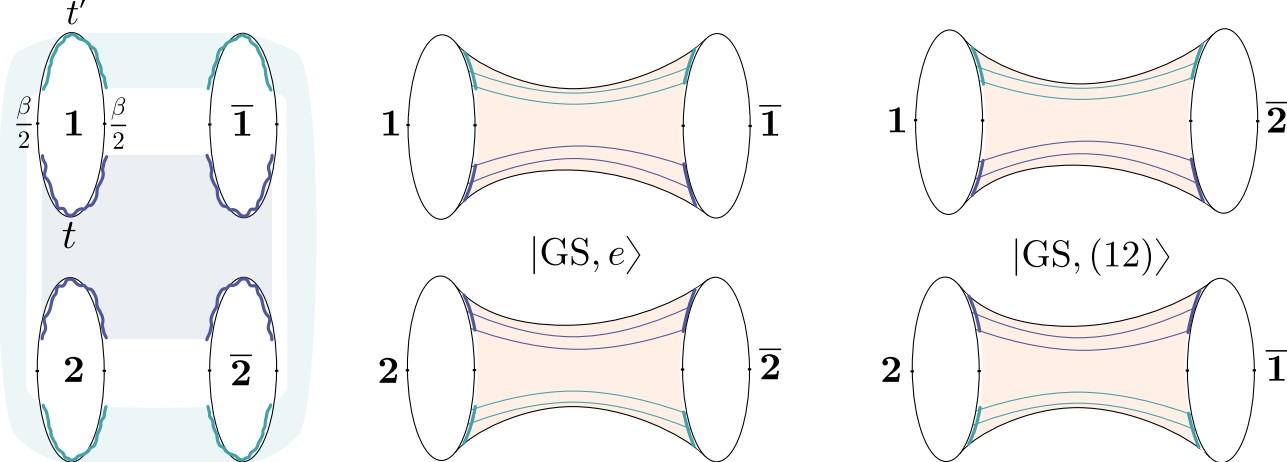

**Figure 17:** On the left, the moment two-point function $G_2(t, t')$ corresponds to an effective CFT partition function with regions which are evolved with an effective Hamiltonian $H_{\text{eff},2}$ coupling two forward and two backward replicas. In the middle, the semiclassical contribution to $G_2(t, t')$ from the ground state $|\text{GS}, e\rangle$ associated to the identity permutation. On the right, the semiclassical contribution from $|\text{GS}, (12)\rangle$ where the two backward replicas have been swapped. The ground states break the replica symmetry of $H_{\text{eff},2}$. For large $t, t'$ the moment two-point function $G_2(t, t')$ is dominated by the ground states and first excitations on the wormholes.

The time (3.64) captures the linear growth of randomness with the circuit time in $\mathcal{E}_\Psi^t$, up to logarithmic factors, derived semiclassically in this subsection.

## 3.3 Linear growth from the double trumpet

As in section 2.3, we will be more explicit about the semiclassical growth of randomness for near-extremal ER caterpillars. When $\beta \to 0$, the frame potential $G_1(t, t) = F_1(t) \propto \text{Tr} \exp(-2tH_{\text{eff},1})$ reduces to the thermal partition function for two coupled SYK systems with Hamiltonian $H_{\text{eff},1}$ at inverse temperature $2t$. This same thermal partition function was analyzed by Maldacena and Qi [61] to study the first-order phase transition between the low-temperature gapped eternal traversable wormhole phase and the high-temperature disconnected phase of two coupled SYK systems.

The geometry of the relevant Euclidean wormhole, shown in Fig. 18, is the double trumpet

$$\mathrm{d}s^2 = \frac{\mathrm{d}\tau^2 + \mathrm{d}\sigma^2}{\sin^2 \sigma}, \qquad \tau \sim \tau + b. \tag{3.65}$$

where the conformal boundaries are located at $\sigma = 0, \pi$. The bottleneck length modulus $b$, as well as the renormalized geodesic length between the dynamical boundaries, is stabilized by the bi-local interaction of the large number of matter fields. Recall that in our case this interaction

arises from an effective average over Brownian couplings in the frame potential $F_1(t)$.[33]

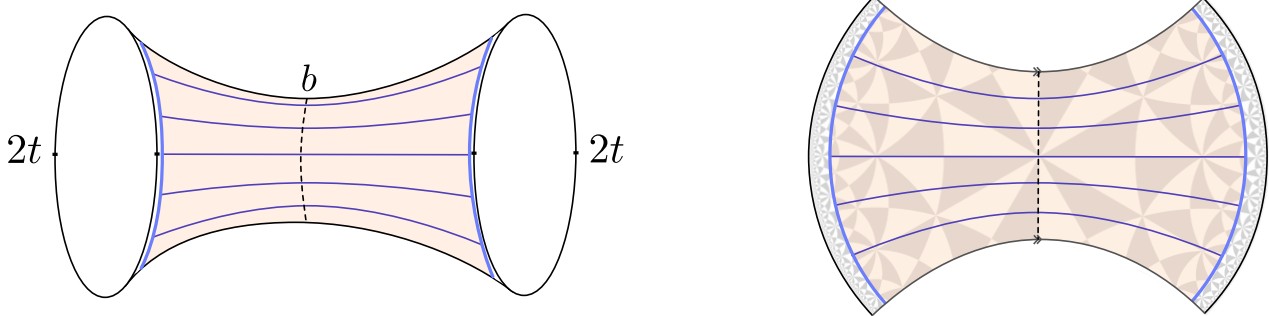

**Figure 18:** On the right, the fundamental domain of the subgroup of $\mathsf{PSL}(2,\mathbf{R})$ isometries which uniformizes the double trumpet in $\mathbf{H}^2$. In the Maldacena-Qi wormhole, the bottleneck modulus $b$ and the geodesic length between boundaries are static and stabilized by the interactions. For large enough $t$ there exist two saddle points with $b_1(t) < b_2(t)$. The "small wormhole" solution with $b_1(t)$ is thermodynamically unstable in the canonical ensemble [61] and thus it will not be considered here.

**Stabilized double trumpet.** The finite-temperature moment two-point function $G_1(t,t')$ given by (3.42) is a more sophisticated version of the thermal partition function of [61]. We will now construct the two-boundary wormhole contributing to this quantity.

The metric of the wormhole is that of the double trumpet (3.65). The two dynamical boundaries of the wormhole are again described by the gauge-fixed single particle Lagrangian (2.41), i.e., in terms of the renormalized length variable $\ell(u)$ between both boundaries. The relevant stabilized double-trumpet is described by a periodic trajectory of the $\ell$-particle, which contains two oscillations, as shown in Fig. 21.

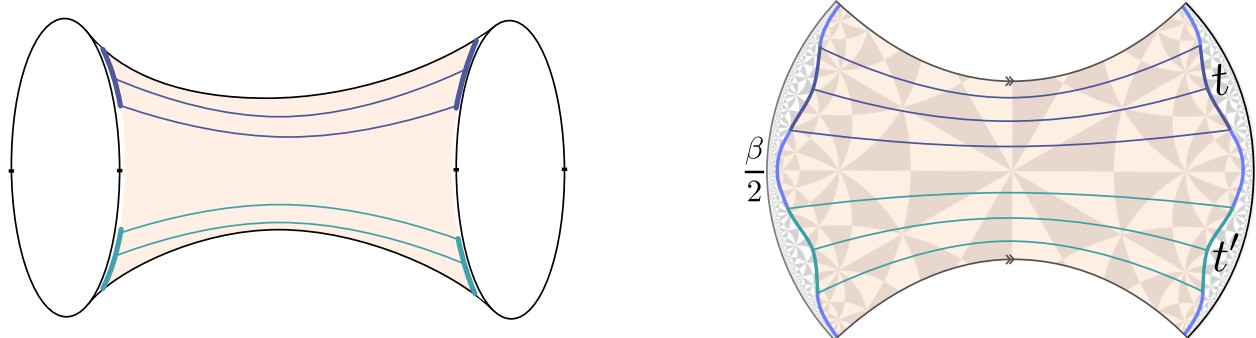

**Figure 19:** Two-boundary wormhole contribution to $G_1(t,t')$.

We now describe this solution more explicitly. As in section 2.3 we will use dimensionless quantities $\{\tilde{u}, \tilde{\beta}, \tilde{t}\}$ to construct the solution, and we consider $\Delta = 1/2$ to be fully explicit. We will pick a reflection-symmetric time coordinate $\tilde{u} \in [-\frac{\tilde{\beta}}{4} - \frac{\tilde{t}}{2}, \frac{\tilde{\beta}}{4} + \frac{\tilde{t}}{2}]$ for the $\ell$-particle, reflecting

---

[33] Notice that the wormhole does not violate factorization because the frame potential $F_1(t)$ does not factorize. This is different to Euclidean wormholes contributing to other quantities which should factorize, and whose presence is attributed to an effective underlying disorder in the microstructure of black holes (see e.g. [92, 121–126]).

one of the two oscillations of the periodic trajectory. The time-dependent interaction is turned on at the same values of the $\tilde{u}$ parameter as in (2.47). In region I ($\frac{\tilde{t}}{2} < |\tilde{u}| < \frac{\tilde{t}}{2} + \frac{\tilde{\beta}}{4}$) the interaction is turned off, and in region II ($|\tilde{u}| < \frac{\tilde{t}}{2}$) the interaction is turned on.

The solution on each of the regions is

$$\ell_I(\tilde{u}) = 2\log\cos\left((\tilde{u} - \tilde{u}_0)\sqrt{\frac{E}{2}}\right) - \log\frac{E}{2}, \tag{3.66}$$

$$\ell_{II}(\tilde{u}) = 2\log\left(-\frac{\sqrt{\eta^2 + 8E'}}{2E'}\cos\left(\tilde{u}\sqrt{\frac{E'}{2}}\right) - \frac{\eta}{2E'}\right). \tag{3.67}$$

The trajectory and its derivative must be continuous across $\tilde{u} = \frac{\tilde{t}}{2}$,

$$\ell_I\left(\frac{\tilde{t}}{2}\right) = \ell_{II}\left(\frac{\tilde{t}}{2}\right), \qquad \dot{\ell}_I\left(\frac{\tilde{t}}{2}\right) = \dot{\ell}_{II}\left(\frac{\tilde{t}}{2}\right). \tag{3.68}$$

The trajectory must be periodic, and this fixes

$$\dot{\ell}_I\left(\frac{\tilde{t}}{2} + \frac{\tilde{\beta}}{4}\right) = 0 \implies \tilde{u}_0 = \frac{\tilde{t}}{2} + \frac{\tilde{\beta}}{4}. \tag{3.69}$$

The energies $E$ and $E'$ are determined from the two conditions (3.68). The solution is represented in Fig.20.

Using these conditions, the jump in energy is determined to be given by

$$E' = E - \frac{\eta\sqrt{\frac{E}{2}}}{\cos\left(\frac{\tilde{\beta}}{4}\sqrt{\frac{E}{2}}\right)}. \tag{3.70}$$

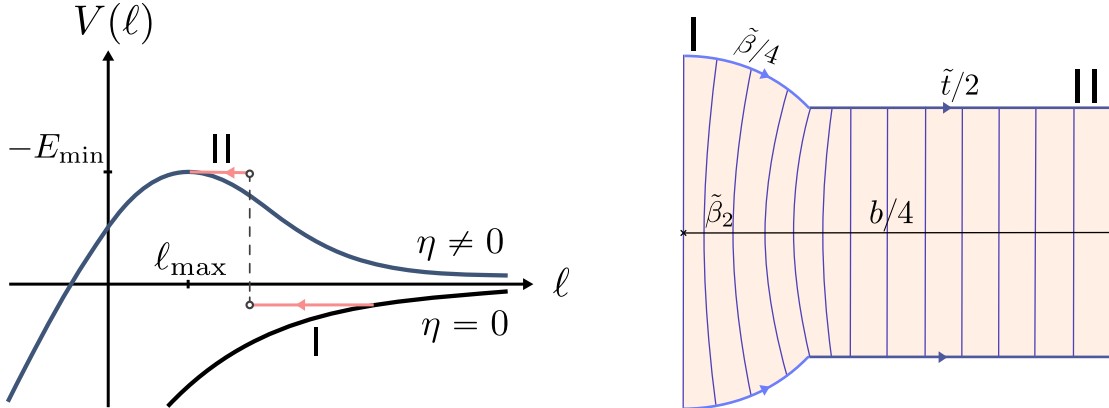

**Figure 20:** Half of an oscillation of the $\ell$-particle. The wormhole corresponds to two oscillations.

At late times $t \gg t_\star$, the particle spends a long time in region II, and thus its energy must be close to the equilibrium point, $E' \approx E_{\min} = -\eta^2/8$. Moreover, recall that $\tilde{\beta} = 8\pi/\eta$. Using

$$E = \frac{\eta^2}{8} x \,, \qquad x \approx 0.2076 \,. \tag{3.71}$$

Thus, the solution in region I is a Euclidean black hole at inverse temperature

$$\tilde{\beta}_2 = \frac{\tilde{\beta}}{\sqrt{x}} \approx 2.2 \, \tilde{\beta} \,. \tag{3.72}$$

For general $0 < \Delta < 1$, the jump in the energy is

$$E' = E - \frac{\eta \left(\frac{E}{2}\right)^{\Delta}}{\cos\left(\frac{\tilde{\beta}}{4}\sqrt{\frac{E}{2}}\right)^{2\Delta}} \,, \tag{3.73}$$

where we recall that $\tilde{\beta} = 2\pi(\eta\Delta/2)^{\frac{1}{2(\Delta-1)}}$ and $E_{\min} = -2\frac{1-\Delta}{\Delta}(\eta\Delta/2)^{\frac{1}{1-\Delta}}$. Imposing that $E' \approx E_{\min}$ for $t, t' \gg t_\star$, this allows us to solve for $\tilde{\beta}/\tilde{\beta}_2$. We plot the numerical solution in Fig. 21. The value of $\ell = \ell_0$ at which the particle jumps from region I to region II is $\ell_0 = \ell_{\max} + 2\log\cos\left(\frac{\pi\tilde{\beta}}{2\tilde{\beta}_2}\right) - 2\log\frac{\tilde{\beta}}{\tilde{\beta}_2}$.

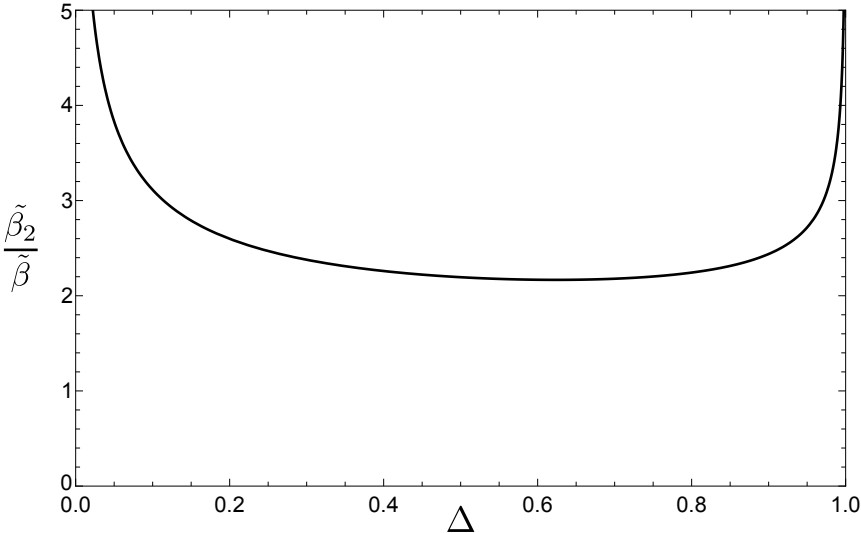

**Figure 21:** Inverse temperature $\tilde{\beta}_2$ of the Euclidean black hole solution on the two-boundary wormhole as a function of the conformal dimension $\Delta$ of the perturbation.

The bottleneck modulus $b$ is determined by the solution. Notice that $b/4$ corresponds to the global time of half an oscillation shown in Fig. 20. Using (2.59) for region I and the approximately constant value of $\ell$ in (2.40) for region II gives

$$\frac{b(t,t')}{\ell_{\Delta_0}} = E_{\text{gap}}(t + t') + 8\,\Delta_0 \operatorname{arctanh}\left(\tan\left(\frac{\pi\tilde{\beta}}{4\tilde{\beta}_2}\right)\right) \,. \tag{3.74}$$

where $E_{\text{gap}}$ and $\ell_{\Delta_0}$ are given by (2.64) and (2.66), while $\Delta_0$ is the lightest excitation on the wormhole.

**Contribution from the wormhole.** Evaluating (2.41), the on-shell action in regions I and II of the wormhole gives

$$\frac{I_{\mathsf{I}}}{N} = -\frac{4\pi^2\tilde{\beta}}{\tilde{\beta}_2^2} + \frac{16\pi}{\tilde{\beta}_2} \tan\left(\frac{\pi\tilde{\beta}}{2\tilde{\beta}_2}\right). \tag{3.75}$$

$$\frac{I_{\mathsf{II}}}{N} = \frac{E_{\min}}{2}\left(\tilde{t} + \tilde{t}'\right) + I_\Delta. \tag{3.76}$$

Thence, the total action of the wormhole is $I_{\mathrm{wh}} = I_{\mathsf{I}} + I_{\mathsf{II}}$, where we have omitted the constant parts of the action (2.41), since these will cancel with the normalization of the states. Here $I_\Delta$ is defined as

$$I_\Delta = 2\int_{\ell_{\max}}^{\ell_0} \sqrt{2\left(-E_{\min} - V(\ell)\right)}. \tag{3.77}$$

Moreover, the one-loop fluctuations around the wormhole at large $t, t'$ come from the lightest excitations, which in this case correspond to the $N$ single fermions,

$$Z_{\text{1-loop}} \approx 1 + Ne^{-E_{\mathrm{gap}}(t+t')}. \tag{3.78}$$

On the other hand, the action from the average norm (2.44) from the saddle point of section 2.3 gives

$$\log Z(t) = -\frac{E_{\min}}{2}t. \tag{3.79}$$

Altogether, from (3.42), we have that the wormhole contributes

$$G_1(t,t') \sim Z_{\mathrm{wh}}(t,t') = \frac{e^{-I_{\mathrm{wh}}}Z_{\text{1-loop}}}{Z(t)Z(t')Z(\beta)^2} \approx \frac{1}{L_{\mathrm{eff}}^2}\left(1 + Ne^{-E_{\mathrm{gap}}(t+t')}\right). \tag{3.80}$$

where the effective dimension is in this case

$$\frac{1}{N}\log L_{\mathrm{eff}}^2 = I_\Delta + \frac{4\pi^2}{\tilde{\beta}}\left(1 - \frac{\tilde{\beta}^2}{\tilde{\beta}_2^2}\right) + \frac{16\pi}{\tilde{\beta}_2} \tan\left(\frac{\pi\tilde{\beta}}{2\tilde{\beta}_2}\right) > 0. \tag{3.81}$$

Although we shall not study it here, the exchange in dominance between the disconnected and the wormhole saddle point contributions as a function of the circuit times $t$ and $t'$ is interpreted as a Maldacena-Qi type of first order thermal phase transition. For large enough $t, t'$ the wormhole contribution dominates the moment two-point function. The slow decay of the wormhole can be interpreted from the fact that this solution does not have a classical entropy at large-$N$.[34]

**Fermion parity.** In the case of the ER caterpillars in the SYK model, there is an additional subtlety to be considered: fermion parity $(-1)^F$ is conserved by the random circuit. In fact, this is only true if the perturbing operators $\{\mathcal{O}_\alpha\}$ are chosen to be even Majorana strings, an implicit assumption which we made in section 2.3. It would be interesting to generalize the considerations of this paper for fermionic perturbations, but we shall not do that here.

---

[34] Moreover, we are ignoring effects associated to particle production in the bulk [61]. These effects are expected to be small for sufficiently small coupling $\eta$, if the operator that drives the random circuit is relevant, $\Delta < 1/2$.

What this means physically is that the ensemble of ER caterpillars $\mathcal{E}_t^\Psi$ will never become random in the total Hilbert space (see [76]). Instead, it will become random on each superselection sector of fermion parity $(-1)^F$.[35] For $k = 1$, the number of invariant states, or ground states of $H_{\text{eff},1}$, increases to 2, corresponding to the ground states of $H_{\text{eff},1}$ within each fixed fermion parity sector. At infinite temperature, the ground space is generated by to the two possible TFDs

$$\left( \psi_i^1 \pm \mathrm{i}\,\psi_i^{\bar{1}} \right) |I, \pm\rangle_{1\bar{1}} = 0 \qquad \forall\, i = 1, ..., N\,. \tag{3.82}$$

In appendix D, we describe the contribution of these states to the thermal partition function in the large-$N$ limit of the model. At finite temperature, we expect a similar structure, where there will be two ground states $|\text{GS}, \pm\rangle$ of $H_{\text{eff},1}$. In the gravitational description, the ground states are described by a connected two-boundary wormhole with a discrete $\mathbf{Z}_2$-gauge field in the bulk. The two possible holonomies associated to this gauge field correspond to the values of $(-1)^F$.

Similarly, for $k > 1$, the number of ground states, and correspondingly the number of first excited states, increases by a factor of $2^k$. Plugging (3.80) in (3.38), with the additional factors of 2 from fermion parity, gives a distance to $k$-design,

$$\Delta_{k,2}(\mathcal{E}_\Psi^t)^2 \approx 2^k N_* \, k! \, k \, e^{-2tE_{\text{gap}}}\,, \tag{3.83}$$

and, thus, a time to approximate $k$-design

$$t_{k,2} \approx \frac{1}{2} E_{\text{gap}}^{-1} \left( k \log(2k) - k + \log k + \log N + 2 \log \varepsilon^{-1} \right)\,. \tag{3.84}$$

### 3.4 The random state ensemble

In section 3.2 we have used the infinite time ensemble $\mathcal{E}_\Psi^\infty$ as the reference "random" ensemble for our definition of an approximate quantum state $k$-design. Here, we provide a polynomial-copy equivalent and more explicit definition of the random state ensemble, $\mathcal{E}_\Psi^{\text{rand}}$, in terms of an ensemble of random purifications of the equilibrium state $\rho_{\text{eq}}$.

The ensemble $\mathcal{E}_\Psi^{\text{rand}}$ consists of states of the form

$$|\widetilde{\Psi}_{\text{rand}}\rangle = \sqrt{L_{\text{UV}}} \sum_{i,j} \left( \sqrt{\rho_{\text{eq}}}\, U \sqrt{\rho_{\text{eq}}} \right)_{ij} |E_i^*\rangle_{\text{L}} \otimes |E_j\rangle_{\text{R}}\,, \tag{3.85}$$

where $U$ is a Haar random unitary. Since the Hilbert space of a holographic CFT is infinite dimensional, one must truncate the spectrum at $E_i \leqslant E_{\text{UV}}$ and consider $U$ to be a Haar random unitary in $\mathsf{U}(L_{\text{UV}})$ where $L_{\text{UV}}$ is the finite dimension of the resulting Hilbert space. As long as the regulator $L_{\text{UV}}$ is sufficiently large, physical quantities of the ensemble will not depend on it, and we will take $L_{\text{UV}} \to \infty$ at the end of every computation in this section. The entries $\sqrt{L_{\text{UV}}} U_{ij}$ have $O(1)$ magnitude. The smooth envelope of the wavefunction of the states (3.85) is controlled by the factors of the equilibrium state $\sqrt{\rho_{\text{eq}}}$.

---

[35] Additional discrete symmetries of the model that arise depending on the value of $N$ (mod 8) are also omitted by choosing a suitable value of $N$ without degeneracies [76, 112].

The ensemble of states (3.85) is unit normalized on average,

$$\mathbb{E}_{\mathcal{E}_{\tilde{\Psi}}^{\mathrm{rand}}}[\langle \widetilde{\Psi}_{\mathrm{rand}} | \widetilde{\Psi}_{\mathrm{rand}} \rangle] = 1 \,. \tag{3.86}$$

The variance of the norm over different realizations of the state is exponentially suppressed in the second Rényi entropy of the equilibrium state,[36]

$$\mathrm{Var}_{\mathcal{E}_{\tilde{\Psi}}^{\mathrm{rand}}}[\langle \widetilde{\Psi}_{\mathrm{rand}} | \widetilde{\Psi}_{\mathrm{rand}} \rangle] = e^{-2S_2(\rho_{\mathrm{eq}})} \,. \tag{3.87}$$

As before, we here assume that $\rho_{\mathrm{eq}}$ has extensive second Rényi entropy $S_2(\rho_{\mathrm{eq}}) \to \infty$ in the thermodynamic limit. Therefore, since the variance of the norm is exponentially suppressed, it is safe to compute things directly for the ensemble (3.85) of states normalized on average.

The states (3.85) represent random purifications of the equilibrium density matrix $\rho_{\mathrm{eq}}$. The reduced states are on average

$$\mathbb{E}_{\mathcal{E}_{\tilde{\Psi}}^{\mathrm{rand}}}[\rho_{\mathsf{L}}] = \mathbb{E}_{\mathcal{E}_{\tilde{\Psi}}^{\mathrm{rand}}}[\rho_{\mathsf{R}}] = \rho_{\mathrm{eq}} \,. \tag{3.88}$$

The 2-norm distance to the equilibrium state is, on average over different realizations of the state, exponentially suppressed,

$$\mathbb{E}_{\mathcal{E}_{\tilde{\Psi}}^{\mathrm{rand}}}\left[ \left\| \rho_{\mathsf{L}} - \rho_{\mathrm{eq}} \right\|_2^2 \right] = \mathbb{E}_{\mathcal{E}_{\tilde{\Psi}}^{\mathrm{rand}}}\left[ \left\| \rho_{\mathsf{R}} - \rho_{\mathrm{eq}} \right\|_2^2 \right] = e^{-S_2(\rho_{\mathrm{eq}})} \,. \tag{3.89}$$

However, as discussed around (2.88), this doesn't imply that each instance is close to the mean.

We will now compute the frame potentials of the random state ensemble $\mathcal{E}_{\Psi}^{\mathrm{rand}}$,

$$F_k^{\mathrm{rand}} = L_{\mathrm{UV}}^{2k} \, \mathbb{E}_{\mathcal{E}_U^{\mathrm{Haar}}} \left| \mathrm{Tr}\left( U \rho_{\mathrm{eq}} V^\dagger \rho_{\mathrm{eq}} \right) \right|^{2k} \,, \tag{3.90}$$

Using the Haar moment superoperator $\hat{\Phi}_{\mathrm{Haar}}^{(k)}$ defined in (3.13) and the replica equilibrium density matrix $\rho_{\mathrm{eq},2k} = \rho_{\mathrm{eq}}^{\otimes 2k}$ the frame potentials read

$$F_k^{\mathrm{rand}} = L_{\mathrm{UV}}^{2k} \, \mathrm{Tr}\left( \rho_{\mathrm{eq},2k}, \hat{\Phi}_{\mathrm{Haar}}^{(k)} \, \rho_{\mathrm{eq},2k} \, \hat{\Phi}_{\mathrm{Haar}}^{(k)} \right) \,. \tag{3.91}$$

In order to evaluate this quantity, we will use the identity

$$\left( \sqrt{\rho_{\mathrm{eq}}} \otimes \sqrt{\rho_{\mathrm{eq}}} \right) |I\rangle = \frac{1}{\sqrt{L_{\mathrm{UV}}}} \, |\rho_{\mathrm{eq}}\rangle \,, \tag{3.92}$$

and the related identity for the invariant state $|W_\sigma\rangle$,

$$\sqrt{\rho_{\mathrm{eq},2k}} \, |W_\sigma\rangle = L_{\mathrm{UV}}^{-k} \, |\rho_{\mathrm{eq}}, \sigma\rangle \,, \qquad \sigma \in \mathrm{Sym}(k) \,, \tag{3.93}$$

where $|\rho_{\mathrm{eq}}, \sigma\rangle$ is the replica equilibrium state associated to the permutation $\sigma$, as defined in

---

[36] To compute averages and variances, we have used the first and second moment superoperators of the Haar ensemble, presented in appendix E. The formula for the variance is a simplification obtained from a Gaussian approximation of the Haar matrix elements; it becomes exact as $L_{\mathrm{UV}} \to \infty$ at fixed $S_2(\rho_{\mathrm{eq}})$. In the opposite limit, $S_2(\rho_{\mathrm{eq}}) \to \ln L_{\mathrm{UV}}$, the variance will vanish as the states are automatically normalized.

(3.45).

In (3.91), the Haar moment superoperators $\hat{\Phi}_{\text{Haar}}^{(k)}$ are orthogonal projectors to the invariant subspace, generated by the invariant states $|W_\sigma\rangle$. Using the explicit form of the projector (E.5), together with (3.93), leads to

$$F_k^{\text{rand}} = \sum_{\sigma,\tau,\omega,\kappa\in\text{Sym}(k)} c_{\sigma,\omega} c_{\kappa,\tau} \langle\rho_{\text{eq}},\sigma|\rho_{\text{eq}},\tau\rangle\langle\rho_{\text{eq}},\kappa|\rho_{\text{eq}},\omega\rangle, \tag{3.94}$$

where $c_{\sigma,\tau}$ is the Weingarten matrix of coefficients that has been introduced in appendix E. Therefore, up to corrections of order $k/L_{\text{eff}}$ or higher Rényi entropies of the equilibrium state $\rho_{\text{eq}}$, the frame potential is dominated by the diagonal term in the sum (3.93) with $\sigma = \tau = \omega = \kappa$ and thus,

$$F_k^{\text{rand}} \approx \frac{k!}{L_{\text{eff}}^{2k}}, \qquad L_{\text{eff}} = \left(\text{Tr}\rho_{\text{eq}}^2\right)^{-1}. \tag{3.95}$$

The frame potentials coincide with $F_k(\infty)$ given by (3.51) for the ensemble $\mathcal{E}_\Psi^\infty$. Likewise, it is straightforward to see that the 2-norm distance between the $k$-th moment states is $O(k/L_{\text{eff}})$, and thus both ensembles are equivalent for $k \ll L_{\text{eff}}$ at the level of the moments.

## 3.5   Randomness/length relation

The main result of this paper is implied by Eqs. (2.26) and (3.64). Neglecting subleading logarithmic corrections in (3.64), the implication is that the ensemble of ER caterpillars of average length $\ell$ and characteristic matter correlation scale $\ell_\Delta$ forms an $\varepsilon$-approximate quantum state $k$-design of the black hole for

$$k \approx 2\,\frac{\ell - \ell_\varepsilon}{\ell_\Delta}, \tag{3.96}$$

where $\ell_\varepsilon = \ell_\Delta \log \varepsilon^{-1}$. Thus, in this precise sense, the geometry of the ER caterpillar behaves as a *bulk* random quantum circuit in the black hole interior, where $\ell$ plays the role of the total elapsed time and $\ell_\Delta$ plays the role of the circuit time.

# 4   Towards random states at exponentially long circuit times

The derivation of the relation (3.96) assumes that $k \ll L_{\text{eff}}$, i.e., that the length of the wormhole is sub-exponential in the second Rényi entropy $S_2(\rho_{\text{eq}}) = \log L_{\text{eff}}$ of the equilibrium state. An obvious question is what happens when $k \gtrsim L_{\text{eff}}$. In this case, we expect that the ensemble of exponentially long ER caterpillars is polynomial-copy indistinguishable, with $\varepsilon$ precision, from the ensemble of random states of the two-sided black hole.

In this section, we make some remarks about the microscopic growth of randomness for $k \gtrsim L_{\text{eff}}$. For simplicity, we will restrict to the microcanonical study of section 3.1, where $L_{\text{eff}} = L = e^S$ is the dimension of the microcanonical window of the black hole. Recall that we

aim to compare the ensemble of microcanonical ER caterpillars and the random state ensemble,

$$|\Psi_M(t)\rangle = \frac{1}{\sqrt{L}} \sum_{i,j=1}^{L} U(t)_{ij} |E_i^*\rangle_{\mathsf{L}} \otimes |E_j\rangle_{\mathsf{R}} \in \mathcal{E}_{\Psi_M}^t \,, \tag{4.1}$$

$$|\Psi_M^{\mathrm{rand}}\rangle = \frac{1}{\sqrt{L}} \sum_{i,j=1}^{L} U_{ij} |E_i^*\rangle_{\mathsf{L}} \otimes |E_j\rangle_{\mathsf{R}} \in \mathcal{E}_{\Psi_M}^{\mathrm{rand}} \,. \tag{4.2}$$

In section 3.1, we found that the ensemble $\mathcal{E}_{\Psi_M}^t$ becomes an approximate quantum state $k$-design, in a circuit time $t_{k,2}$ scaling approximately linearly with $k$. For a more precise expression, see (3.28). In this analysis, we have restricted to values $k \leqslant L$ in (3.23). We now extend this study for $k > L$ and provide some microscopic arguments in the microcanonical window of the black hole in favor of a persisting linear growth of randomness for $k \gg L$. We contrast this seemingly ever-lasting linear growth with the saturation of randomness for any physical observer with access to the states and finite resolution in Hilbert space.

The 2-norm distance between the moments of both ensembles is determined by (3.9), a relation which holds for any value of $k$. We will therefore analyze the frame potentials $F_k(t)$ and $F_k^{\mathrm{rand}}$ defined in (3.7) and (3.8), for any value of $k$.

**Frame potentials of the random state ensemble.** Consider the $k$-th frame potential of the random state ensemble, $F_k^{\mathrm{rand}}$, for arbitrary large values of $k$. Recall that this quantity is determined by the $k$-th moment of the Haar ensemble, $f_{k,L}$,

$$F_k^{\mathrm{rand}} = \frac{f_{k,L}}{L^{2k}} \,, \qquad f_{k,L} \equiv \mathbb{E}_{\mathcal{E}_{U_M}^{\mathrm{Haar}}} |\mathrm{Tr}\, U|^{2k} \,. \tag{4.3}$$

In section 3.1 we have used the fact that [117–119]

$$f_{k,L} = k! \,, \qquad \text{for } k \leqslant L \,. \tag{4.4}$$

The generalization to $k > L$ requires noticing that $f_{k,L}$ has a combinatorial origin. Given a permutation $\pi \in \mathrm{Sym}(k)$, an *increasing subsequence of $\pi$* is a sequence $i_1 < i_2 < \cdots < i_n$ such that $\pi(i_1) < \pi(i_2) < \cdots < \pi(i_n)$. The value of $f_{k,L}$ is the number of permutations $\pi \in \mathrm{Sym}(k)$ such that $\pi$ has no increasing subsequence of length greater than $L$ [119].

From the perspective of section 3.1, $f_{k,L}$ corresponds to the dimension of the invariant subspace on the full $2k$ replica Hilbert space, $f_{k,L} = \mathrm{Tr}\, \hat{\Phi}_{\mathrm{Haar}}^{(k)}$. The invariant subspace is linearly generated by the states $|W_\sigma\rangle$ of the form (3.14). The invariant states are not orthogonal and, in particular, the overlap depends on combinatorial data, $\langle W_\tau | W_\sigma \rangle = L^{-\#\mathrm{cycles}(\sigma\tau^{-1})}$. The dimension of the invariant space is the rank of the Gram matrix of overlaps $G_{\tau,\sigma} = \langle W_\tau | W_\sigma \rangle$. For $k \leqslant L$, the rank is the total number of invariant states, and (4.4) follows. For $k > L$, the invariant states are not linearly independent and the rank is strictly smaller than $k!$, in accord with the combinatorial definition of $f_{k,L}$ above.

A closed-form expression for $f_{k,L}$ is not known for $k > L$. In appendix G we provide details on asymptotic formulas for the moments $f_{k,L}$ for different scalings of $k$ with $L$. The relevant

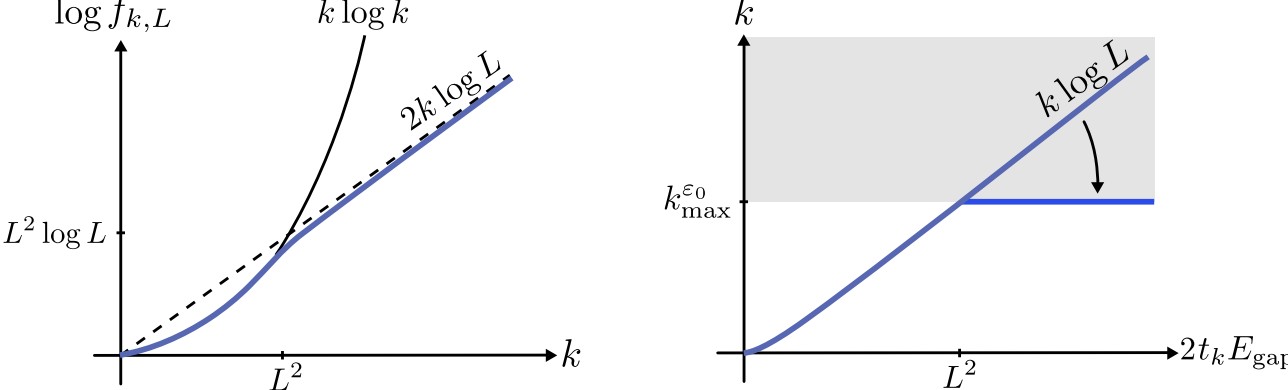

**Figure 22:** On the left, the log of the $k$-th moment $f_{k,L}$ for the Haar ensemble in $\mathsf{U}(L)$ as a function of $k$. On the right, the linear growth of randomness persists for exponential timescales. However, for any observer with finite resolution $\varepsilon_0$ in Hilbert space there is a maximum $k_{\max}^{\varepsilon_0}$ after which the ensemble of ER caterpillars becomes indistinguishable from the random state ensemble, and randomness effectively saturates. For physically reasonable observers, $\varepsilon_0 \gtrsim L^{-1}$, and $k_{\max}^{\varepsilon_0} \sim L^2$ scales with the two-sided microcanonical Hilbert space dimension.

scalings are

$$
f_{k,L} \sim \begin{cases} k! & \text{for} \quad k \lesssim L^2\,, \\ k!\,e^{-k\,H(x)} \sim L^{2k}e^{-k(1+H(x)+2\log x)} & \text{for} \quad k \sim \frac{L^2}{x^2} \quad \text{and} \quad \frac{1}{2} \lesssim x < 2\,, \\ L^{2k}k^{-\frac{L^2}{2}} & \text{for} \quad k \gtrsim L^2\,. \end{cases} \tag{4.5}
$$

where $H(x)$ is defined in (G.13). As shown in Fig. 22, there is a smooth transition between $k!$ and $L^{2k}$ scalings when $k \sim L^2$.

**Expected linear growth.** We shall now consider the frame potential $F_k(t)$ for the ensemble of states $\mathcal{E}_{\Psi_M}^t$,

$$
F_k(t) = \frac{f_{k,L}(t)}{L^{2k}}\,, \qquad f_{k,L}(t) \equiv \mathbb{E}_{\mathcal{E}_{U_M}^t}\,\big|\mathrm{Tr}\,U(2t)\big|^{2k}\,. \tag{4.6}
$$

From (3.21), $f_{k,L}(t)$ is determined by the thermal partition function, at inverse temperature $2t$, for the $2k$-replica effective Hamiltonian $H_{\mathrm{eff},k}$ defined in (3.20). Under genericity assumptions for the set of drive operators $\{\mathcal{O}_\alpha\}$, the invariant states (3.14) can be assumed to be the only states necessary to generate the ground space of the effective Hamiltonian $H_{\mathrm{eff},k}$. Their energy vanishes, $H_{\mathrm{eff},k}\,|W_\sigma\rangle = 0$. The Haar moment superoperator (3.13) is therefore $\hat{\Phi}_{\mathrm{Haar}}^{(k)} = \exp(-\infty H_{\mathrm{eff},k})$.

According to (4.5), for $k \gtrsim L^2$, the relative dimension of the ground space of $H_{\mathrm{eff},k}$ with respect to the total $2k$ replica dimension is suppressed by a factor of $k^{-L^2/2}$. Since there is a large room for excited states, it is therefore reasonable to assume that the same structure for the first excited states of $H_{\mathrm{eff},k}$ extends from $k \lesssim L$. Namely, the first excited states correspond to single replica excitations on top of the ground states. The difference is now that all of the invariant states are not linearly independent, and one must only consider a subset of them. If this structure extends, the gap of $H_{\mathrm{eff},k}$ given by $E_{\mathrm{gap}}$ will be independent of $k$. To be clear, up to numerics on small $L$ systems, we do not have stronger evidence for this structure. It would

be interesting to analyze these assumptions in more detail, although we will not do that here. The regime $k > L$ is completely out of control for semiclassical methods to compute the frame potential $F_k(t)$, like the ones used in sections 3.2 and 3.3 with the gravitational path integral.

Under these assumptions, the distance to design (3.10) implies the ensemble will become an $\varepsilon$-approximate quantum state $k$-design in a time

$$t_{k,2} \sim \frac{1}{2} E_{\text{gap}}^{-1} \begin{cases} k \log k - k + 2 \log \varepsilon^{-1} & \text{for} \quad k \lesssim L^2 \,, \\ k \log L + 2 \log \varepsilon^{-1} & \text{for} \quad k \gtrsim L^2 \,. \end{cases} \tag{4.7}$$

We illustrate both behaviors in Fig. 22. Therefore, we expect that randomness never saturates, and in fact keeps growing linearly for the continuous random circuits used in this paper.

**Finite resolution and the saturation of randomness.** It may seem counter-intuitive that randomness can grow forever, even if we have a resolution $\varepsilon$ in the definition of an approximate quantum state $k$-design. Notice that $\varepsilon$ is a resolution in 2-norm distance for the moment states of the ensembles, which live in the replicated Hilbert space. This is different from introducing a fixed resolution $\varepsilon_0$ in the original Hilbert space.

To see this, consider introducing a resolution $\varepsilon_0$ in trace distance in $\mathsf{U}(L)$, i.e. for $U, V \in \mathsf{U}(L)$, introduce the equivalence relation

$$U \sim V \;\Leftrightarrow\; \left\| U - V \right\|_1 < \varepsilon_0 \,. \tag{4.8}$$

With a finite resolution $\varepsilon_0$, the space of unitaries $\mathsf{U}(L)$ becomes a discrete set, and its size $|\mathsf{U}(L)|_{\varepsilon_0}$, i.e. the number of $\varepsilon_0$-balls that fit into $\mathsf{U}(L)$, can be estimated to scale as

$$|\mathsf{U}(L)|_{\varepsilon_0} \sim e^{-L^2 \log \varepsilon_0^{-1}} \,. \tag{4.9}$$

For $\varepsilon_0$ small but fixed, this number is doubly exponential in the entropy.

For a physical observer with finite resolution, the ensemble of random circuits $\mathcal{E}_{U_M}^t$ effectively corresponds to a discrete ensemble of unitaries, of size $\left| \mathcal{E}_{U_M}^t \right|_{\varepsilon_0}$. Likewise, the ensemble of microcanonical ER caterpillars $\mathcal{E}_\Psi^t$ becomes a discrete ensemble of states. In this case, the inverse of the size of the ensembles lower bounds the value of the frame potentials [68]

$$F_k(t) \equiv \frac{1}{L^{2k}} \mathbb{E}_{\mathcal{E}_{U_M}^t} \left| \text{Tr} \left( U(t) V(t)^\dagger \right) \right|^{2k} \geq \left| \mathcal{E}_{U_M}^t \right|_{\varepsilon_0}^{-1} \,, \tag{4.10}$$

where the inequality follows from taking $U(t) = V(t)$ in the expectation value.

This bound implies that for $\mathcal{E}_{\Psi_M}^t$ to be an approximate discrete quantum state $k$-design we must have $F_k(t) \approx F_k^{\text{rand}} = f_{k,L}/L^{2k}$ and hence its size must be lower bounded by

$$\frac{L^{2k}}{f_{k,L}} \lesssim \left| \mathcal{E}_{U_M}^t \right|_{\varepsilon_0} \,. \tag{4.11}$$

Moreover, it must obviously be true that $\left| \mathcal{E}_{U_M}^t \right|_{\varepsilon_0} \leq |\mathsf{U}(L)|_{\varepsilon_0}$. This condition is incompatible with (4.11) at sufficiently large $k$. Using the scaling of the moments $f_{k,L}$ in (4.5), and assuming that the observer has an exponential resolution in the entropy, $\varepsilon_0^{-1} \sim L$ (which could be too

optimistic), the lower bound implies that there cannot exist discrete approximate unitary $k$-designs for $k \geqslant k_{\mathrm{max}}^{\varepsilon_0}$ for

$$k_{\mathrm{max}}^{\varepsilon_0} \sim \varepsilon_0^{-2} \sim L^2 \,. \tag{4.12}$$

Thus, for any realistic purpose, randomness saturates at the value $k_{\mathrm{max}}^{\varepsilon_0}$, as shown in Fig. 22. We remark that $L^2 = e^{2S}$ corresponds to the dimension of the microcanonical window of the two-sided black hole; hence, the required inverse of the resolution is $\varepsilon_0^{-1} \gtrsim e^S$, i.e., the square root of the dimension. For a one-sided situation we would expect $\varepsilon_0^{-1} \gtrsim e^{S/2}$.

**No recurrences for randomness.** Poincaré recurrences are not expected for the behavior of randomness, as a consequence of the Markovian nature of the ensemble of random circuits. The frame potentials $F_k(t)$ are non-increasing functions of time $t$,

$$\frac{\mathrm{d}}{\mathrm{d}t} \log F_k(t) = -\langle H_{\mathrm{eff},k} \rangle_{2t} \leqslant 0 \,, \tag{4.13}$$

where $\langle H_{\mathrm{eff},k} \rangle_{2t} = \mathrm{Tr}(\rho_{2t} H_{\mathrm{eff},k})$ is the average energy of the effective Hamiltonian in the canonical ensemble $\rho_{2t} \propto \exp(-2t H_{\mathrm{eff},k})$. The non-positivity of (4.13) follows from the positive semi-definiteness of $H_{\mathrm{eff},k}$. Physically, the Poincaré recurrences of individual random circuits are not self-averaging and these effects average out to zero over the ensemble.

# 5   Discussion

In this paper we have constructed an ensemble of ER caterpillars of a two-sided black hole using a gradually-cooled random quantum circuit in their preparation and found that the length of the wormhole for the states parametrizes the amount of randomness defining their microscopic wavefunctions. More precisely, we have found that:

---

*The ensemble of ER caterpillars of average length $\ell$ and characteristic matter correlation scale $\ell_\Delta$ forms an $\varepsilon$-approximate quantum state $k$-design of the black hole for*

$$k \approx 2 \frac{\ell - \ell_\varepsilon}{\ell_\Delta} \,, \tag{5.1}$$

*where $\ell_\varepsilon = \ell_\Delta \log \varepsilon^{-1}$.*

---

In this quantitative sense the geometry of the ER caterpillars behaves as a bulk random quantum circuit, where $\ell$ plays the role of the total time and $\ell_\Delta$ plays the role of the circuit time. We now proceed to discuss some consequences of Eq. (5.1) and interesting future research directions.

## Holographic circuit complexity and volume

Eq. (5.1) has potential implications for the proposed relation between quantum circuit complexity and the volume of the ER bridge, based on the bounds on the unitary circuit complexity of any element of a $k$-design [68, 127]. At face value, these bounds imply that the unitary circuit

complexity to prepare a state in $\mathcal{E}_\Psi^{t_k}$ grows at least linearly with $k$ and with the entropy $S$. Roughly, using these bounds in our setting, (5.1) implies a relation of the form

$$\mathcal{C}(\mathcal{E}_\Psi^{t_k}) \;\gtrsim\; kS \approx \frac{\mathrm{Vol} - \mathrm{Vol}_\varepsilon}{2G\ell_\Delta}\,, \tag{5.2}$$

given that the geometry of the ER caterpillars presented in this paper is cylindrical, i.e., $\mathrm{Vol} = 4GS\ell$ and $\mathrm{Vol}_\varepsilon = 4GS\ell_\varepsilon$. Eq. (5.2) has the form of the Complexity=Volume conjecture, derived from semiclassical considerations in this paper.

However, there are reasons to expect that the bound (5.2) is not tight. The states of $\mathcal{E}_\Psi^t$ all have parts of Euclidean evolution in their wavefunction as a consequence of implementing the gradual cooling process to the random circuit, and this is a non-unitary operation.

As a toy model of the ER caterpillars, one can consider a cylindrical tensor network, of depth $\ell/\ell_\Delta$, composed of $O(S)$ non-unitary random few-body gates per layer. Such a circuit is essentially a holographic random tensor network model of the interior [128]. Due to the non-unitary gates, it is not clear whether the unitary circuit complexity of such a tensor network scales linearly with $S$. It would be interesting to quantify this more precisely, to see if the volume of the wormhole guards the proposed relation with the unitary circuit complexity to prepare more general states of the black hole, or whether the volume simply provides a lower bound of the form (5.2).

**Holography and the black hole interior**

A quantum state $k$-design is $k$-copy indistinguishable from the ensemble $\mathcal{E}_\Psi^{\mathrm{rand}}$ of random states of the black hole. Eq. (5.1) implies that the random state ensemble is $k$-copy indistinguishable within $\varepsilon$-precision from the ensemble of ER caterpillars of length $k\ell_\Delta + \ell_\varepsilon$. In this paper we have presented a single way to build states with the random circuit. However, it is natural to expect that there are many other ways to do this, leading to approximate $k$-designs with macroscopically distinct interior geometries, which also would be $k$-copy indistinguishable from the ER caterpillars of this paper. In the specific setup of this paper, we could imagine choosing different types of matter perturbations to construct different approximate $k$-designs.

In this context, given two macroscopically distinct $k$-designs of the black hole, it would be interesting to elucidate what specific $(k+1)$-copy measure over the ensemble of states is able to distinguish them.

Conceptual problems arise when $k$ is allowed to scale exponentially with the entropy $S$ of the black hole. In this case we have argued that any observer with a resolution $\varepsilon_0 \gtrsim e^{-S/2}$ in the microcanonical Hilbert space of the black hole is unable to distinguish the ensemble of ER caterpillars from the ensemble of random states $\mathcal{E}_{\Psi_M}^{\mathrm{rand}}$ of the black hole. As a consequence, any sufficiently generic pure state of the black hole must have an overlap of $1 - O(e^{-S/2})$ with *a single* exponentially long ER caterpillar of the ensemble, at least when considering the wavefunctions within the microcanonical window of the black hole. This signals a clear limitation of the holographic description of the black hole interior: macroscopically distinct very long interiors appear quantum indistinguishable from the outside, up to a very small resolution.

A possibility to resolve this problem is to declare that the holographic description is what defines the interior. The consequence is that the low-energy EFT breaks down at macroscopic scales for very long wormholes and that all of the different interiors are physically $e^{-S/2}$-close (see e.g. [60, 129–133] contemplating this possibility). It would be good to have a way to decide whether this happens, or whether the holographic description of the interior is simply limited.

An important point which we have ignored in this discussion is that the ensembles of states that we have constructed live on infinite-dimensional Hilbert spaces. A priori, it is unclear whether different macroscopic interiors lead to the same notion of a quantum state $k$-design, given that this notion refers to a specific random state ensemble $\mathcal{E}_\Psi^{\text{rand}}$, consisting of the random purifications of the equilibrium density matrix $\rho_{\text{eq}}$. For two different ensembles of states with macroscopically distinct interiors, we expect to have two different equilibrium states $\rho_{\text{eq}}$ and $\rho'_{\text{eq}}$. The equilibrium states can be far from each other in trace distance, even if both of them are ensemble-equivalent to the same canonical Gibbs state in the thermodynamic limit. Thus, in principle, there is still the possibility that there is always a way of distinguishing features of the interior, but this requires performing experiments that access the high-energy tails of the wavefunctions of the states.

Finally, if the semiclassical description of the ensemble of states is to be trusted for very long wormholes, the states do not contain firewalls close to the horizon. In fact, the semiclassical interiors of the states are as different from a firewall as they can be: they contain very large interiors. Eq. (5.1) implies, however, that distinguishing the ensemble from the typical state ensemble is very hard. It would be interesting to understand how this is compatible with the old expectation that there exist firewalls in typical states of the black hole [131, 134–137].

**Semiclassical disorder and Euclidean gravity**

In this paper, we have introduced explicit disorder in the semiclassical description. Specifically, we have considered an ensemble of bulk states containing ER caterpillars with different semiclassical details. The ensemble of states is prepared in the complexified past via disordered boundary conditions for the bulk fields. We have seen that the Brownian correlations of the boundary conditions generate time-independent effective interactions between replicas which support Euclidean wormholes for average quantities over the ensemble, such as the frame potentials. In our case there is no factorization problem, because the quantities to which these wormholes contribute do not factorize.

This is to be contrasted with the modern perspective on the nature of Euclidean wormhole contributions to the gravitational path integral. These are understood to arise from a "semiclassically invisible" disorder over the pseudo-random microscopic features of black holes (see e.g. [92, 121–126]). With this interpretation, CFT random tensor networks were constructed in [138], and these serve to model the pseudo-random microscopics of individual wavefunctions. It would be interesting to see whether there is any connection between both notions of disorder, wormholes, and random circuits.

## Acknowledgments

We thank Stefano Antonini, Vijay Balasubramanian, José Barbón, Pawel Caputa, Horacio Casini, Gong Cheng, Jordan Cotler, Roberto Emparan, Albion Lawrence, Alex May, Dmitry Melnikov, Rob Myers, Matt Headrick, Tim Schuhmann, Steve Shenker, Alejandro Vilar-López, Zixia Wei and Beni Yoshida for discussions. This work was performed in part at the Aspen Center for Physics, which is supported by the National Science Foundation grant PHY-2210452, and by the Simons Foundation (1161654, Troyer). MS and BS acknowledge support from the U.S. Department of Energy through DE-SC0009986 and QuantISED DE-SC0020360. The work of JM is supported by CONICET, Argentina. J.M acknowledges hospitality and support from the International Institute of Physics, Natal, through the Simons Foundation award number 1023171-RC. Brandeis code: BR-TH-6722.

## A  Backrection of the exterior random circuit

In this appendix we consider the states (2.1). On average over different realizations of the random circuit, the energy of the black hole will grow as $\mathbb{E}\left[E_{\mathsf{R}}(t)\right] \approx M + Jt\,\delta E$, at a rate given by the absorptive part of its response function

$$\delta E = \frac{K}{2\pi} \int_{-\infty}^{\infty} \mathrm{d}\omega\, \omega\, \mathrm{Im}\chi(\omega)\,. \tag{A.1}$$

In this expression we have introduced the standard linear response function in thermal physics

$$\chi_{\alpha\alpha'}(t - t') = -\mathrm{i}\theta(t - t')\mathrm{Tr}\left(\rho_\beta[\mathcal{O}_\alpha(t), \mathcal{O}_{\alpha'}(t')]\right)\,, \tag{A.2}$$

averaged over the perturbations $\chi(t) = K^{-1} \sum_\alpha \chi_{\alpha\alpha}(t)$ and $\chi(\omega) = \int_{-\infty}^{\infty} \mathrm{d}t\, \chi(t)\, e^{\mathrm{i}\omega t}$. In (A.2) the operators are evolved in the interaction picture, where $\mathcal{O}_\alpha(t) = \exp(-\mathrm{i}tH_0)\mathcal{O}_\alpha \exp(\mathrm{i}tH_0)$. The expression (A.1) does not contain a linear term in the perturbation given that the couplings have zero mean. The Brownian perturbation is white-noise correlated, hence $\delta E$ contains contributions from all of the frequencies of the response function.

By the fluctuation-dissipation theorem, (A.1) can also be expressed in terms of the average Wightman thermal two-point function $G_\beta(t) = K^{-1} \sum_\alpha \mathrm{Tr}(\rho_\beta \mathcal{O}_\alpha(t)\mathcal{O}_\alpha)$, namely as

$$\delta E = \frac{K}{2\pi} \int_{-\infty}^{\infty} \mathrm{d}\omega\, \omega\, e^{-\beta\omega}\, G_\beta(\omega) = -\mathrm{i} \left.\frac{\mathrm{d}}{\mathrm{d}t}G_\beta(t)\right|_{t=0^+}, \tag{A.3}$$

where, in the first form, the frequency-spectrum of the two-point function $G_\beta(\omega) = \int_{-\infty}^{\infty} \mathrm{d}t\, G_\beta(t)\, e^{\mathrm{i}\omega t}$ is weighted by thermal factors that arise from the KMS condition $G_\beta(-\omega) = e^{-\beta\omega}G_\beta(\omega)$.

We can consider choosing the perturbations $\mathcal{O}_\alpha$ of the random circuit to correspond to $K$ different modes $\{\Phi_{\omega_\alpha} : \alpha = 1, ..., K\}$ of a bulk scalar field, with frequencies $\omega_\alpha \approx \omega_0$. In this case, the dominant frequency in the response function will be $\omega_0$ and the increase in energy will be

$$\delta E \sim \omega_0 K\,. \tag{A.4}$$

The frequency $\omega_0$ can be chosen to be really small compared to $T_H$, given that the response function peaks at the discrete energy separations in the spectrum of the black hole. In the semiclassical description of the black hole this is a continuum of energies. However, in the microscopic description the energy separations are revealed at $O(T_H \exp(-S))$ energy scales. This indicates that one could have the quantum circuit act for circuit times $Jt$ which could be up to exponential in $S$. We show this class of sub-Rindler scale random circuits of the black hole in Fig. 23. It would be interesting to see if this class of random circuit can actually be implemented, or if there is any fundamental limitation to using these modes of the Hawking radiation, in particular of the modes with $\omega_0 \ll T_H/S$.

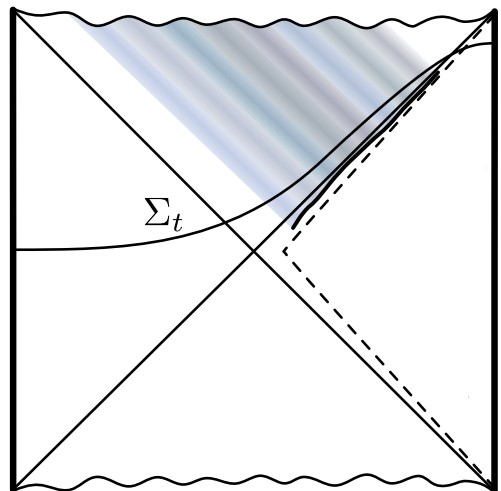

**Figure 23:** The operators $\mathcal{O}_\alpha$ insert perturbations within the Rindler region of the black hole, delimited by the dashed line. The matter has been dimmed to represent that it produces a small backreaction of the perturbations is very small .

Another possibility is to regularize the perturbations microcanonically to avoid a substantial backreaction for very long times. The large AdS black hole of ADM mass $M$ has an associated microcanonical band of the holographic CFT, of energies $E \in [M - \frac{\Delta M}{2}, M + \frac{\Delta M}{2}]$, with $\Delta M \sim T_H \ll M$. The microcanonical band defines a Hilbert subspace of dimension $L \sim e^S$. We could choose regularized perturbations $\mathcal{O}_\alpha = \oplus_M \mathcal{O}_\alpha^M$ with $\mathcal{O}_\alpha^M = \Pi_M \mathcal{O}_\alpha \Pi_M$ acting non-trivially within each microcanonical band, with the use of the orthogonal projector $\Pi_M$ to the band. These perturbations of $H_0$ are block-diagonal and so is the resulting time-evolution operator $U(t) = \oplus_M U_M(t)$. In this case the perturbations do not inject energy to the black hole beyond the microcanonical band, and effectively, only the parts of the state (2.1) within each microcanonical window evolve in time.

The problem is, however, that the projectors $\Pi_M$ into high-energy windows will be highly complex operators expressed in terms of the "computational basis" of CFT fields with a gravitational description. In fact, it is not clear whether such a projection is compatible with a

semiclassical description for the projected operator.[37] The microcanonical unitaries $U_M(t)$ can serve in microscopic models, or in discrete tensor network toy models, but they loose connection to the semiclassical bulk.

# B    A simple toy model of the states

In section 2, we have presented an ensemble of ER caterpillars $\mathcal{E}_\Psi^t$ where each draw has rather complicated geometric details arising from the time-dependent couplings used to prepare the state. To better grasp some of the geometric properties of individual states, it is useful to consider the following simplified toy model.

Consider replacing the random circuit $U(t)$ by an ensemble of operators $\mathcal{O}$ that create spherical thin shells of dust particles. Such operators are composed of a large number of primaries inserted at different points along the spatial sphere [143], in an approximately homogeneous arrangement.[38] The ensemble $\mathcal{E}_\mathcal{O}$ is defined by a random variable describing the number of particles that the thin shells can have, i.e., the rest mass $m$ of the shel. Equivalently, the random variable is the conformal dimension $\Delta$ of the corresponding operator, given that the mass of the shell is $m = \Delta \ell_{\mathrm{AdS}}^{-1}$.

For concreteness, we shall define $\mathcal{E}_\mathcal{O}$ in terms of the normally distributed random variable

$$m \sim \mathcal{N}(m_0, \sigma_m^2), \tag{B.1}$$

of mean $m_0$ and variance $\sigma_m^2$.

We can then model the gradually cooled random circuit by an operator of the form

$$\mathcal{O}(n) = \mathcal{O}_1 e^{-\delta\beta H_0} \mathcal{O}_2 e^{-\delta\beta H_0} \cdots e^{-\delta\beta H_0} \mathcal{O}_n e^{-\delta\beta H_0}. \tag{B.2}$$

Here $n$ is the number of shell operators, which plays the role of the discrete circuit time, $n = Jt$. Each operator $\mathcal{O}_i$ is an independent draw from the ensemble $\mathcal{E}_\mathcal{O}$.

The ensemble of states $\mathcal{E}_\Psi^n$ that we consider has the form

$$|\widetilde{\Psi}(n)\rangle = \frac{1}{\sqrt{Z(\beta)}} \sum_{i,j} e^{-\frac{\beta}{4}(E_i + E_j)} \mathcal{O}(n)_{ij} |E_i^*\rangle_\mathsf{L} \otimes |E_j\rangle_\mathsf{R}. \tag{B.3}$$

These states are the discrete-time analogs of (2.8) with the ensemble of thin-shell operators playing the role of the random circuit.

The advantage of considering these states is that the dual geometry of individual instances

---

[37] There is an exception for which $\Pi_M$ is known in the gravitational variables: BPS black holes whose near-horizon dynamics is described by the $\mathcal{N} \geqslant 2$ supersymmetric JT theory. In $\mathrm{AdS}_5 \times \mathbf{S}^5$ these correspond to $\frac{1}{16}$-BPS black holes [139, 140]. In this case the BPS projector $\Pi_M$ is the infinite Euclidean evolution with the Hamiltonian in the relevant sector of fixed charges. In the bulk this operation corresponds to the projection into a particular quantum state of the wormhole of finite expected renormalized length [141, 142].

[38] Thin shell operators have been introduced in various different contexts in the literature (see e.g. [124, 143–150]).

can be obtained from the norm,

$$Z_\Psi(n) = \langle \widetilde{\Psi}(n)|\widetilde{\Psi}(n)\rangle = Z(\beta)^{-1}\mathrm{Tr}\left(e^{-\frac{\beta}{2}H_0}\mathcal{O}(n)e^{-\frac{\beta}{2}H_0}\mathcal{O}(n)^\dagger\right), \tag{B.4}$$

which in this case is an Euclidean CFT path integral. Thus $Z_\Psi(n)$ can be more easily evaluated in a saddle point approximation in the bulk in terms of an Euclidean manifold $M_\Psi$. The semi-classical dual to (B.3) is shown in Fig. 24. We omit the details of the Euclidean preparation here, see e.g. [138, 145] for details.

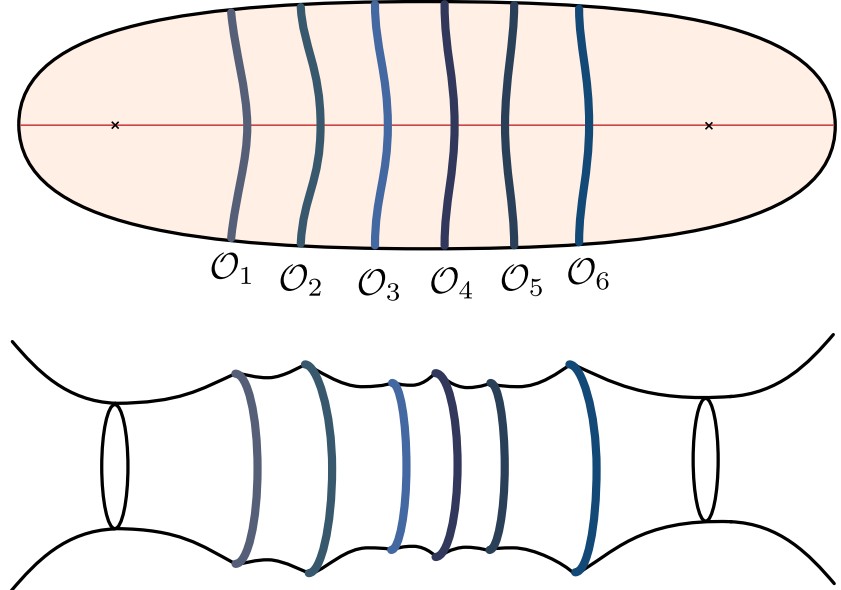

**Figure 24:** On top, the Euclidean saddle point geometry $M_\Psi$ providing the dominant contribution to $Z_\Psi(n)$ for a given instance of a spherical ER caterpillar with $n = 6$. The shell operators are chosen randomly from a gaussian distribution of operators with different masses, and for this reason they have different Euclidean trajectories. The assumption is that to leading order the trajectories correspond to gaussian contractions, and the different masses play the role of different flavors. On the bottom, the semiclassical dual to $|\widetilde{\Psi}(n)\rangle$, prepared on the red cut, has a cylidnrical geometry with length proportional to the number of shells. The variance of the length over the ensemble $\mathcal{E}_\Psi^n$ is suppressed by the central limit theorem.

For a given instance, the total length of the spherical ER caterpillar is given by [145]

$$\frac{\ell_\Psi(n)}{\ell_{\mathrm{AdS}}} = \alpha\ell_{\mathrm{AdS}}(m_1 + m_2 + ... + m_n). \tag{B.5}$$

where $\alpha$ is some coefficient that depends on the dimension, as well as on the small Euclidean time $\delta\beta$ used to cool the state.

The expectation value of the length of the wormhole over the ensemble of states $\mathcal{E}_\Psi^n$ is given by

$$\ell(n) = \mathbb{E}_{\mathcal{E}_\Psi^n}[\ell_\Psi(n)], \qquad \frac{\ell(n)}{\ell_{\mathrm{AdS}}} = \alpha n m_0 \ell_{\mathrm{AdS}}. \tag{B.6}$$

The variance of the length is then,

$$\sigma_\ell^2 = \mathrm{Var}_{\mathcal{E}_\Psi^n}[\ell_\Psi(n)] = n(\alpha\ell_{\mathrm{AdS}}^2)^2 \sigma_m^2 \,. \tag{B.7}$$

Therefore the ensemble of ER caterpillars will have a self-averaging notion of length when

$$\frac{\sigma_\ell}{\ell(n)} = \frac{\sigma_m}{m_0\sqrt{n}} \ll 1 \,, \tag{B.8}$$

i.e. when $n \gg 1$. In fact, more generally, from the central limit theorem, $\ell_\Psi(n)$ asymptotes for large $n$ to a gaussian random variable with mean $\ell(n)$ and variance $\propto \sigma_m^2 n$, irrespectively of the form of the original probability distribution defining $\mathcal{E}_\mathcal{O}$. Thus, the variance is suppressed with respect to the average by $1/\sqrt{n}$ over the ensemble.

The microscopic interpretation of the operator $\mathcal{O}$ dual to a fixed thin shell is not so well controlled, and thus we will not develop this toy model further. In particular, the semiclassical description of a single heavy thin shell is compatible with a gaussian ETH ensemble of microscopic operators (see [124, 126, 148, 149]). So, even for a single semiclassical realization of the ensemble, there would exist a boundary "pseudo-random" ensemble of microscopic states compatible with it [138]. Moreover, for thin shells with finitely many insertions of the same primary, there might exist semiclassical corrections to the naive gaussian propagator for the thin shell which might eventually become important. Still, a significant technical advantange of these toy models is that the precise geometry can be found in general dimensions.

## C   Comments on the effective Hamiltonian

In this appendix we discuss the phase diagram of the effective Hamiltonians $H_{\mathrm{eff},k}$ involving $k$ forward and $k$ backward replicas indexed by $r$ and $\bar{r}$. For convenience, we repeat its form here:

$$H_{\mathrm{eff},k} = \sum_r (H_0^r + H_0^{\bar{r}}) + \frac{J}{2}\sum_{\alpha=1}^K \left(\sum_r \mathcal{O}_\alpha^r - \sum_r \mathcal{O}_\alpha^{\bar{r}\,*}\right)^2 \,. \tag{C.1}$$

This Hamiltonian has a replica symmetry which includes independent permutations of the forward ($r$) and backward ($\bar{r}$) replicas. Here we assume that the individual replicas possess a thermodynamic limit and consider the phase diagram in the thermodynamic limit. Note that the $\mathcal{O}^{\bar{r}\,*}$ operators have appropriate factors of charge conjugation, reflection, and time-reversal symmetries as needed; see e.g. [99].

A priori, the phase diagram depends on the single replica Hamiltonian $H$, the set of drive operators $\{\mathcal{O}_\alpha\}$, and the coupling $J$. In particular, whether the ground states of $H_{\mathrm{eff},k}$ spontaneously break the replica symmetry depends on the system details. This observation suggests that the study of these effective Hamiltonians is an interesting general problem. For example, in the future it would be interesting to study spin glass models (to probe the extent to which random circuits fail or succeed at thermalizing glassy systems), topological models (to probe the slow mixing and potential stability of topologically ordered phases), and other solvable models such as vector models (to gain further concrete data about the behavior of $H_{\mathrm{eff},k}$ at weak

coupling). Here we sketch out a minimal scenario that we expect to hold in sufficiently chaotic systems.

**Two-replica case $k = 1$.** Consider first one forward and one backward contour. As $J \to \infty$, the effective Hamiltonian reduces to

$$\frac{J}{2} \sum_{\alpha=1}^{K} \left( \mathcal{O}_\alpha^1 - \mathcal{O}_\alpha^{\bar{1}\,*} \right)^2 . \tag{C.2}$$

This operator is positive and has the infinite temperature TFD state as an exact zero-energy ground state. Note that the total Hilbert space must be regulated to be finite-dimensional if this state is to be well defined.

With a sufficiently general set $\{\mathcal{O}_\alpha\}$, the infinite temperature TFD is the unique ground state. In many cases, there will also be a gap to excitations above this unique ground state. These properties can be verified in an explicit example as discussed in [75].

Away from the $J \to \infty$ limit, the single-replica Hamiltonian terms in $H_{\text{eff},1}$ must be taken into account. Much work, including [61,99–101], has shown that the ground state of the effective Hamiltonian can approximate a TFD state at a tunable temperature $\beta(J)$.

In particular, in [99] it is argued that if ETH holds, then the ground state of $H_{\text{eff},1}$ will be a TFD-like state with a well-peaked energy distribution. It differs from the canonical TFD state only in that it may have a different width of its energy distribution.

The assumption of ETH actually yields two important conclusions. First, the ground state of $H_{\text{eff},1}$ lives in a diagonal subspace spanned by $|E_i\rangle \otimes |E_i\rangle$.[39] Second, the ground state wavefunction has the form

$$|\text{GS}\rangle = \sum_i c_i |E_i^*\rangle \otimes |E_i\rangle , \tag{C.3}$$

where

$$c_i = \sqrt{\frac{e^{-(E_i - \overline{E})^2 / 2\sigma_E^2}}{\rho(E_i)}} . \tag{C.4}$$

In this form, $\overline{E}$, and $\sigma_E^2$ are constants and $\rho(E) = e^{S(E)}$ is the thermodynamic density of states, normalized to have $\sum_i c_i^2 = 1$. The parameter $\overline{E}$ sets the average energy and $\sigma_E^2$ controls the width of the energy distribution. By varying the coupling $J$, one effectively varies both $\overline{E}$ and $\sigma_E^2$. The TFD at a given temperature has the same form with parameters $E_\beta$ and $\sigma_{E,\text{TFD}}^2$; adjusting $J$ and $\beta$ to set $\overline{E} = E_\beta$ leaves open the possibility that the variances, $\sigma_E^2$ and $\sigma_{E,\text{TFD}}^2$, differ.

The TFD-like nature of the ground state becomes exact in two limits: it reduces to the zero-temperature TFD when $J = 0$ and to the infinite temperature TFD when $J \to \infty$. If the ground state is TFD-like for all $J$ and if there is no phase transition as a function of $J$, it follows that a TFD-like state for any desired value of $\beta$ can be obtained by dialing $J$. The only difference is

---

[39] The restriction to the diagonal subspace can be obtained in our setting even without ETH. If each time step includes an extra period of real time evolution with a Gaussian distributed duration, then $\frac{\kappa}{2}(H_0^1 - H_0^{\bar{1}})^2$ is included in the effective Hamiltonian. When $\kappa$ is large, this term forces the ground state into the diagonal subspace.

that the resulting ground state may have a different energy variance than the TFD state.

This analysis makes a clear prediction for the behavior of the overlap between the ground state and the optimized TFD state. The overlap of two Gaussians with the same central value depends only on the $\sigma_E$ parameters. The overlap is

$$\sqrt{\frac{2\sigma_E \tilde{\sigma}_E}{\sigma_E^2 + \tilde{\sigma}_E^2}}, \tag{C.5}$$

and an extensive energy variance in the thermodynamic limit yields an overlap which is generically less than unity but non-vanishing as $N \to \infty$.

**Multi-replica case** $k > 1$. Now consider $k > 1$ forward contours and an equal number of backward contours. In the large $J$ limit, with the same set of assumptions that led to a unique gapped ground state for $H_{\text{eff},1}$, $H_{\text{eff},k}$ is expected to have $k!$ ground states that spontaneously break the replica symmetry. These ground states are obtained by choosing one of the $k!$ pairings between the forward and backward contours and associating a $k = 1$ ground state to each pair of contours (just the maximally entangled state when $J \to \infty$). Provided $k$ is less than the dimension of the Hilbert space, these $k!$ states will be linearly independent. Crucially, if $H_{\text{eff},1}$ is gapped, then $H_{\text{eff},k}$ is expected to be gapped as well. These expectations have been substantiated in various concrete models, e.g. [75] and appendix D.

Moving away from $J = \infty$, the Hamiltonian terms must again be considered. Since the system spontaneously breaks a discrete symmetry as $J \to \infty$, one expects a stable phase characterized by the same pattern of symmetry breaking for large but finite $J$. However, there is no guarantee that this phase persists down to small $J$. Even if there is no phase transition when $k = 1$, there could be a critical $J$ depending on $k$ below which the phase structure of $H_{\text{eff},k}$ changes.

The simplest ansatz is that the pattern of replica symmetry breaking persists for all $J > 0$ in the thermodynamic limit. Then the picture of the ground states is qualitatively similar to the $J \to \infty$ limit: $H_{\text{eff},k}$ has $k!$ ground states built from a choice of pairing and $k$ copies of the $k = 1$ TFD-like ground state. It is this ansatz which we assume in the main text; technically, this amounts to assuming the dominance of certain saddle point configurations of the gravitational path integral. We are not aware of other competing saddles that would change this conclusion, but we have not been able to rule them out.

With this ansatz, the ground state energy and low energy excitations are simple to describe. Consider without loss of generality the standard pairing, $1\bar{1}$, $2\bar{2}$, etc.; in this case $H_{\text{eff},k}$ may be decomposed as

$$H_{\text{eff},k} = \sum_r H_{\text{eff},1}^{(r\bar{r})} - \frac{J}{2} \sum_\alpha \sum_{r \neq s} (\mathcal{O}_\alpha^r - \mathcal{O}_\alpha^{\bar{r}\,*})(\mathcal{O}_\alpha^s - \mathcal{O}_\alpha^{\bar{s}\,*}). \tag{C.6}$$

Since the ground state is a tensor product of pairwise entangled states, if follows that

$$\langle \text{GS} | (\mathcal{O}_\alpha^r - \mathcal{O}_\alpha^{\bar{r}\,*})(\mathcal{O}_\alpha^s - \mathcal{O}_\alpha^{\bar{s}\,*}) | \text{GS} \rangle = \langle \text{GS} | (\mathcal{O}_\alpha^r - \mathcal{O}_\alpha^{\bar{r}\,*}) | \text{GS} \rangle \langle \text{GS} | (\mathcal{O}_\alpha^s - \mathcal{O}_\alpha^{\bar{s}\,*}) | \text{GS} \rangle, \tag{C.7}$$

provided $r \neq s$. Furthermore, the symmetry of the $k = 1$ ground state then implies that

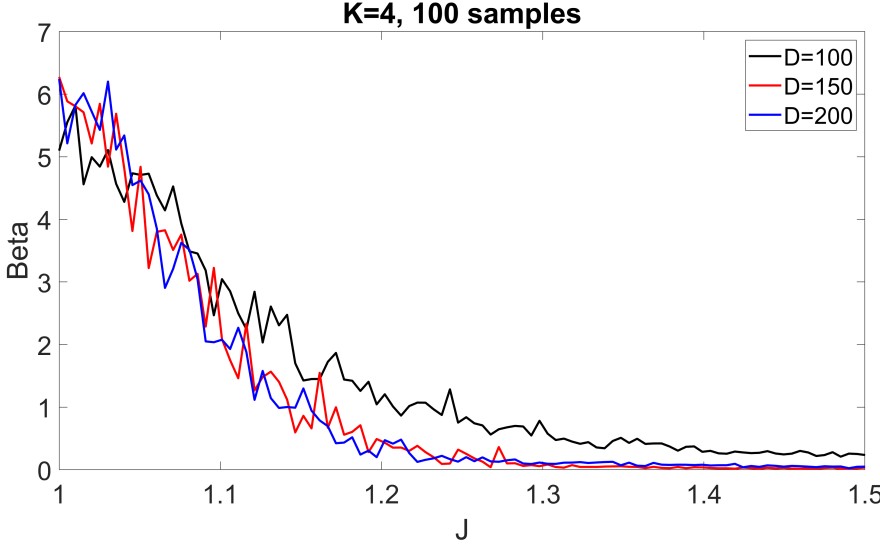

**Figure 25:** Optimized $\beta$ as a function of $J$ in the GOE model with $K = 4$ operators and $N_s = 100$ samples. Single replica Hilbert space dimensions are $D = 100, 150, 200$.

$\langle \mathrm{GS}|(\mathcal{O}_\alpha^r - \mathcal{O}_\alpha^{\bar{r}}{}^*)|\mathrm{GS}\rangle = 0$. Hence, the ground state energy is

$$\langle \mathrm{GS}|H_{\mathrm{eff},k}|\mathrm{GS}\rangle \approx k\langle \mathrm{GS}|H_{\mathrm{eff},1}|\mathrm{GS}\rangle. \tag{C.8}$$

Moreover, the lowest energy excitations correspond to excitations on top of a single pair, i.e. an excitation of $H_{\mathrm{eff},1}$.

**Numerics.** To substantiate the claims made in the $k = 1$ discussion above and to investigate the dimension dependence of the overlap, we present numerical results for a simple random Hamiltonian model. In this model, the single copy Hamiltonian $H_0$ is drawn from the Gaussian orthogonal ensemble (GOE) of dimension $D$. We also include $K$ perturbations $\mathcal{O}_\alpha$ each drawn independently from the GOE. We project $H_{\mathrm{eff},1}$ onto the diagonal subspace spanned by $|E_i\rangle \otimes |E_i\rangle$ where $|E_i\rangle$ are the eigenstates of $H_0$. We then diagonalize the resulting matrix and extract the ground state. Finally, we scan over $\beta$ values looking for the TFD state with the highest overlap with the projected $H_{\mathrm{eff},1}$ ground state. The highest overlap TFD state and the corresponding $\beta$ depend on $J$, as well as on $K$ and the choice of ensemble. We also average the absolute value of the optimized overlap over $N_s$ samples.

We show the $J$-dependence of the optimal $\beta$ in Fig. 25 and the corresponding overlap in Fig. 26. For this calculation, we set $K = 4$, took $N_s = 100$ samples, and considered dimensions $D = 100, 150, 200$. We see some variation with the Hilbert space size, although $D = 150$ and $D = 200$ are similar, suggesting a large-$D$ limit is already visible up to statistical errors. Since the range of $J$ is only $[1, 1.5]$, the variation of $\beta$ shown in Fig. 25 is relatively rapid. At small $J$ (not shown), we observed a non-monotonicity of the overlap possibly suggesting a phase transition of some sort. The overlap data in Fig. 26 is consistent with non-vanishing overlap in the thermodynamic limit of large $D$.

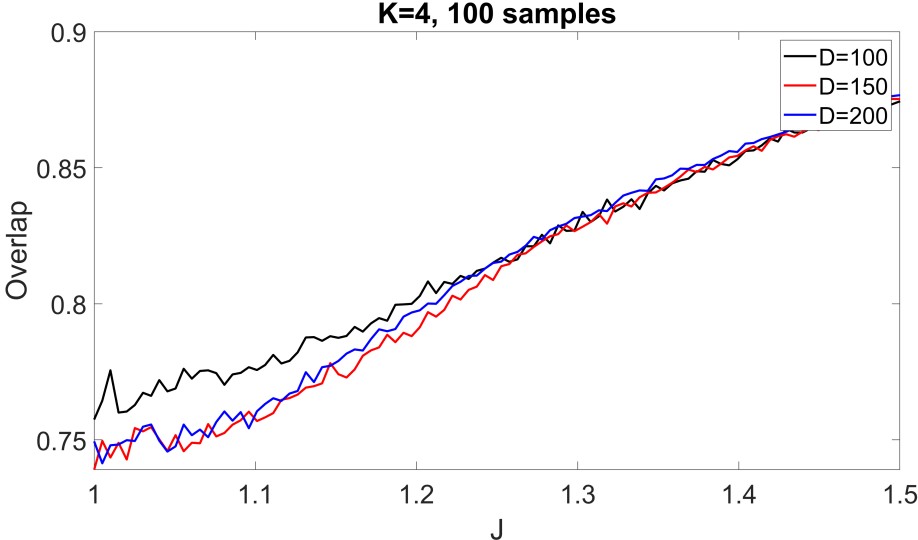

**Figure 26:** Optimized overlap (absolute value) as a function of $J$ in the GOE model with $K = 4$ operators and $N_s = 100$ samples. Single replica Hilbert space dimensions are $D = 100, 150, 200$.

## D  SYK caterpillars

In this appendix, we consider the ensemble of ER caterpillars $\mathcal{E}_\Psi^t$ of the form (2.8) defined in two SYK models. We will analyze the ensemble below the low-temperature regime discussed in sections 2.3 and 3.3. We first define the random circuit perturbing the $q$-SYK model Hamiltonian with a bosonic perturbation

$$H(t) = H_{\text{SYK}} + \mathrm{i}^{\frac{p}{2}} \sum_{I_p} g_{I_p}(t)\, \psi_{I_p} \,, \tag{D.1}$$

where we have defined the collective index $I_p \equiv (i_1, ..., i_p)$, with $i_1 < ... < i_p$, and the Majorana string $\psi_{I_p} \equiv \psi_{i_1}...\psi_{i_p}$. Moreover, the $q$-SYK Hamiltonian $H_{\text{SYK}}$ is given by (2.28), and the scale of the SYK couplings is set by $\mathcal{J}$. The couplings of the perturbations are white-noise gaussian correlated variables,

$$\mathbb{E}\big[g_{I_p}(t)\big] = 0 \,, \qquad \mathbb{E}\big[g_{I_p}(t)g_{I'_p}(0)\big] = \frac{JN}{\binom{N}{p}}\, \delta(t)\delta_{I_p I'_p} \,. \tag{D.2}$$

The ensemble of states $\mathcal{E}_\Psi^t$ has the general form outlined in section (2.8). Applying the general considerations of sections 2 and 3 to this ensemble, the physical properties will be captured by the $2k$-replica effective Hamiltonian

$$H_{\text{eff},k} = \sum_{r=1}^{k} \left( H_{\text{SYK}}^r + H_{\text{SYK}}^{\bar{r}} \right) + JN \left( k - \sum_{r,s} \psi_p^{r\bar{s}} + (-1)^{\frac{p}{2}} \sum_{r<s} (\psi_p^{rs} + \psi_p^{\bar{r}\bar{s}}) \right) , \tag{D.3}$$

where we have defined the fermion bilinears

$$\psi_p^{ab} \equiv \frac{1}{\binom{N}{p}} \sum_{i_1 < \dots < i_p} \psi_{I_p}^a \psi_{I_p}^b \,. \tag{D.4}$$

Again, the index $r(\bar{r})$ labels the forward (backward) replica. The indices $a, b$ include both $r$ and $\bar{r}$. The factor of $(-1)^{\frac{p}{2}}$ represents the fact that for $p \equiv 2 \pmod 4$ the perturbation does not possess time-reflection symmetry, and the complex conjugate of the operators in the backward replicas pick up a relative minus sign.[40]

To analyze the properties of the effective Hamiltonian with the path integral of the model, we consider the thermal partition function

$$Z_k(t) = \mathrm{Tr} \exp(-tH_{\mathrm{eff},k}) \,. \tag{D.5}$$

For the infinite temperature case, we have seen that $F_k(t) = Z_k(2t)$ is the frame potential of the ensemble of states $\mathcal{E}_\Psi^t$ prepared with the random circuit at infinite temperature, i.e., the states of the form (3.4). At finite temperature, one must instead consider the moment two-point functions $G_k(t, t')$ introduced in section 3.2. However, this adds complications that we shall not analyze here. Also, we will be brief here, most of what follows is review of [61, 75, 92].

Let us denote by $\psi_i^a(s)$ the Majorana fermion of the forward and backward replicas. The quantity $Z_k(t)$ corresponds to an Euclidean partition function of the Majorana fermions on $2k$ closed contours of time $t$. Consider the bi-local collective fields $G_{ab}(s, s') = \frac{1}{N} \sum_i \psi_i^a(s)\psi_i^b(s')$ and $\Sigma_{ab}(s, s')$. After averaging over SYK couplings, and integrating out the Majorana fermions, we get

$$Z_k(t) = \int \prod_{a,b} \mathcal{D}G_{ab}\mathcal{D}\Sigma_{ab} \exp\left(-I[G_{ab}, \Sigma_{ab}]\right) \,, \tag{D.6}$$

for the large-$N$ effective action

$$\frac{I[G_{ab}, \Sigma_{ab}]}{N} = -\log \mathrm{Pf}(\partial_s \delta_{ab} - \Sigma_{ab}) +$$
$$+ \frac{1}{2} \int_0^t \mathrm{d}s\mathrm{d}s' \left(\sum_{a,b} \left[\Sigma_{ab}G_{ab} - \mathcal{J}^2 c_{ab}G_{ab}^q\right] + 2J\,\delta(s - s')\left(k - \mathrm{i}^p \sum_{a,b} \kappa_{ab}G_{ab}^p\right)\right) \,. \tag{D.7}$$

---

[40] Here we think about bosonic perturbations, so $p$ is even. Then, there is a subtlety which has been mentioned in section 3.3. In this case, there is a conserved global symmetry by the ensemble of random circuits $\mathcal{E}_U^t$, corresponding to fermion parity $(-1)^F$. A similar analysis can be carried in a bosonic $q$-spin model, where this complication is absent [75]. This means that the states will only become block-diagonal random on each superselection sector of $(-1)^F$ (see [76]). Moreover, depending on the value of $N \pmod 8$, there can exist additional discrete symmetries, associated to the particle-hole symmetry of the model. We consider $N \neq 4 \pmod 8$ to avoid degeneracies and additional complications.

for the coefficients

$$c_{ab} = \begin{cases} c_{rs} = c_{\bar{r}\bar{s}} = 1 \,, \\ c_{r\bar{s}} = c_{\bar{r}s} = (-1)^{\frac{q}{2}} \end{cases}, \qquad \kappa_{ab} = \begin{cases} \kappa_{rs} = \kappa_{\bar{r}\bar{s}} = (-1)^{\frac{p}{2}} \\ \kappa_{r\bar{s}} = \kappa_{\bar{r}s} = -1 \end{cases}. \tag{D.8}$$

The Schwinger-Dyson equations of this model are thus

$$G_{ab} = (\partial_s - \Sigma)_{ab}^{-1} \,, \tag{D.9}$$

$$\Sigma_{ab} = \mathcal{J}^2 q c_{ab} G_{ab}^{q-1} + J p \delta(s - s') \mathrm{i}^p G_{ab}^{p-1} \kappa_{ab} \,. \tag{D.10}$$

**Infinite temperature wormhole.** When $\mathcal{J}/J \to 0$ these equations reduce to those for the Brownian $p$-SYK model studied in [92] for $k = 1$ and in [75] for any $k$. We will write down the large-$N$ semiclassical wormhole solutions for the purely Brownian SYK model studied in [75,92].

Consider $k = 1$ and (D.9) and (D.10) with $J/\mathcal{J} \to \infty$. In this case $\Sigma_{11} = \Sigma_{\bar{1}\bar{1}} = 0$. Moreover, (D.10) requires that

$$\Sigma_{1\bar{1}}(s) = \mathsf{S}_{1\bar{1}} \, \delta(s) \,, \tag{D.11}$$

The reason for a delta function is that here $s$ is the difference between the fermion times, and for a constant mass term this has to be zero. Note that we have already imposed that $\Sigma_{1\bar{1}}(s, s')$ cannot depend on $s + s'$. Similarly $\Sigma_{\bar{1}1}(s) = -\Sigma_{1\bar{1}}(s)$. This leads to the equation

$$\begin{pmatrix} G_{11}^{(n)} & G_{1\bar{1}}^{(n)} \\ G_{\bar{1}1}^{(n)} & G_{\bar{1}\bar{1}}^{(n)} \end{pmatrix} = -\begin{pmatrix} \mathrm{i}\omega_n & \mathsf{S}_{1\bar{1}} \\ -\mathsf{S}_{1\bar{1}} & \mathrm{i}\omega_n \end{pmatrix}^{-1} \,. \tag{D.12}$$

Inverting the matrix we get

$$G_{11}^{(n)} = G_{\bar{1}\bar{1}}^{(n)} = -\frac{\mathrm{i}\omega_n}{\mathsf{S}_{1\bar{1}}^2 - \omega_n^2} \,, \qquad G_{1\bar{1}}^{(n)} = -G_{\bar{1}1}^{(n)} = \frac{\mathsf{S}_{1\bar{1}}}{\mathsf{S}_{1\bar{1}}^2 - \omega_n^2} \,. \tag{D.13}$$

This yields the real time solutions[41]

$$G_{11}(s) = G_{\bar{1}\bar{1}}(s) = (\mathrm{sign}(s) + A)\cos(\mathsf{S}_{1\bar{1}}s) - B\sin(\mathsf{S}_{1\bar{1}}s) \,, \tag{D.14}$$

$$G_{1\bar{1}}(s) = -G_{\bar{1}1}(s) = (1 + A)\sin(\mathsf{S}_{1\bar{1}}s) + B\cos(\mathsf{S}_{1\bar{1}}s) \,. \tag{D.15}$$

This solution can be obtained by doing the Fourier transform of (D.13) for $\omega \in \mathbf{R}$ and picking the correct $\mathrm{i}\epsilon$ prescription, namely $(\mathsf{S}_{1\bar{1}}^2 - \omega^2 + \mathrm{i}\epsilon)$. Moreover $A$ and $B$ parametrize the two independent homogeneous solutions to (D.12). The values of $A$ and $B$ are determined by the condition that (D.14) and (D.15) must be antiperiodic, i.e., that only the appropriate Matsubara frequencies survive,

$$A = 0 \,, \qquad B = -\tan\left(\frac{\mathsf{S}_{1\bar{1}}t}{2}\right) \,. \tag{D.16}$$

---

[41] In our normalization of the fermions $\psi_i^2 = 1$, so the free fermion propagator is $G_{\mathrm{free}}(s) = \mathrm{sign}(s)$.

Noting that $G_{1\bar{1}}(0) = -\tan\left(\frac{S_{1\bar{1}}t}{2}\right)$, the remaining equation of motion (D.10) reduces to

$$S_{1\bar{1}} = -Jp\mathrm{i}^p \tan\left(\frac{S_{1\bar{1}}t}{2}\right)^{p-1}. \tag{D.17}$$

The two non-trivial solutions $S_{1\bar{1}}(t) = \pm\mathrm{i}\mu(t)$ are purely imaginary, $\mu(t) \in \mathbf{R}^+$. They approach $\mu(t) \to Jp$ at late times. In terms of the real parameters, these two solutions are

$$G_{11}^{\pm}(s) = G_{\bar{1}\bar{1}}^{\pm}(s) = \mathrm{sign}(s)e^{-\mu|s|} + \frac{2\sinh(\mu s)}{1 + e^{\mu t}}, \tag{D.18}$$

$$G_{1\bar{1}}^{\pm}(s) = -G_{\bar{1}1}^{\pm}(s) = \mp\mathrm{i}\left(e^{-\mu|s|} - \frac{2\cosh(\mu s)}{1 + e^{\mu t}}\right). \tag{D.19}$$

for $s \in (-t, t)$ and its periodic images to extend them to $s \in \mathbf{R}$.

As $t \to \infty$ the solutions behave as

$$G_{11}^{\pm}(s) = \mathrm{sign}(s)\, e^{-Jp|s|}, \tag{D.20}$$

$$G_{1\bar{1}}^{\pm}(s) = \pm\mathrm{i}e^{-Jp|s|}, \tag{D.21}$$

These are the correlation function of a gapped Hamiltonian with gap $E_{\mathrm{gap}} = Jp$. In fact, in this case the effective Hamiltonian $H_{\mathrm{eff},1}$ is trivially gapped, i.e. it corresponds to a massive $1\bar{1}$ Dirac fermion (see [75, 76, 92]).

The fermion determinant is simply the real-time partition function for time $t$ of a free Dirac fermion of mass $S = \mathrm{i}\mu$, which gives $2\cosh(t\mu)$. The effective action is then

$$Z_1(t) = \int dG_{1\bar{1}}d\Sigma_{1\bar{1}} \exp\left(-N\left[-\log(2\cosh(tS_{1\bar{1}})) + tS_{1\bar{1}}G_{1\bar{1}}(0) + Jt\left(1 - \mathrm{i}^p G_{1\bar{1}}(0)^p\right)\right]\right). \tag{D.22}$$

The solution $G_{1\bar{1}} = \Sigma_{1\bar{1}} = 0$ is a saddle point, which does not contain correlations between replicas. Its on-shell action is given by $-\dfrac{I_{\mathrm{disc}}}{N} = -Jt$, so its contribution decays exponentially fast in $JN$. On the other hand, another two saddle points are $G_{1\bar{1}}^{\pm}(0) \approx \pm\mathrm{i}$ and $S_{1\bar{1}}^{\pm} = \mp Jp\mathrm{i}$ for large enough values of $t$, which gives the on-shell action

$$-\frac{I_{\mathrm{wh}}}{N} = \log(2\cosh(Jtp)) - Jtp \longrightarrow e^{-2Jpt} \tag{D.23}$$

The on-shell action becomes time-independent at late times, and its contribution corresponds to the plateau of the frame potential. Note that the saddle point value of the action vanishes at late times, as for the double cone.

At late times $t \gg t_{\star}$ this gives

$$Z_1(t) \approx 2\exp\left(Ne^{-2Jpt}\right) \approx 2 + 2Ne^{-2Jpt}. \tag{D.24}$$

The associated onset timescale can be estimated to be

$$t_\star \sim J^{-1} \log N \,, \tag{D.25}$$

where in this case of infinite temperature, $E_{\text{gap}} = 2Jp$. Note that the timescale at which the disconnected contribution becomes negligible is smaller than (D.25).

For $k > 1$, we can also solve for the $G_{ab}$ and $\Sigma_{ab}$ assuming a replica symmetry breaking solution into $2 \times 2$ submatrices. This amounts to only considering the two-replica disconnected solution and the two-replica wormhole. The result is then [75]

$$Z_k(t) = \sum_{n=0}^{k} 2^n n! \binom{k}{n}^2 Z_{\text{wh}}^n Z_{\text{disc}}^{k-n} \,. \tag{D.26}$$

The dominant contribution at late times comes from $n = k$ and this gives

$$Z_k(t) \approx 2^k k! \left(1 + Nk e^{-2Jpt}\right) \tag{D.27}$$

The analysis in this appendix supports the results obtained in section 3.3 in the low-temperature regime of the SYK model, namely (3.83). We shall not attempt to study finite but not so small temperatures here, although an analytic treatment should be possible at large-$q$, following the two-replica analysis of [61].

# E    Unitary $k$-designs for Brownian systems

In this appendix we study a different measure of $k$-randomness, which is intrinsic to the random circuit used to prepare the states, rather than to the states themselves. To begin, we will consider a finite Hilbert space $\mathcal{H}$ of dimension $L = e^S$. We shall consider an ensemble of unitaries $\mathcal{E}_U$, formally defined by some probability measure – either discrete or continuous – in $\mathsf{U}(L)$. The $k$-th moment superoperator of the ensemble $\mathcal{E}_U$ is defined as

$$\hat{\Phi}_{\mathcal{E}_U}^{(k)} \equiv \mathbb{E}\underbrace{\left[U \otimes ... \otimes U \otimes U^* \otimes ... \otimes U^*\right]}_{2k} \,. \tag{E.1}$$

A *unitary $k$-design* is an ensemble of unitaries $\mathcal{E}_U$ which reproduces the $k$-th moment of the Haar distribution,

$$\mathcal{E}_U \text{ is a } k\text{-design} \quad \Leftrightarrow \quad \hat{\Phi}_{\mathcal{E}_U}^{(k)} = \hat{\Phi}_{\text{Haar}}^{(k)} \,. \tag{E.2}$$

Physically, a 1-design can serve to model a system which is locally thermalized on average, while a 2-design is fully scrambled for any observable as measured by the four-point OTOC [68].

An *$\varepsilon$-approximate unitary $k$-design* is an ensemble of unitaries $\mathcal{E}_U$ which reproduces the first $k$ moments of the Haar distribution, given some resolution $\varepsilon$. For our purposes the resolution will be given in terms of the trace distance between moment superoperators, so

$$\mathcal{E}_U \text{ is an } \varepsilon\text{-approximate } k\text{-design} \quad \Leftrightarrow \quad \left\|\hat{\Phi}_{\mathcal{E}_U}^{(k)} - \hat{\Phi}_{\text{Haar}}^{(k)}\right\|_1 < \varepsilon \,. \tag{E.3}$$

Other definitions of approximate unitary designs also exist. These are most commonly formulated in terms of the diamond distance between moment channels $\|\Phi_{\mathcal{E}}^{(k)} - \Phi_{\text{Haar}}^{(k)}\|_\diamond$ as defined in [151, 152]. The diamond distance is physically relevant because it quantifies the one-shot distinguishability between quantum channels when ancilla-assisted measurements are allowed. However, all of the different definitions of approximate $k$-designs are equivalent up to factors of the dimension (see [65]). In particular, $\|\Phi_{\mathcal{E}}^{(k)} - \Phi_{\text{Haar}}^{(k)}\|_\diamond \leqslant \|\hat{\Phi}_{\mathcal{E}}^{(k)} - \hat{\Phi}_{\text{Haar}}^{(k)}\|_1$ so the definition that we are using here is stronger (for details we refer the reader to Appendix B of [76]).[42]

**Moment superoperators of Haar.** As a consequence of the Schur-Weyl duality (see e.g. [65, 68, 115, 116]), the Haar moment superoperator is the orthogonal projector into the invariant subspace of the full $2k$ replica Hilbert space, generated by the invariant states $|W_\sigma\rangle$ of the form

$$|W_\sigma\rangle = \bigotimes_{r=1}^{k} |I\rangle_{r\sigma(\bar{r})} , \qquad \sigma \in \text{Sym}(k) . \tag{E.4}$$

Explicitly, the $k$-th moment superoperator of the Haar ensemble is

$$\hat{\Phi}_{\text{Haar}}^{(k)} = \sum_{\sigma,\tau} c_{\sigma,\tau} |W_\sigma\rangle\langle W_\tau| , \tag{E.5}$$

where the matrix of coefficients $c_{\sigma,\tau}$ is called the *Weingarten matrix* [117]. These coefficients are determined from the condition that $\hat{\Phi}_{\text{Haar}}^{(k)}$ is an orthogonal projector,

$$\hat{\Phi}_{\text{Haar}}^{(k)} |W_\sigma\rangle = |W_\sigma\rangle \qquad \forall\, \sigma \in \text{Sym}(k) . \tag{E.6}$$

More explicitly, the Gram matrix of overlaps between the invariant states is given by combinatorial data

$$G_{\sigma,\tau} \equiv \langle W_\sigma|W_\tau\rangle = L^{\#\text{cycles}(\sigma\tau^{-1})} , \tag{E.7}$$

where $\#\text{cycles}(\sigma)$ counts the number of cycles of $\sigma$. The condition (E.6) amounts to

$$\sum_\tau c_{\sigma,\tau} G_{\tau,\kappa} = \delta_{\sigma\kappa} \tag{E.8}$$

For $k \leqslant L$ the Gram matrix $G_{\sigma,\tau}$ has full rank and the Weingarten matrix is the inverse of the Gram matrix, $c_{\sigma,\tau} = G_{\sigma,\tau}^{-1}$. For $k > L$ the representation (E.5) is not unique.

**Trace distance to design as a thermal partition function.** For the ensemble of random circuits $\mathcal{E}_U^t$, generated by the Hamiltonian $H(t) = \sum_\alpha g_\alpha(t)\mathcal{O}_\alpha$ with Brownian couplings $g_\alpha(t)$, the $k$-th moment superoperator is the Euclidean time evolution operator [75, 76]

$$\hat{\Phi}_t^{(k)} = \exp(-tH_{\text{eff},k}) , \qquad H_{\text{eff},k} = \frac{J}{2} \sum_{\alpha=1}^{K} \left( \sum_{r=1}^{k} \mathcal{O}_\alpha^r - \mathcal{O}_\alpha^{\bar{r}\,*} \right)^2 , \tag{E.9}$$

Under general assumptions for the set of drive operators $\{\mathcal{O}_\alpha\}$, the invariant states $|W_\sigma\rangle$ are the

---

[42] Other definitions of approximate $k$-designs involve the 2-norm $\|\hat{\Phi}_{\mathcal{E}}^{(k)} - \hat{\Phi}_{\text{Haar}}^{(k)}\|_2$, i.e. the frame potential [68]. From the monotonicity of Schatten norms, it follows that these definitions are also weaker than (E.3).

only ground states of $H_{\text{eff},k}$. From this condition, the ensemble of infinite time random circuits $\mathcal{E}_U^\infty$ is a unitary $k$-design for any $k$,

$$\hat{\Phi}_{\text{Haar}}^{(k)} = \hat{\Phi}_\infty^{(k)} = \exp(-\infty H_{\text{eff},k})\,. \tag{E.10}$$

At finite time $t$, we can compute the trace distance to design $\big\|\hat{\Phi}_t^{(k)} - \hat{\Phi}_{\text{Haar}}^{(k)}\big\|_1$ exactly. To do this one needs to use that both superoperators commute $[\hat{\Phi}_t^{(k)}, \hat{\Phi}_{\text{Haar}}^{(k)}] = 0$ and that they moreover coincide in the ground space of $H_{\text{eff},1}$, while the Haar superoperator vanishes outside of it. Therefore, the Schatten $p$-norm distance between both superoperators can be expressed as

$$\big\|\hat{\Phi}_t^{(k)} - \hat{\Phi}_{\text{Haar}}^{(k)}\big\|_p = \left(\big\|\hat{\Phi}_t^{(k)}\big\|_p^p - \big\|\hat{\Phi}_{\text{Haar}}^{(k)}\big\|_p^p\right)^{1/p}, \qquad \big\|\hat{\Phi}_t^{(k)}\big\|_p^p = \text{Tr}\exp(-tpH_{\text{eff},1})\,. \tag{E.11}$$

For the Haar ensemble, the $p$-norm of $\hat{\Phi}_{\text{Haar}}^{(k)}$ simply corresponds to the number of independent invariant ground states

$$\big\|\hat{\Phi}_{\text{Haar}}^{(k)}\big\|_p^p = k! \qquad \text{for } k \leqslant L\,. \tag{E.12}$$

The trace distance to $k$-design is then

$$\big\|\hat{\Phi}_t^{(k)} - \hat{\Phi}_{\text{Haar}}^{(k)}\big\|_1 = \text{Tr}\exp(-tH_{\text{eff},1}) - k! \qquad k \leqslant L\,. \tag{E.13}$$

Note that the 2-norm distance of the ensemble of unitaries

$$\big\|\hat{\Phi}_t^{(k)} - \hat{\Phi}_{\text{Haar}}^{(k)}\big\|_2^2 = \text{Tr}\exp(-2tH_{\text{eff},1}) - k! \qquad k \leqslant L\,. \tag{E.14}$$

The right hand side of (E.14) agrees with the 2-norm distance of the corresponding two-sided states studied in section 3.1. The trace distance (E.13) also uniquely depends on the spectral randomness of $U(t)$ for the ensemble of random circuits $\mathcal{E}_U^t$, as explained in section 3.1 (see [76]).

## F    Frame potential vs spectral form factor

Consider a finite-dimensional Hilbert space $\mathcal{H}$ of dimension $L$. It is instructive to compare the first frame potential $F_1(t) = L^{-2}\mathbb{E}|\text{Tr}\,U(t)|^2$ for the ensemble of random circuits $\mathcal{E}_U^t$ from the more familiar spectral form factor $\text{SFF}_0(t) \equiv L^{-2}|\text{Tr}\exp(-itH_0)|^2$ for the time-independent Hamiltonian of the black hole $H_0$. The latter has been treated extensively in the literature of black holes and quantum chaos (see e.g. [92, 112, 153–156]).

In order to compare both quantities, we will consider a random circuit $U(t)$ generated by the Schrödinger-picture Hamiltonian with Brownian couplings

$$H(t) = H_0 + \sum_\alpha g_\alpha(t)\mathcal{O}_\alpha\,, \qquad U(t) = \mathsf{T}\exp\left(-\mathrm{i}\int_0^t \mathrm{d}s\,H(s)\right)\,. \tag{F.1}$$

In this case the first frame potential is a real-time partition function of a complex effective

Hamiltonian

$$F_1(t) = L^{-2}\text{Tr}\exp(-\mathrm{i}H_{\text{eff},1}t)\,, \qquad H_{\text{eff},1} = H_0^1 - H_0^{\bar{1}} - \frac{\mathrm{i}J}{2}\sum_\alpha \left(\mathcal{O}_\alpha^1 - \mathcal{O}_\alpha^{\bar{1}\,*}\right)^2\,. \qquad \text{(F.2)}$$

The imaginary part of the effective Hamiltonian is non-positive, and this will make the excited states decay. Under genericity assumptions for the perturbations $H_1$ contains a unique ground state [75]. In Fig. 27 we numerically compare both quantities for a random Hamiltonian $H_0$ from GUE with a single small random perturbation $\mathcal{O}$ from GUE. In what follows, we list the general main differences between these two quantities:

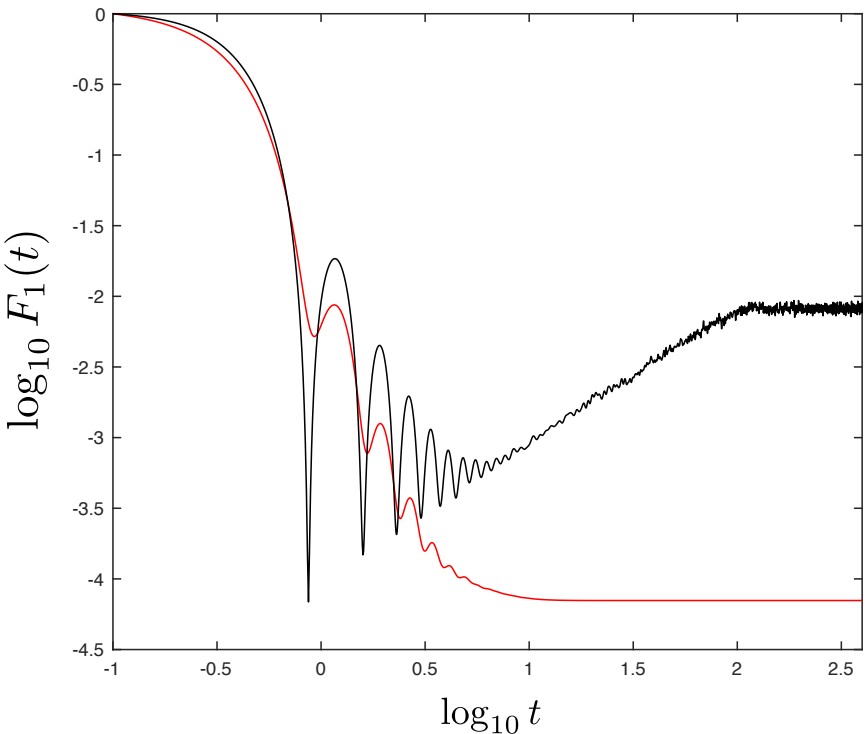

**Figure 27:** In black, the SFF$_0$ averaged over 500 realizations of the time-independent Hamiltonian $H_0$ in the GUE. In red, the first frame potential $F_1(t)$ for a fixed Brownian perturbation of $H_0$, averaged over the same 500 realizations of $H_0$. In this plot $L = 128$ and $H_0$ is drawn from GUE with variance $\log L$ in the maximally mixed state. The perturbation $\mathcal{O}$ is taken from GUE with unit variance in the maximally mixed state.

- The SFF$_0(t)$ for chaotic systems displays the characteristic slope-ramp-plateau structure. The ramp is a consequence of the universal random matrix like spectral correlations in the spectrum of $H_0$. For $F_1(t)$, the small time-dependent perturbation of $H_0$ gets rid of the ramp. This is expected given that the perturbation breaks time-translation symmetry.

- The plateau value of $F_1(t)$ becomes suppressed by a factor of $L^{-1}$ with respect to the plateau value of the SFF$_0$. This was already noted in [92] for Brownian SYK. The reason is that $U(t)$ will incorporate the eigenphase repulsion characteristic of the CUE (3.31) at

late enough times. On the other hand, $\exp(-itH_0)$ will not, even at Heisenberg timescales $t_H \sim L$ where the plateau of the $\mathrm{SFF}_0$ is attained.[43] In fact, the phases of $\exp(-it_H H_0)$ will be uniform on the unit circle but Poisson-distributed, due to the large number of windings around the circle. These different eigenphase correlations lead to the different values of the plateaux, as summarized in Table 1.

| | $\mathbb{E}[\rho(\theta)]$ | $\mathbb{E}[\delta\rho(\theta)\delta\rho(\theta')]$ | $\mathbb{E}\left[|\mathrm{Tr}\,U|^2\right]$ |
|---|---|---|---|
| CUE | $\dfrac{L}{2\pi}$ | $\dfrac{L}{2\pi}\delta(\theta-\theta') - \dfrac{\sin\left(L\frac{\theta-\theta'}{2}\right)^2}{\left(2\pi\,\sin\left(\frac{\theta-\theta'}{2}\right)\right)^2}$ | $1$ |
| Poisson | $\dfrac{L}{2\pi}$ | $\dfrac{L}{2\pi}\delta(\theta-\theta')$ | $L$ |

**Table 1:** The eigenphase repulsion in the CUE suppresses the first spectral moment.

- The connected eigenphase correlations take exponentially long times in the entropy to dominate the $\mathrm{SFF}_0$ [112] or correlation functions [157]. This is the so-called "dip time" at which the ramp overcomes the slope in the $\mathrm{SFF}_0$. Therefore, it is harder to signal random matrix behavior directly from these quantities,[44] given that they are dominated by disconnected correlations at intermediate times.[45] Here we see that a small time-dependent perturbation of $H_0$ will suppress the disconnected contributions in such a way that the approach of $F_1(t)$ to $F_1(\infty)$ is controlled by the connected correlations at intermediate $O(\log L)$ times. The onset of the connected correlations (i.e. the analog of the Thouless time) is the timescale $t_\star$ that we have defined in this paper.

**Stabilizing the moduli space of double cones.** In gravity, the ramp of $\mathrm{SFF}_0(t)$ can be computed semiclassically in a WKB approximation from the integration over the classical

---

[43] This can be traced back to the fact that the trajectory $\exp(-itH_0)$ explores a proper submanifold of the unitary manifold, as a consequence of the conservation of $H_0$ along the trajectory. This submainfold has dimension $L$, while the full space of unitaries $\mathsf{U}(L)$ has dimension $L^2$.

[44] This has also been pointed out recently in [158] in connection with the quantum chaos conjecture. In particular, [158] presented examples of fast scrambling Hamiltonians with Poisson spectral statistics. The quantum chaos conjecture asserts a relation between early-time chaos and late-time random matrix behavior; in this sense [158] pointed that the conjecture relies on the assumption that the Hamiltonian has some degree of locality.

[45] One might think that substracting the disconnected contributions from the $\mathrm{SFF}_0$ allows to isolate the universal random matrix correlations easily at much earlier times. However this requires the knowledge of the disconnected correlations to exponential precision. This is similar to 'unfolding' the spectrum. In the case of a general system without disorder this is difficult to do in practice.

moduli space of double-cone wormhole solutions [92]

$$\mathrm{SFF}_0(t) \supset \frac{1}{2\pi} \int_0^t \mathrm{d}T_{\mathrm{rel}} \int \mathrm{d}E \, . \tag{F.3}$$

The classical moduli space is parametrized by the canonically conjugate variables $\{T_{\mathrm{rel}}, E\} = 1$, where $E$ is the energy and $T_{\mathrm{rel}}$ is the relative timeshift between both boundaries.

Given the analysis on this paper, it is natural to expect that the connected contribution to the frame potential $F_1(t)$ is captured by a stabilized wormhole. In this case of infinite temperature, and where $U(t)$ corresponds to a Schrödinger picture random circuit, the wormhole in question is a stabilized version of the double-cone.[46]

Namely, assuming that backreaction effects are small, the stabilization arises from the boundary correlations, which generate time-independent effective interactions between both replicas. At early times, when the WKB approximation remains valid, the sources create a classical effective potential in the moduli space of solutions[47]

$$F_1(t) \supset \frac{1}{2\pi} \int_0^t \mathrm{d}T_{\mathrm{rel}} \int \mathrm{d}E \, e^{t \, \mathrm{Im} \, H_{\mathrm{eff},1}(T_{\mathrm{rel}}, E)} \, , \tag{F.4}$$

where we have used that the Lorentzian part of the effective Hamiltonian $H_0^1 - H_0^{\bar{1}}$ vanishes at the level of the classical moduli space of solutions. Recall that $\mathrm{Im} \, H_{\mathrm{eff},1}(T_{\mathrm{rel}}, E) \leqslant 0$.

However, the WKB approximation breaks down when $t$ is of the order of the gap of the effective Hamiltonian. In this case, one must consider the quantization of the moduli space of double cone solutions, in the form

$$F_1(t) \supset \mathrm{Tr}_{\mathrm{dc}} \exp\left(-\mathrm{i}t\hat{H}_1\right) , \tag{F.5}$$

where $\mathrm{Tr}_{\mathrm{dc}}$ corresponds to the trace on the Hilbert space $L^2(\mathbf{S}_t^1)$ of double cone configurations. In principle, solving for the spectrum of $\hat{H}_1$ in this Hilbert space gives the desired structure

$$F_1(t) \supset 1 + N_*(t)e^{-tE_{\mathrm{gap}}} , \tag{F.6}$$

where 1 comes from the ground state $|\mathrm{GS}_{\mathrm{dc}}\rangle$ and the second term comes from the first excited states. Here $N_*(t)$ is an oscillatory function which incorporates the effects of the real part of $\hat{H}_1$, and $E_{\mathrm{gap}}$ is minus the imaginary part of the eigenvalue of $H_1$ for the first excited states. The ground state $|\mathrm{GS}_{\mathrm{dc}}\rangle$ is the infinite temperature state, which in the moduli space of double cones is a linear superposition of double cones of all energies, with a wavefunction $\Psi_{\mathrm{GS}}(T_{\mathrm{rel}}) \propto \delta(T_{\mathrm{rel}})$.

From these considerations, it is reasonable to expect that the plateau value of the frame

---

[46] In the finite-temperature case we have shown in this paper that the connected frame potential $F_1(t)$ for the gradually cooled $U(\Gamma_t)$ corresponds to an eternal traversable wormhole, i.e., the Maldacena-Qi wormhole for near-extremal black holes.

[47] This is different from the mechanism in [154, 155] for the suppression of the ramp in the Loschmidt SFF [159]. Such a suppression arises from the "vacuum energy" (i.e. a flat potential in the $T_{\mathrm{rel}}$ direction) of the moduli space of double cones, sourced by the time-independent perturbation of the Hamiltonian with opposite signs of the couplings on both boundaries.

potential $F_1(\infty)$ for a few Brownian perturbations of $H_0$ is captured by a quantized double-cone wormhole configuration. Adding a large number of perturbations makes the situation much more subtle, given that backreaction effects must be included, and these are known to deform the tip of the double cone in the wrong direction [92, 155]. But in principle, there should be a way to smoothly connect the putative solution when a large number of perturbations are included to the infinite temperature version of an eternal traversable wormhole stabilized by matter.

Another interesting question is whether one can have a similar stabilization mechanism for the ramp which could explain the plateau of the $\mathrm{SFF}_0(t)$ gravitationally via an effective potential induced in the moduli space of double cone solutions which lifts the $U(1)$ zero mode. Without boundary sources, this effective potential must come from other gravitational effects. Since the plateau is attained at exponentially long times after the Thouless time, the gap of such an effective potential should be exponentially small in the entropy of the black hole.

**Other spectral measures.** In [92] the quantity $\mathrm{SFF}_t(n) \equiv \mathbb{E}[|\mathrm{Tr}\, U(t)^n|^2]$ was analyzed in relation to the SFF.[48] This quantity serves as an analog of the SFF for periodically-driven Floquet systems, where $n$ is the number of periods the system is evolved, and $U(t)$ is the time-evolution operator of a single period. Interpreted this way, this quantity provides a discrete analog of the ramp-plateau structure of the SFF, given that for the Haar ensemble $\mathrm{SFF}_{\mathrm{Haar}}(n) \equiv \mathbb{E}_{\mathcal{E}_{\mathrm{Haar}}}[|\mathrm{Tr}\, U^n|^2] = \min\{n, L\}$. For $U(t)$ given by a Brownian SYK time-evolution, the quantity $\mathrm{SFF}_t(n)$ admits a connected semiclassical saddle point configuration which spontaneously breaks $\mathbf{Z}_n \times \mathbf{Z}_n \to \mathbf{Z}_n$ [92]. The linear growth in $n$ then follows from the residual $\mathbf{Z}_n$ "zero mode". This is the discrete analog of the way the double cone spontaneously breaks $U(1) \times U(1) \to U(1)$ and produces a linear ramp.

The quantity $\mathrm{SFF}_t(n)$ partially characterizes the two-point correlation of the eigenvalue density of $U(t)$, $\mathbb{E}_{\mathcal{E}_U^t}[\rho_t(\theta)\rho_t(\theta')]$, by a Fourier transform in $n$. From the definition, it is clear that $\mathrm{SFF}_t(1)$ corresponds to the first frame potential for the ensemble of random circuits $U(t)$, $\mathrm{SFF}_t(1) = F_1(t)$. However, the $k$-th frame potential $F_k(t) = L^{-2k}\mathbb{E}_{\mathcal{E}_U^t}[|\mathrm{Tr}\, U(t)|^{2k}]$ for $k > 1$ that we consider in this paper depends on much finer features of the spectrum of $U(t)$, since it is controlled by the $2k$-th moment of the spectral density $\mathbb{E}_{\mathcal{E}_U^t}[\rho_t(\theta_1)...\rho_t(\theta_{2k})]$ instead. Ideally, one could study general spectral measures of the form $\mathbb{E}_{\mathcal{E}_U^t}[(\mathrm{Tr}\, U(t)^n)^k(\mathrm{Tr}\, U(t)^{\bar{n}})^{\bar{k}}]$ which provide a more complete characterization of the spectral distribution of $U(t)$.

# G  Distribution of the length of the longest increasing subsequences

In this appendix, we provide asymptotic formulas for the moments of the Haar distribution $f_{k,L}$. These moments have a combinatorial interpretation: the value of $f_{k,L}$ is the number of permutations $\pi \in \mathrm{Sym}(k)$ such that $\pi$ has no increasing subsequence of length greater than $L$ [119, 160, 161].

- **Sub-exponential regime:** $k \leqslant L$

---

[48] We thank Steve Shenker for pointing this out to us.

In this case $f_{k,L}$ is

$$f_{k,L} = k! \,. \tag{G.1}$$

For $k \leqslant L$ there are simply no subsequences of length greater than $L$. Therefore the number $f_{k,L}$ is the size of $\text{Sym}(k)$, i.e., $k!$. As explained in sections 3.1 and 4, $f_{k,L}$ is the dimension of the subspace generated by the invariant states, a subspace of the $L^{2k}$ dimensional replica Hilbert space. Therefore, it is clear that the behavior $k!$ for $f_{k,L}$ cannot last for $k \gg L^2$. Otherwise the invariant subspace would be larger than the replica Hilbert space, which is not possible. Modifications in the scaling of $f_{k,L}$ become important before that.

- **Intermediate regime:** $L < k \lesssim 4L^2$

Consider $k = L+1$. In this case there is one increasing subsequence of length greater than $L$, and one permutation which contains it, namely the identity permutation $e \in \text{Sym}(L+1)$. This implies that $f_{(L+1),L} = (L+1)! - 1$. Thus, $f_{k,L}$ starts growing less rapidly with $k$ when $k > L$.

In the regime $1 \ll L < k \lesssim 4L^2$ the asymptotic behavior of $f_{k,L}$ was determined in the seminal work of Baik, Deift and Johansson [162] (see also [160, 161] for additional details). The idea is to think of $f_{k,L}$ as a probability distribution. To do this, consider the uniform distribution defined in $\text{Sym}(k)$, namely $\text{Prob}(\sigma) = 1/k!$ for $\sigma \in \text{Sym}(k)$. Then, consider the length of the longest increasing subsequence $l_k$ associated with each element of the permutation group as a random variable. By definition, the cumulative probability distribution of $l_k$ is

$$\text{Prob}(l_k \leqslant L) \equiv \frac{f_{k,L}}{k!} \,. \tag{G.2}$$

Secondly, define the Tracy-Widom distribution [163]

$$F(t) = e^{-\int_0^\infty (x-t)\, u^2(x)\, dx} \,, \tag{G.3}$$

where $u(x)$ is the solution to the Painlevé II equation

$$u_{xx} = 2u^3 + xu \quad \text{with} \quad u \sim -Ai(x) \quad \text{as} \quad x \to \infty \,. \tag{G.4}$$

The Tracy-Widom distribution satisfies $F(t) \to 1$ as $t \to \infty$ and $F(t) \to 0$ as $t \to -\infty$.

Let $\chi$ be a random variable whose cumulative probability distribution is given by $F(t)$. Then [162] showed that as $k \to \infty$

$$\chi_k \equiv \frac{l_k - 2\sqrt{k}}{k^{1/6}} \sim \chi \quad \text{in the distributional sense} \,. \tag{G.5}$$

Equivalently, we have

$$\lim_{k \to \infty} \text{Prob}\left(\chi_k \equiv \frac{l_k - 2\sqrt{k}}{k^{1/6}} \leqslant t\right) = F(t) \quad \forall\, t \in \mathbf{R} \,. \tag{G.6}$$

From this formula we obtain

$$\text{Prob}(l_k \leqslant L) = \text{Prob}\left(\frac{l_k - 2\sqrt{k}}{k^{1/6}} \leqslant \frac{L - 2\sqrt{k}}{k^{1/6}}\right) = F\left(\frac{L - 2\sqrt{k}}{k^{1/6}}\right). \tag{G.7}$$

For $k < L^2/4$, in the limit $L \to \infty$ we simply obtain

$$\text{Prob}(l_k \leqslant L) \sim 1, \tag{G.8}$$

which from (G.2) implies that $f_{k,L} \sim k!$, in accord with (G.1). On the other hand, for $k > L^2/4$ we arrive at

$$\text{Prob}(l_k \leqslant L) \sim 0, \tag{G.9}$$

as $L \to \infty$. Since in this limit the denominator $k!$ in (G.2) diverges, we cannot conclude the scaling of $f_{k,L}$ from here. For that, we need to understand the subleading corrections to the scaling.

Consider the intermediate regime $k = L^2/x^2$, with $x \in \mathbf{R}$. Then for $x > 2$, i.e. $k < L^2/4$ we have [162]

$$\lim_{k \to \infty} \text{Prob}(l_k \leqslant L) = \lim_{k \to \infty} \text{Prob}(l_k \leqslant x\sqrt{k}) = 1 - e^{-\sqrt{k}I(x)}, \tag{G.10}$$

where $I(x) = 2x \cosh^{-1}(x/2) + 2\sqrt{x^2 - 4}$. Using (G.2) this leads to

$$f_{k,L} \sim k!\,(1 - e^{-\sqrt{k}I(x)}). \tag{G.11}$$

In the opposite case, i.e for $k = L^2/x^2$ with $1/2 \lesssim x < 2$ we have[49] [162]

$$\lim_{k \to \infty} \text{Prob}(l_k \leqslant L) = \lim_{k \to \infty} \text{Prob}(l_k \leqslant x\sqrt{k}) = e^{-k\,H(x)}, \tag{G.12}$$

where

$$H(x) = -\frac{1}{2} + \frac{x^2}{8} + \log\frac{x}{2} - \left(1 + \frac{x^2}{4}\right)\log\left(\frac{2x^2}{4 + x^2}\right). \tag{G.13}$$

This leads to

$$f_{k,L} \sim k!\,e^{-k\,H(x)}. \tag{G.14}$$

Summarizing, when $k = L^2/x^2$, with $1/2 \lesssim x$, the leading scaling is

$$f_{k,L} \sim k^k e^{-k(1+H(x))}. \tag{G.15}$$

Note that, replacing $k = L^2/x^2$, the scaling becomes $f_{k,L} \sim L^{2k}\,e^{-k(1+H(x)+2\log x)}$ so this asymptotic behavior already contains the correct scaling of the replicated Hilbert space dimension $L^{2k}$, with an exponentially suppressed factor $e^{-2k\log x}$.

---

[49] In [162] this formula is expected to be valid in the extended regime $0 \lesssim x < 2$. However, for $x < 0.48789$, the formula provides a value of $f_{k,L}$ than $L^{2k}$, which is not possible. We thus consider the formula in a more limited regime of validity.

- **Exponential regime:** $k \gg L^2/4$

The formula (G.5) gives a useless result when $k$ grows much faster than $L^2$. Recently, in [164], a different approach to compute the asymptotic scaling of $f_{k,L}$ was taken. The approach is based on Hayman's work [165] on the generalization of Stirling's formula to H-admissible functions, see e.g. [166, 167]. In this case one starts from the generating function of the cumulative probability distribution, namely

$$F_L(z) \equiv \sum_{k=0}^{\infty} \frac{\text{Prob}(l_k \leqslant L)}{k!} z^k , \qquad (z \in \mathbf{C}, L \in \mathbf{N}) . \qquad (G.16)$$

The explicit form of this function was found in [168]. It corresponds to

$$F_L(z^2) = D_L(z) , \qquad D_L(z) \equiv \det M(z) , \qquad M_{mn} \equiv I_{m-n}(2z) , \qquad m, n = 1 \cdots L , \qquad (G.17)$$

where $I_m$, $m \in \mathbf{Z}$, are the modified Bessel functions.[50] The work of [164] shows that $F_L(z)$ is an H-admissible function, and therefore it admits the following generalized Stirling's approximation. We first define the auxiliary function

$$a_L(r) \equiv r \frac{\mathrm{d}}{\mathrm{d}r} \log F_L(r) , \qquad b_L(r) \equiv r \frac{d}{dr} a_L(r) , \qquad r > 0 . \qquad (G.18)$$

Then for each $n \in \mathbf{N}$, the equation $a_L(r_{L,k}) = k$ has a unique solution $r_{L,k} > 0$ with $b_L(r_{L,k}) > 0$. The generalized Stirling's approximation is

$$\text{Prob}(l_k \leqslant L) = \frac{k! \, F_L(r_{L,k})}{r_{L,k}^k \sqrt{2\pi b_L(r_{L,k})}} \, (1 + o(1)) , \qquad \text{as} \quad k \to \infty . \qquad (G.19)$$

To be able to use this approximation, we need an explicit form for $F_L(z)$, which is evaluated as a complicated determinant. An approximation for $F_L(z)$ will suffice for the present purposes. To show that $F_L(z)$ are H-admissible functions, it is shown in [164] that

$$D_L(z) = F_L(z^2) = c_L \, z^{-\nu_L} e^{\tau_L z} (1 + O(z^{-1})) , \qquad (z \to \infty , \ |\arg z| \leqslant \frac{\pi}{2} - \delta) , \qquad (G.20)$$

where

$$c_L = (L - 1)! , \qquad \nu_L = L^2/2 , \qquad \tau_L = 2L . \qquad (G.21)$$

From the definitions of the auxiliary functions, this implies that for $r \to \infty$ we also have

$$a_L(r) = L\sqrt{r} - L^2/4 , \qquad b_L(r) = \frac{L}{2}\sqrt{r} . \qquad (G.22)$$

Solving the equation $a_L(r) = k$ in this limit leads to

$$r_{L,k} = \frac{k}{L^2} \left( k + \frac{L^2}{2} \right) . \qquad (G.23)$$

---

[50] The formula (G.5) was derived from a double-scaling limit of (G.17)

We can now evaluate formula (G.19) using the leading term in $r_{L,k}$. As explained in [164], this approximation is only valid for $k \gg L^4$. This leads to

$$f_{k,L} = k! \operatorname{Prob}(l_k \leqslant L) = \frac{(L-1)! \, (k!)^2 \left(\frac{e}{k}\right)^{2k} L^{2k+L^2/2}}{(2\pi)^{L/2} \, 2^{L^2/2} \sqrt{\pi} \, k^{(L^2+1)/2}} (1 + o(1)) \,. \tag{G.24}$$

The form of this expression can be further simplified using Stirling's formula,

$$f_{k,L} = \frac{(L-1)! \, L^{2k+L^2/2}}{(2\pi)^{(L-1)/2} \, (2k)^{(L^2-1)/2}} (1 + o(1)) \,. \tag{G.25}$$

This expression has also been derived in [169].

In the limit we are interested in, $k \gg L^2 \gg 1$, the asymptotic expression for the moments becomes

$$f_{k,L} \sim L^{2k} \, k^{-L^2/2} \,. \tag{G.26}$$

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
