# Peer review of "Random Circuits in the Black Hole Interior"

_SciPost Physics_

## Round 2 · Referee Report · Anonymous (Referee 1) · 2025-3-6

Strengths

  • Papers explores the black hole interior, which is an important open question in holography.

  • Very well written.

Weaknesses

Some aspects related to the setup are confusing.

Report

This is an interesting paper that attempts to quantify the extent to which the black hole interior describes randomness in the CFT state. Previous considerations studied the time-evolution of the thermofield-double state, and related the growth of the wormhole length in the black hole interior to the complexity of the quantum state, which increase under time evolution. However, these ideas are hard to make precise due to the challenge of making precise definitions of complexity in CFTs. Here, the authors explore another idea. They consider a family of TFD-like states, which are TFD states perturbed by sources for simple (low-$\Delta$) operators. The operators are inserted on particular time-contours to produce states of constant energy, even with the insertion of operators, and the sources are taken to be random with a Gaussian time-organized distribution. The averages gives rise to Einstein-Rosen caterpillars, modifications of the TFD state where the wormhole length is larger, and the inside geometry contains matter insertions correlated on some length-scale.

The authors show this ensemble of states, once the Gaussian average is performed, gives a quantum state $k$-design for true Haar random states of the CFT. This means that at large effective time of the circuit, the ensemble of states becomes k-copy indinstinguishable from a Haar random state.

This is a great paper and a useful addition to the literature. I would be happy to recommend it for publication, but there are a set of confusing points (most of them related to their setup) that the authors should first clarify. I list them below;

Requested changes

  1. Above (1.5), the authors talk about the "difference" between $\rho_{eq}$ and the thermal state. Is this meant to be a precise statement or simply a statement that one difference between the two states is the variance in energy. Moreover, the energy variance is an important piece of the thermal states, and is captured by the specific heat. Should one not be worried that the ensemble of states have a different specific heat? I could not find a discussion of this in section 2. The authors should clarify.

  2. I am confused about FIg. 5 and the way to cut it to prepare the state that the authors want to study. The idea from Fig. 3 is very clear, one starts adding sources at $t=0$ and this modifies the state and makes matter start to fall in. In Fig. 5, where should one cut the Euclidean contour to produce the state that we want to study? Following the idea of Fig. 3, I would have said right after the last time fold on contour 1. But the words around here, and also equation (2.8), make it look like they would cut Fig. 5 in the two places where $\beta/2$ is written. This is confusing, and the authors should clarify this point.

  3. I also do not understand why there is no matter outside the horizon for the states described by the authors. Do they claim this is true for a single realization, or is this a property of the ensemble? If it is true for a single realization, I do not understand why. Inserting sources for matter fields always comes with tails. Even if you insert them in the middle of the Euclidean contour that prepares the TFD state. These tails could be suppressed by considering very heavy operators $O_\alpha$ with large scaling dimension, but I did not think this is what the authors were doing. If it is a property of the ensemble, then I would like to understand what can be said about the variance of the matter fields outside the horizon, even if they vanish on average. The authors should clarify these points.

  4. The caption of Fig. 6 is also confusing. The authors write the operator $e^{-\tau H_{eff}} \ket{TFD}$. I think here they should not write the ket state, if they talk about a property of an operator. There also seems to be some confusion in notation between $t$ and $\tau$ here). Is $\tau$ not the same as $t$ in (2.15)? Finally, I do not understand why the authors call this geometry a Euclidean wormhole. Is there not a single Euclidean boundary for the geometry $M$? This definitely seems to be the case for the top left figure. The authors should clarify these points.

  5. I am also confused about the statement that the authors want $K\sim N^2$. I thought $K$ labelled the number of distinct operators that one would insert. In bottom-up models, one considers a finite number of matter fields. Even in top-down models, there can formally be an infinite number of matter fields due to the KK reduction on internal manifolds, but one should be very careful in trying to scale the number of fields with $N$ and still trusting supergravity. This is an important distinction with the SYK model that has $N$ fermions to play with, which is often ignored in its application to holography due to the dominance of the Schwarzian dynamics, but here it is important. If the authors really need $K\sim N^2$, I am confused whether one can ever apply the analysis of their paper to any realistic example of AdS/CFT in $d>1$.

  6. Above (2.21) the authors mention that the isometry is fake and only emergent after taking the ensemble. Can the authors comment on the variance of this isometry? If the variance is small, I suppose then that all realization themselves have an approximate isometry. If it is large, then the isometry should not be trusted.

  7. The scaling with $k$ found by the authors comes from the number of excited states, which is factorial in $k$. The authors seem to not make a big deal of this, and treat a factorial growth essentially as an exponential growth, but I would argue that it is much faster. Of course, if you take a log, then the enhancement is only logarithmic but I think the authors should discuss this enhancement at least a little bit more than what they do, which is currently only in the form "up to log corrections" or "approximately linear in k". Does is not have any physical importance?

Recommendation

Ask for minor revision

  • validity: top
  • significance: high
  • originality: high
  • clarity: top
  • formatting: perfect
  • grammar: perfect

Author:  Martin Sasieta  on 2025-06-05  [id 5547]

(in reply to Report 1 on 2025-03-06)

We thank the referee for the good questions that have led to an improvement of the manuscript. Here are our detailed responses:

  1. Yes, the energy variance is different from the canonical ensemble in general. This is the main difference in so far as the energy distribution is well peaked and thus described just by its mean and variance. As to whether this is a significant difference, it depends. Taking the example of energy eigenstates, we know via ETH that local observables will be approximately thermal to a good approximation, and moreover, being intensive quantities, they will be insensitive to the precise ensemble in the thermodynamic limit. By contrast the energy variance is identically zero. Yet both the canonical ensemble and energy eigenstates are thought to have the same coarse grained black hole geometry, so in this sense the exterior geometry does not depend on having a particular energy variance, at least to leading order in $N$. The outcome of a physical measurement of the heat capacity obtained by coupling the system to a reservoir is a distinct issue in general, although of course we agree that the variance is tied to the heat capacity (at constant volume) in the canonical ensemble. If a particular value of the energy variance is crucial for the application of interest, there are methods available for adjusting it, for example, by making the amount of imaginary time evolution have some random component. We discuss several of these points and attempt to further clarify the issues raised by the referee in improved text in section 2.2, 2.4, and appendix C.

  2. In page 11, when introducing the ensemble of states before (2.8) we have included an explanation for our choice of how to apply the finite-temperature random circuit.

  3. If the equilibrium state overlaps substantially with the thermal state in the thermodynamic limit, we do not expect that there is classical matter outside of the horizon even for single realizations. One reason is that in this case the average geometry is that of Schwarzschild and any classical matter must have a positive (null) energy, so its backreaction effects cannot average out to zero. However, as shown in Ref. [61], typically for this to be the case one needs perturbations $\mathcal{O}_\alpha$ which are relevant enough. In higher dimensions these would only act deep in the bulk, and thus they would not spread out substantially to the exterior. However, as we explain in section 2.2., 2.4 and appendix C, this is not generally the case. If this is not the case, as the referee points out, we do expect deviations from thermality which could arise at leading order, signaled by classical matter outside of the black hole. We have added a clarification of this in the caption of Fig. 6.

  4. We have modified the caption of Fig. 6 to correct that $\exp(-\tau H_{\text{eff},1})|{\text{TFD}}\rangle$ is the Euclidean evolution of a state, as the referee points out. We have also added the relation between the coordinate $\tau$, the total boundary time $t$ and the onset time $t_\star$ of the approximate time-translation symmetry. We have also clarified that $M$ has the geometry of an Euclidean wormhole locally in this region, while, as the referee points out, the global topology of $M$ is that of a disk (times a sphere).

  5. We have added footnote 11 to clarify how it is possible to take $K \sim N^2$ perturbations in AdS/CFT without leaving the supergravity regime. As the referee points out, a primary with conformal dimension $O(N^2)$ is beyond supergravity. The point is that it is not necessary that the drive operators $\mathcal{O}_\alpha$ are different primaries. They could all be coming from the same primary operator, applied at different spatial points on the sphere, with independent Brownian couplings at each point. They could also be non-local operators on the sphere which create particles deep in the bulk. With this choice, one creates a large number of perturbations which are well described by supergravity but which together backreact on spacetime. As we mention in footnote 14, constructing these solutions explicitly in higher dimensional AdS/CFT is an interesting open problem. Our symmetry-based argument suffices for the purposes of this paper, and it relies on the assumption outlined on page 13. Such an assumption is supported by general microscopic studies such as Ref. [99], where they argue that $H_{\text{eff},1}$ has a unique GS similar to the TFD under very general ETH-like ansatze for the $\mathcal{O}_\alpha$. Thus, as long as the assumption is correct for choices of the $\mathcal{O}_\alpha$, the results of this paper should apply in higher dimensional AdS/CFT as well.

  6. Above (2.21) we have clarified that the Lorentzian eternal traversable wormhole relies on the exact isometry being present in the Euclidean section. If this symmetry is broken, even weakly as it is for individual instances, then this analytic continuations generally leads to a complex metric. Perhaps the complex parts of the metric are small, but naively one loses this interpretation.

  7. We have expanded Sec. 3.5 to include the role of the logarithmic corrections to the scaling. We have made more transparent that we are using a leading asymptotic form of this relation in the introduction and conclusions. We have improved the presentation of the relation in the introduction and conclusions accordingly.

---

## Round 2 · Referee Report · Anonymous (Referee 2) · 2025-4-8

Report

This is a very interesting paper. The authors gave a novel construction of long wormholes supported by matter and rigorously established the complexity property of an ensemble of such states. I have one minor comment. It would be helpful to compare the construction given in this paper and that of figure 7 in https://arxiv.org/pdf/1312.3296.

Recommendation

Publish (easily meets expectations and criteria for this Journal; among top 50%)

  • validity: -
  • significance: -
  • originality: -
  • clarity: -
  • formatting: -
  • grammar: -

Author:  Martin Sasieta  on 2025-06-05  [id 5548]

(in reply to Report 2 on 2025-04-08)

We thank the referee for pointing out this similarity. In footnote 14 we added a comment comparing the average geometry with the setup of Fig. 7 of Ref. [14].

---

## Editorial Decision

resubmitted